# The transition from monocyte to tissue-resident macrophage requires DHPS

Gustavo E. Carrizo[1], Pianpian Lin[1,11], Seung Hyun Lee[1,11], Kevin Shenderov[1,2], Camille Blériot[3,4], Minsun Cha[1], Lena Schimmelpfennig[5], Zhen Shen[1], Nikki van Teijlingen Bakker[5], Katarzyna M. Grzes[1], Beth Kelly[1], Niloufar Safinia[6], Kate L. Schole[7], Yaarub Musa[5], Gerhard Mittler[5], Yoh Zen[6], Edward J. Pearce[1,8], Florent Ginhoux[3,4], David E. Sanin[1], Daniel J. Puleston[1,5,9 ✉] & Erika L. Pearce[1,5,10 ✉]

Tissue-resident macrophages (RTMs) form during embryogenesis, self-renew locally, and regulate tissue homeostasis by clearing dead cells and debris[1–6]. During tissue damage, however, bone-marrow-derived monocytes enter tissues and differentiate into RTMs, repairing the tissue and replenishing macrophages in the niche[1]. The universal cell-intrinsic mechanisms that control the monocyte-to-RTM transition and the maintenance of mature RTMs across tissues remain elusive[3]. Here we show that deoxyhypusine synthase (DHPS), an enzyme that mediates spermidine-dependent hypusine modification of translation factor eIF5A[5,7], is required for RTM differentiation and maintenance. Mice with myeloid cell lack of DHPS (*Dhps*-ΔM mice) had a global defect in RTMs across tissues, resulting in persistent but ultimately futile monocyte influx. Transcriptional analyses of DHPS-deficient macrophages indicated a block in their ability to differentiate into mature RTMs, whereas proteomics revealed defects in cell adhesion and signalling pathways. Sequencing of ribosome-engaged transcripts identified a subset of mRNAs involved in cell adhesion and signalling that rely on DHPS for efficient translation. Imaging of DHPS-deficient macrophages in tissues showed differences in morphology and tissue interactions, which were correlated with their failed RTM differentiation. DHPS-deficient macrophages were also defective in critical homeostatic RTM functions including efferocytosis and tissue maintenance. Together, our results demonstrate a cell-intrinsic, tissue-agnostic pathway that drives differentiation of monocyte-derived macrophages into RTMs.

Essentially all tissues are populated by self-renewing long-lived tissue-resident macrophages (RTMs) that clear damaged cells and debris. In the brain, RTMs prune neuronal synapses, in the lung they remove surfactant, and in the liver they eliminate senescent red blood cells (RBCs). The core responsibility of RTMs is to maintain tissue health and homeostasis through these homeostatic functions[1]. RTMs colonize tissues during embryogenesis, with distinct regulatory nodes controlling acquisition and maintenance of a given tissue's macrophage program. For instance, in the lung, GM-CSFR signalling is essential for alveolar macrophage (lung RTM) development, whereas transcription factor GATA-6 controls peritoneal cavity RTM differentiation[2]. However, we have limited understanding of regulatory factors that are important for RTMs irrespective of their tissue of residency and how these programs influence the differentiation of monocytes into RTMs across tissues[3,4].

Polyamines are ubiquitous metabolites whose biosynthesis is augmented in metabolically active cells. Polyamines have critical roles in many cell functions, including in translation, during which spermidine is the substrate for a two-step reaction mediated first by deoxyhypusine synthase (DHPS) and then by deoxyhypusine hydroxylase to post-translationally 'hypusinate' eIF5A, a process in which a conserved lysine is converted into the amino acid hypusine[5,7]. Historically, eIF5A has been considered to be the only protein that contains hypusine, with the only functions of DHPS and deoxyhypusine hydroxylase being to hypusinate eIF5A[8]. Hypusinated eIF5A enhances the translation efficiency of certain mRNA transcripts that lead to ribosome stalling[9], including those with polyproline motifs[10,11], but it has been unclear exactly which transcripts are within this set and how this process is affected in diverse biological contexts in mammalian cells. Previous work by our group and others has shown that polyamine

[1]Bloomberg-Kimmel Institute for Cancer Immunotherapy, Department of Oncology, Johns Hopkins University School of Medicine, Baltimore, MD, USA. [2]Division of Pulmonary and Critical Care Medicine, Department of Medicine, Johns Hopkins University School of Medicine, Baltimore, MD, USA. [3]Singapore Immunology Network (SIgN), Agency for Science, Technology and Research, Singapore, Singapore. [4]INSERM U1015, Gustave Roussy Cancer Campus, Villejuif, France. [5]Max Planck Institute of Immunobiology and Epigenetics, Freiburg im Breisgau, Germany. [6]Department of Inflammation Biology, Institute of Liver Studies, School of Immunology and Microbial Sciences, James Black Centre, King's College London, London, UK. [7]Department of Molecular Biology and Genetics, Johns Hopkins University School of Medicine, Baltimore, MD, USA. [8]Department of Molecular Microbiology and Immunology, Bloomberg School of Public Health, Johns Hopkins University, Baltimore, MD, USA. [9]Precision Immunology Institute and the Tisch Cancer Institute, Icahn School of Medicine at Mount Sinai, New York, NY, USA. [10]Department of Biochemistry and Molecular Biology, Bloomberg School of Public Health, Johns Hopkins University, Baltimore, MD, USA. [11]These authors contributed equally: Pianpian Lin, Seung Hyun Lee. ✉e-mail: daniel.puleston@mssm.edu; epearce6@jhmi.edu

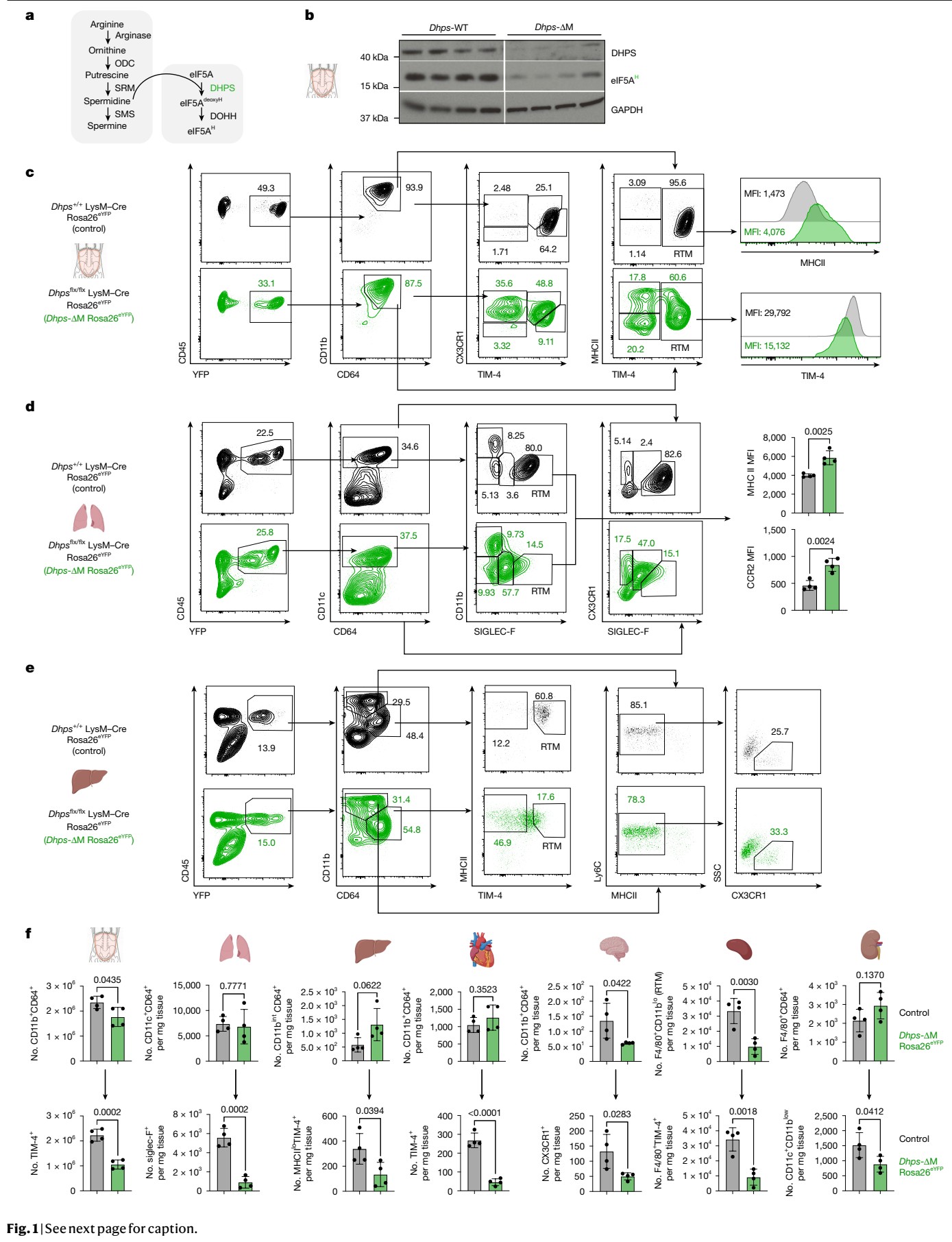

**Fig. 1 | See next page for caption.**

**Fig. 1 | *Dhps*-ΔM mice have a defect in RTMs. a**, Polyamine biosynthesis and eIF5a hypusination pathways. **b**, Immunoblot of indicated proteins on FACS-sorted F4/80[hi]CD11b[+] peritoneal macrophages from *Dhps*-WT and *Dhps*-ΔM mice. **c**–**e**, Representative flow cytometry plots of macrophage subpopulations in peritoneal cavity (**c**), lungs (**d**) and liver (**e**) in *Dhps*[+/+] Rosa26[eYFP] (control) or *Dhps*-ΔM Rosa26[eYFP] reporter mice. CD45[+]YFP[+] cells were gated on singlets and live cells. **f**, Absolute numbers of macrophages and RTMs across tissues from *Dhps*[+/+] Rosa26[eYFP] (control) or *Dhps*-ΔM Rosa26[eYFP] reporter mice. Images reflect peritoneal cavity, lung, liver, heart, brain, spleen and kidney. Representative plots and graphs summarize results of at least two independent experiments. Data are mean ± s.d., representative of *n* = 4 biological replicates. Statistical analyses were performed using two-tailed *t*-tests; *P* values are shown. DOHH, deoxyhypusine hydroxylase. Illustrations in **b**–**f** were created using BioRender (https://biorender.com).

metabolism directs T cell lineage choices and pathogenic potential in inflammation[12,13], and that hypusinated eIF5A boosts respiration, in part by enhancing the translation of certain tricarboxylic acid cycle enzymes, and as such contributes to macrophage alternative activation[14]. Other studies have investigated the role of DHPS deficiency in myeloid cell inflammation, observing that in obese mice, DHPS deletion suppresses inflammatory macrophage accumulation in adipose tissue and improves glucose tolerance[15], and that myeloid cell DHPS expression is required to clear gastrointestinal pathogens by controlling translation of antimicrobial factors[16]. Here we investigated the role of DHPS in macrophages in steady state using mouse models of myeloid or macrophage-specific gene deletion and found a striking deficiency in mature RTMs across tissues in the absence of DHPS.

## *Dhps*-ΔM mice have a global RTM defect

To examine the polyamine–hypusine pathway (Fig. 1a) in macrophages, we bred mice with *loxP*-flanked exons 2–7 of *Dhps*[17] with mice expressing LysM–Cre to generate mice with DHPS deleted in myeloid cells, including monocytes and macrophages (*Dhps*-ΔM mice). We crossed these mice to Rosa26[eYFP] mice to generate *Dhps*[+/+] LysM–Cre Rosa26[eYFP] (control) and *Dhps*-ΔM Rosa26[eYFP] mice, in which YFP reports Cre expression and thus DHPS deletion when loxP is present. DHPS was deleted in macrophages from *Dhps*-ΔM mice, leading to decreased eIF5A hypusine expression (Fig. 1b and Supplementary Figs. 1 and 2). We gated on YFP[+] cells and assessed macrophages across tissues. Whereas macrophage number (defined by F4/80 and CD11b, CD11c or CD64, depending on tissue) differed to some extent between 8–10-week old (control) and *Dhps*-ΔM Rosa26[eYFP] mice, macrophages were present across all tissues in both genotypes when measured by flow cytometry (Fig. 1c–f and Extended Data Figs. 1–3) and by imaging (defined by F4/80 or IBA1) (Supplementary Figs. 3–6). However, *Dhps*-ΔM Rosa26[eYFP] mice exhibited substantial defects in RTMs in the peritoneum (TIM-4[+]), lung (Siglec-F[+]), liver (TIM-4[+]), heart (TIM-4[+]), brain (CX3CR1[+]), spleen (TIM-4[+]) and kidney (CD11c[+]CD11b[low])[18–22] compared with controls (Fig. 1c–f and Extended Data Figs. 1–3), findings supported by imaging of TIM-4[+] RTMs in the liver and spleen (Extended Data Fig. 4).

## *Dhps*[−/−] monocytes do not form mature RTMs

We focused on RTMs in peritoneum, lung and liver, distinct niches in which LysM–Cre is strongly expressed (Supplementary Fig. 7). RTMs declined over time in *Dhps*-ΔM mice (Fig. 2a). In early adulthood, many RTM reservoirs are sustained through self-renewal, receiving little contribution from monocytes, the RTM precursor cell. However, unlike in control mice, we observed persistent monocytes in the peritoneal cavity of *Dhps*-ΔM mice that we reasoned could reflect sensing of a lack of mature TIM-4[+] RTMs (Fig. 2b). We performed parabiosis (Extended Data Fig. 5a) to evaluate the monocyte contribution to RTMs. RTM pools in lung, peritoneal cavity and liver of the wild-type (WT):WT mice contained only host cells, with no contribution from the congenically marked WT parabiont. However, among the WT:*Dhps*-ΔM mice, RTM niches in the *Dhps*-ΔM mice comprised cells from the WT parabiont (Fig. 2c and Extended Data Fig. 5b,c). These data indicate that the tissues of *Dhps*-ΔM mice, despite being populated with F4/80[+] macrophages, received continual monocytic influx, which we propose was driven by a dearth of fully developed RTMs. Furthermore, experiments with bone marrow chimeras revealed that monocytes from CD45.2[+] *Dhps*-ΔM bone marrow precursors failed to repopulate RTM pools in irradiated WT recipient mice when competing with CD45.1[+] *Dhps*-WT monocytes (Fig. 2d and Extended Data Fig. 5d,e). These data suggest that the RTM defect in *Dhps*-ΔM mice results from poor macrophage survival in the tissue, which drives constant monocytic influx.

To capture what happened to RTM niches after acute macrophage depletion, we injected clodronate liposomes (CL) into the trachea of mice to deplete macrophages in the bronchoalveolar space. In control mice, CL led to rapid macrophage depletion followed by monocyte recruitment and differentiation into SIGLEC-F[+]CD11b[low] alveolar macrophages over time (Fig. 2e,f). In *Dhps*-ΔM mice, however, monocytes entered and differentiated into a persistent population of CD64[+]CD11c[+] macrophages but failed to re-establish the local SIGLEC-F[+]CD11b[low] lung RTM pool (Fig. 2e,f). Also evident was a persistent influx of CD11b[+]Ly6c[+] monocytes (Fig. 2g). In control mice, CL injected intraperitoneally (i.p.) rapidly depleted peritoneal macrophages[23], followed by monocyte recruitment and differentiation into macrophages (Extended Data Fig. 5f,g). In *Dhps*-ΔM mice, however, monocytes persisted in the tissue after CL but failed to re-establish the local RTM population (Extended Data Fig. 5f,g). Overall, these data show a collapse in the tissue-residency potential of several DHPS-deficient RTM populations, resulting in persistent but ultimately futile monocytic infiltration to restore the RTM reservoir.

## *Dhps*[−/−] macrophages have survival defects

The RTM defect in *Dhps*-ΔM mice manifests as continual turnover of RTM reservoirs, compelled by low macrophage survival driving continual monocytic influx to tissues. We assessed Ki-67 and active caspase-3 expression, proliferation and impending cell death indicators, respectively, in F4/80[+]TIM-4[+] peritoneal cavity and F4/80[+] CD11c[+] lung macrophages in control and *Dhps*-ΔM mice. DHPS-deficient macrophages expressed less Ki-67 and more active caspase-3 (Fig. 2h,i). To probe proliferation and survival, we administered IL-4 complexes (IL-4c), which drive accumulation of peritoneal RTMs through self-renewal without recruitment from blood monocytes[24], plus EdU, into the peritoneal cavity of *Dhps*-WT and *Dhps*-ΔM mice. Control but not DHPS-deficient macrophages accumulated and proliferated (Extended Data Fig. 5h,i). Adoptive transfer of equal numbers of CellTrace Violet (CTV)-labelled peritoneal macrophages from *Dhps*-WT and *Dhps*-ΔM mice into congenic recipients showed that DHPS-deficient cells proliferated less than controls, and, although there was some CTV dilution, macrophages did not accumulate (Extended Data Fig. 5j,k), indicating increased cell death. Thus, DHPS-deficient macrophages have a decreased capacity for proliferation coupled with increased death, leading to defective tissue persistence. Notably, there was no difference in peritoneal or liver RTMs, or monocyte expression, between genotypes of colony stimulating factor 1 receptor (CSF1R), the ligand for which is a critical growth and survival factor for monocyte–macrophage development[21] (Supplementary Fig. 8).

## Mature RTM persistence requires DHPS

*Dhps*-WT and *Dhps*[flx] mice crossed to CX3CR1–ERT2cre-Rosa26[eYFP] mice generated offspring in which DHPS could be inducibly deleted

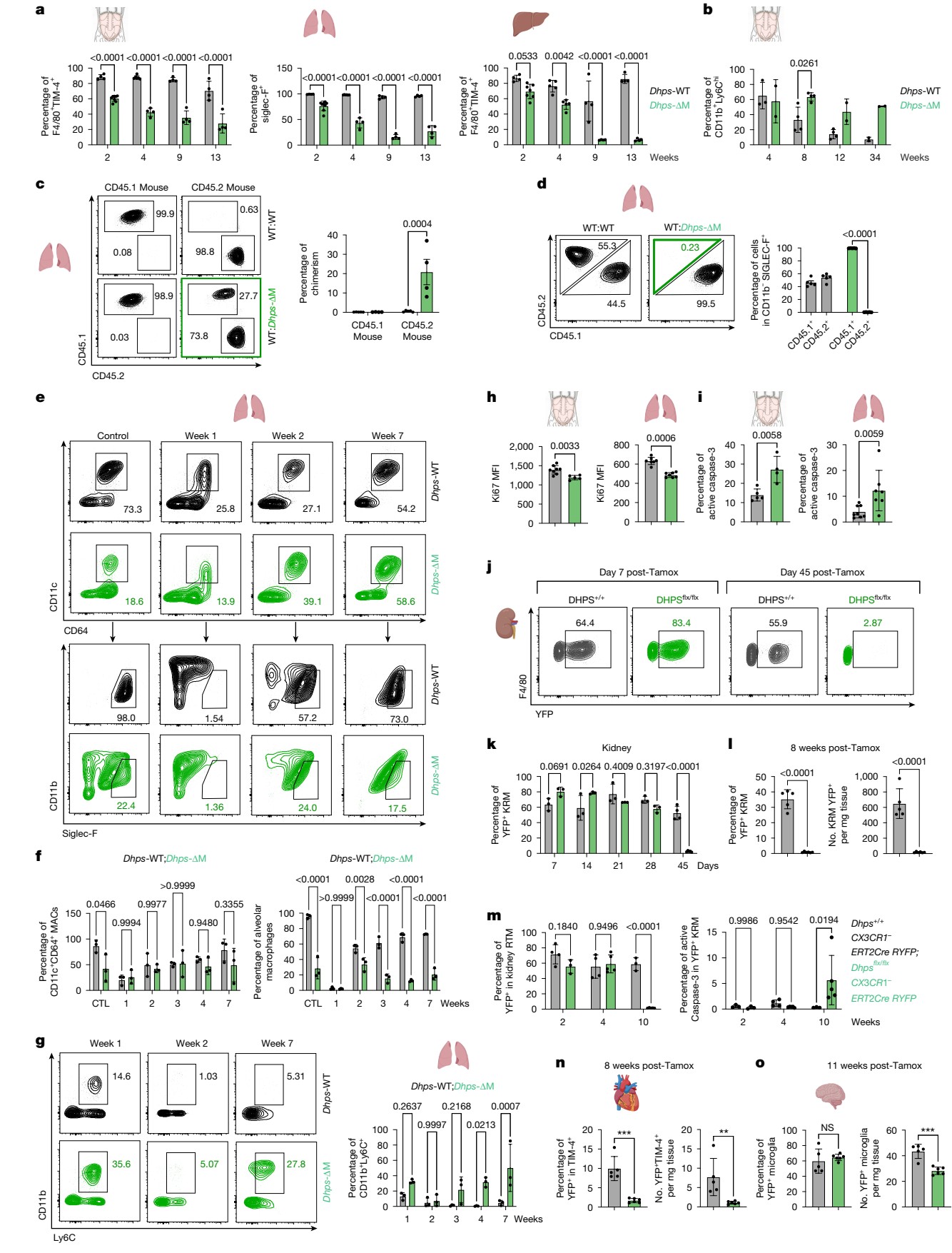

**Fig. 2** | See next page for caption.

**Fig. 2 | DHPS is essential for monocyte-to-RTM maturation and macrophage survival. a,b**, Flow cytometry of RTMs (**a**) and Ly6C[+] monocytes (**b**); gated: Lin[−](CD3, CD19, NK1-1)F4/80[−]. **c**, CD45.1 versus CD45.2 8 weeks postparabiosis, lung RTMs (F4/80[+]CD64[+]CD11b[−]Siglec-F[+]) and chimerism (%), WT:*Dphs*-ΔM. **d**, Bone marrow (1:1) CD45.1 plus CD45.2 *Dphs*-WT or CD45.1 WT plus CD45.2 *Dphs*-ΔM cells were infused into irradiated CD45.1 WT recipients; chimerism 12 weeks later in lung RTMs is shown. **e–g**, CD11c[+]CD64[+] macrophages (**e**), CD11b[low]Siglec-F[+] macrophages (**f**) and Ly6C[+] monocytes (**g**) in BAL after intratracheal CL; control (CTL) mice (**e** and **f**) received PBS; BAL was collected at 7 weeks. **h,i**, Ki-67 MFI (**h**) and percentage of active caspase-3 (**i**) in peritoneal (F4/80[+]TIM-4[+]) and alveolar (F4/80[+]CD11c[+]) macrophages. **j–l**, Frequencies (**j,k**) and numbers per milligram tissue (**l**) of YFP[+] kidney RTMs (KRM; F4/80[+]CD64[+] CD11c[+]CD11b[low]). **m**, Active caspase-3 in YFP[+] KRM. **n,o**, YFP[+] cardiac RTMs (F4/80[+]TIM-4[+]) (**n**) and YFP[+] microglia (CD11b[+]CD45[int]) (**o**) post-tamoxifen i.p. in *Dphs*[+/+] (control) and *Dphs*[flx/flx] CX3CR1–ERT2cre-Rosa26[eYFP] mice. Representative plots with graphs summarize results of two or more experiments,

except in **b** and **e**–**g**, in which they represent one experiment. Data are presented as the mean ± s.d. In **a**, *n* = 4 (2, 9, 13 weeks) and 6 (4 weeks), *Dphs*-WT; and *n* = 8 (2 weeks) and 4 (4, 9, 13 weeks), *Dphs*-ΔM for peritoneum, lung. For liver, *n* = 4 (2, 9, 13 weeks) and 6 (4 weeks), *Dphs*-WT; and *n* = 7 (2 weeks), 5 (4 weeks), and 4 (9, 13 weeks), *Dphs*-ΔM. In **b**, *n* = 3 (4 weeks), 4 (8, 12 weeks) and 3 (34 weeks), *Dphs*-WT; and *n* = 2 (4 weeks), 4 (8 weeks) and 2 (12, 34 weeks), *Dphs*-ΔM. In **c**, **d**, **f** and **k**, *n* = 4, 5, 3, and 3, respectively. In **h**, *n* = 8, *Dphs*-WT; *n* = 5, *Dphs*-ΔM, peritoneal; and *n* = 7, *Dphs*-WT and *Dphs*-ΔM alveolar macrophages. In **i**, *n* = 5, *Dphs*-WT; *n* = 4, *Dphs*-ΔM, peritoneal; and *n* = 8, *Dphs*-WT; and *n* = 7, *Dphs*-ΔM alveolar. In **l**, *n* = 5 (control) and *n* = 6 (*Dphs*[flx/flx]). In **m**, *n* = 4 (2, 4 weeks) and 3 (10 weeks), control; and *n* = 3 (2) and 5 (4, 10 weeks), *Dphs*[flx/flx]. In **n** and **o**, *n* = 5 (control), *n* = 6, *Dphs*[flx/flx]. *n* indicates biological replicates. Statistics: two-tailed *t*-tests, two-way analyses of variance; *P* values are shown. NS, not significant; Tamox, tamoxifen. Illustrations in **a**–**e**, **g**–**j**, **n** and **o** were created using BioRender (https://biorender.com).

in cells expressing CX3CR1. We administered tamoxifen to *Dphs*-WT CX3CR1–ERT2cre-Rosa26[eYFP] and *Dphs*[flx/flx] CX3CR1–ERT2cre-Rosa26[eYFP] mice to delete DHPS in mature kidney RTMs[25] in adult mice. YFP expression confirmed Cre-mediated gene deletion after tamoxifen. Kidney RTMs were lost by day 45 post-tamoxifen in *Dphs*[flx/flx] CX3CR1–ERT-2cre-Rosa26[eYFP] mice (Fig. 2j–l), a phenotype that was confirmed by imaging (Extended Data Fig. 6) and was correlated with increased active caspase-3 expression in any surviving YFP[+] DHPS-deficient RTMs 10 weeks post-tamoxifen (Fig. 2m). We also observed a trend of increased active caspase-3 expression in YFP[+] DHPS-deficient RTMs at 5 weeks post-tamoxifen by imaging (Supplementary Fig. 9). Microglia (brain RTMs), and heart RTMs to some extent, also expressed CX3CR1 (ref. 26). Numbers and frequency of YFP[+] heart RTMs declined at 8 weeks, (Fig. 2n), whereas microglia numbers were diminished at 11 weeks (Fig. 2o and Supplementary Fig. 10) post-tamoxifen. Thus, mature RTMs rely on DHPS for persistence.

## Blocked RTM differentiation

We next performed single-cell RNA sequencing (scRNA-seq) of peritoneal exudate cells from *Dphs*-WT and *Dphs*-ΔM mice. Data were subsetted to include only macrophages, on the basis of a range of expression of key macrophage markers (*Adgre1*, *Csf1r*, *H2-Ab1*, *Cd68*, *Lyz2*, *Itgam*, *Mertk*). Initial analysis revealed differential clustering between genotypes (Fig. 3a), including diminished frequency of canonical, mature RTMs in *Dphs*-ΔM mice as defined by *Timd4* expression (cluster 3) (Fig. 3a–c). Cells from *Dphs*-ΔM mice also had distinct populations that were largely not observed in controls (clusters 1 and 2) (Fig. 3a,b), of which cluster 1 was associated with *Ccr2* expression, a marker of recently infiltrated monocyte-derived macrophages[27] (Fig. 3c). We propose that cluster 1 represents monocyte-derived macrophage infiltration triggered by a scarcity of mature RTMs in *Dphs*-ΔM mice. We then assessed the top differentially expressed genes (DEGs) among all clusters (Supplementary Fig. 11) and found that cluster 1 showed increased expression of *Itgb5* (ref. 28), *Cx3cr1* (ref. 29) and *Ly6c2* (ref. 30), genes associated with protective macrophage phenotypes and cell interactions within tissues. Cluster 2, a population with high expression of *Adgre1* (F4/80) that was significantly expanded in *Dphs*-ΔM mice, had no *Ccr2* and *Timd4* expression and probably represents an immature monocyte-derived macrophage population unable to develop to mature RTMs (cluster 3). Again, when assessing the top DEGs (Supplementary Fig. 11) we identified *Cxcl2* (ref. 31) and *Folr2* (ref. 32) expression in cluster 2, and *Timd4* along with *Tgfb2* (ref. 33), *Wnt2* (ref. 34) and *Nt5e*[35] in cluster 3, demonstrating broad differences in phenotypes of peritoneal macrophages that may be correlated with more immature macrophages (cluster 2) versus mature RTMs (cluster 3). Cluster 4 might represent a population that is transitioning from immature macrophages (cluster 2) to RTMs (cluster 3), as evidenced by its

*Timd4*, *Marco* and *Tgfb2* expression (Fig. 3a and Supplementary Fig. 11). Notably, expression of *Slc7a2*, encoding SLC7A2, which transports ornithine and polyamines into cells, was augmented in clusters 2, 3 and 4, perhaps indicating a role of polyamine uptake in macrophage differentiation (Supplementary Fig. 11). *Wnt2*, *Tgfb2* and *Cd63* were among the most significantly downregulated genes in DHPS-deficient cells in clusters 2 and 4 and 3, respectively (Supplementary Fig. 12), perhaps indicating that expression of these genes, which have roles in cell adhesion, trafficking and differentiation[33,36,37], is critical for RTM development. Pathway analysis of DEGs between control and DHPS-deficient macrophages indicated that cluster 2 exhibited decreases in chromatin remodelling, translation and proliferation (Fig. 3d and Supplementary Table 1), along with increased expression of inflammatory genes (Fig. 3e and Supplementary Table 2), findings that were to some extent echoed in DHPS-deficient cells in clusters 1 and 3 (Supplementary Fig. 13).

To confirm our findings in another tissue, we performed scRNA-seq on CD45[+]YFP[+] (LysM–Cre[+]) cells sorted from digested lungs of *Dphs*[+/+] LysM–Cre Rosa26[eYFP] (control) and *Dphs*-ΔM Rosa26[eYFP] mice (Supplementary Fig. 14). We focused only on macrophages, selecting cells on the basis of a range of markers relevant to this tissue (*Adgre1*, *Csf1r*, *H2-Ab1*, *Cd68*, *Lyz2*, *Itgam*, *Mertk*, *Itgax*, *Siglecf*). Our analysis revealed significant differences between DHPS-deficient and control macrophages (Extended Data Fig. 7 and Supplementary Fig. 15), with loss of cluster 0 and enrichment of other clusters (including cluster 1) in DHPS-deficient macrophages. When examining genes expressed by alveolar macrophages (*Adgre1*, *Itgax* and *Siglecf*), we found that cluster 0 was exclusively *Siglecf* positive and therefore represented mature alveolar macrophages, and this cluster was specifically absent from DHPS-deficient macrophages (Extended Data Fig. 7). These results mirror our flow cytometry data, which showed that *Dphs*-ΔM Rosa26[eYFP] mice did not lack F4/80+ (*Adgre1*) CD11c[+] (*Itgax*) macrophages but rather lacked mature Siglec-F[+] RTMs (Fig. 1d,f).

On the basis of our scRNA-seq from peritoneal macrophages (Fig. 3a–e), we speculated that the enriched intermediate clusters in DHPS-deficient macrophages (for instance, cluster 2), which were not *Timd4*[+], could represent macrophages blocked in RTM differentiation. In the lung scRNA-seq data (Extended Data Fig. 7), we identified two clusters, 1 and 10, that were almost exclusively found in *Dphs*-ΔM Rosa26[eYFP] mice. We questioned whether these clusters represented immature lung macrophages blocked in their differentiation towards mature RTMs, analogous to our observations in the peritoneum. To test this, we identified the top upregulated DEGs of cluster 2 (Fig. 3a,b), the main intermediate cluster, from our peritoneal macrophage dataset and searched for the same signature in the lung macrophage scRNA-seq dataset (Extended Data Fig. 8). This signature mapped to clusters 1 and 10 in the lung, suggesting that the cells we proposed to be blocked in differentiation in one tissue could be observed in a second tissue.

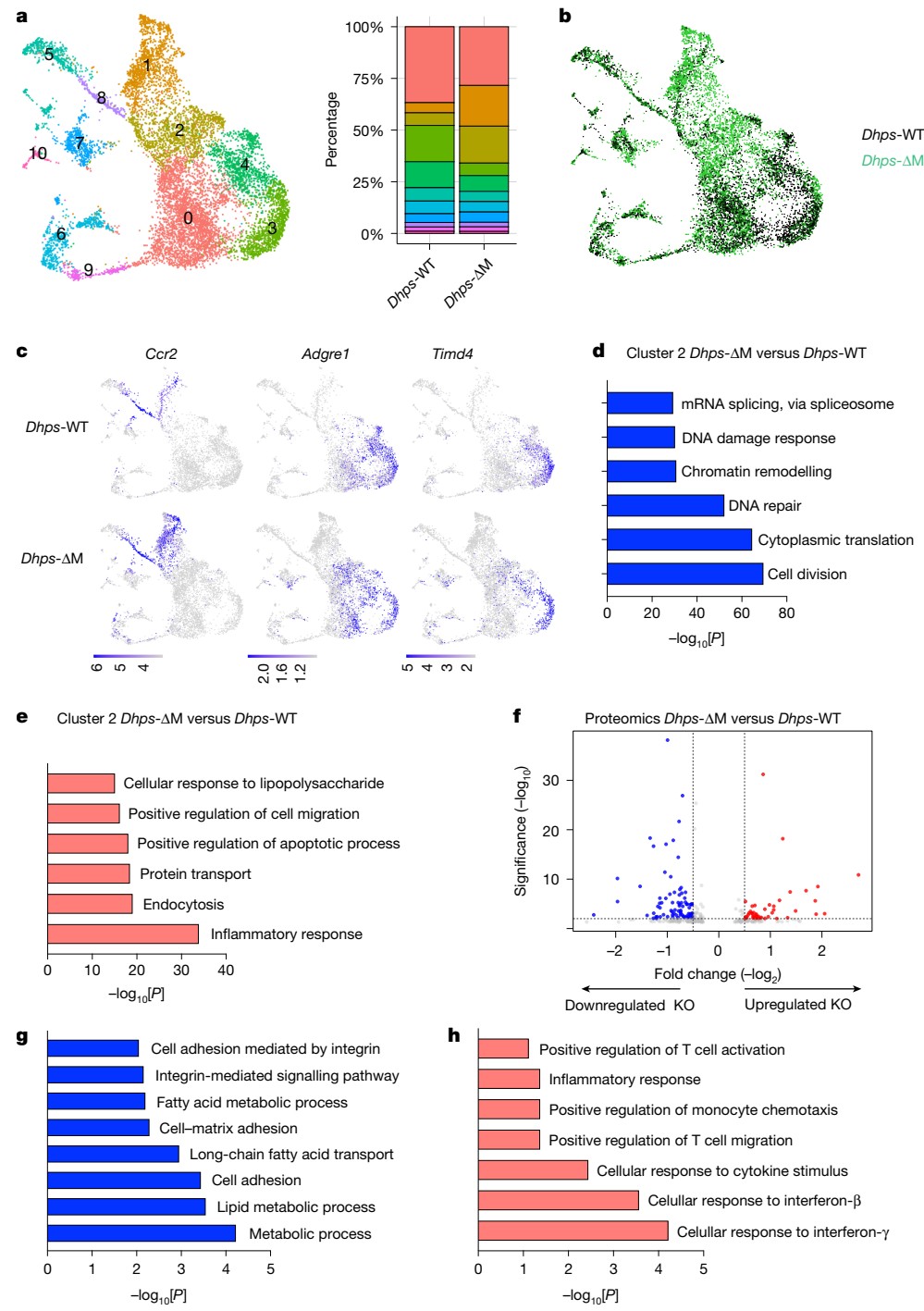

**Fig. 3 | Single-cell transcriptional analysis indicates a block in monocyte-to-RTM maturation in the absence of DHPS. a**, scRNA-seq clustering analysis of monocytes and macrophages from the peritoneal cavity of *Dhps*-WT and *Dhps*-ΔM mice. The proportions of each cluster within conditions are represented as percentages. **b**, Overlapping clustering distribution between *Dhps*-WT and *Dhps*-ΔM mice. **c**, *Ccr2*, *Adgre1* (F4/80) and *Timd4* expression in *Dhps*-WT and *Dhps*-ΔM mice. **d,e**, DAVID pathway enrichment analysis for downregulated (**d**) and upregulated (**e**) genes in cluster 2 (*Adgre1* (F4/80⁺; *Timd4*⁻) *Dhps*-ΔM versus *Dhps*-WT. **f**, Proteomics analysis: volcano plot of differentially expressed proteins in F4/80⁺CD11b⁺ sorted peritoneal macrophages from *Dhps*-WT and *Dhps*-ΔM mice. **g,h**, Selected downregulated (**g**) and upregulated (**h**) pathways from DAVID pathway enrichment analysis. scRNA-seq and proteomics data represent one experiment with three biological replicates per condition.

This indicated that these cells represent a transitional state from immature macrophages to RTMs that is independent of tissue. When DHPS was absent, whether in the peritoneal cavity or the lung, these cells failed to acquire the RTM signature imposed by the tissue and remained as immature macrophages. These results support a requirement for DHPS for macrophages to differentiate into mature RTMs across tissues.

## Defects in cell adhesion and signalling

We next assessed global protein expression in F4/80⁺ peritoneal macrophages from *Dhps*-WT and *Dhps*-ΔM mice (Fig. 3f). Pathway enrichment analysis revealed significant decreases in metabolism, cell adhesion and integrin-mediated signalling pathways, along with increases in immune activation and inflammation (Fig. 3g,h and

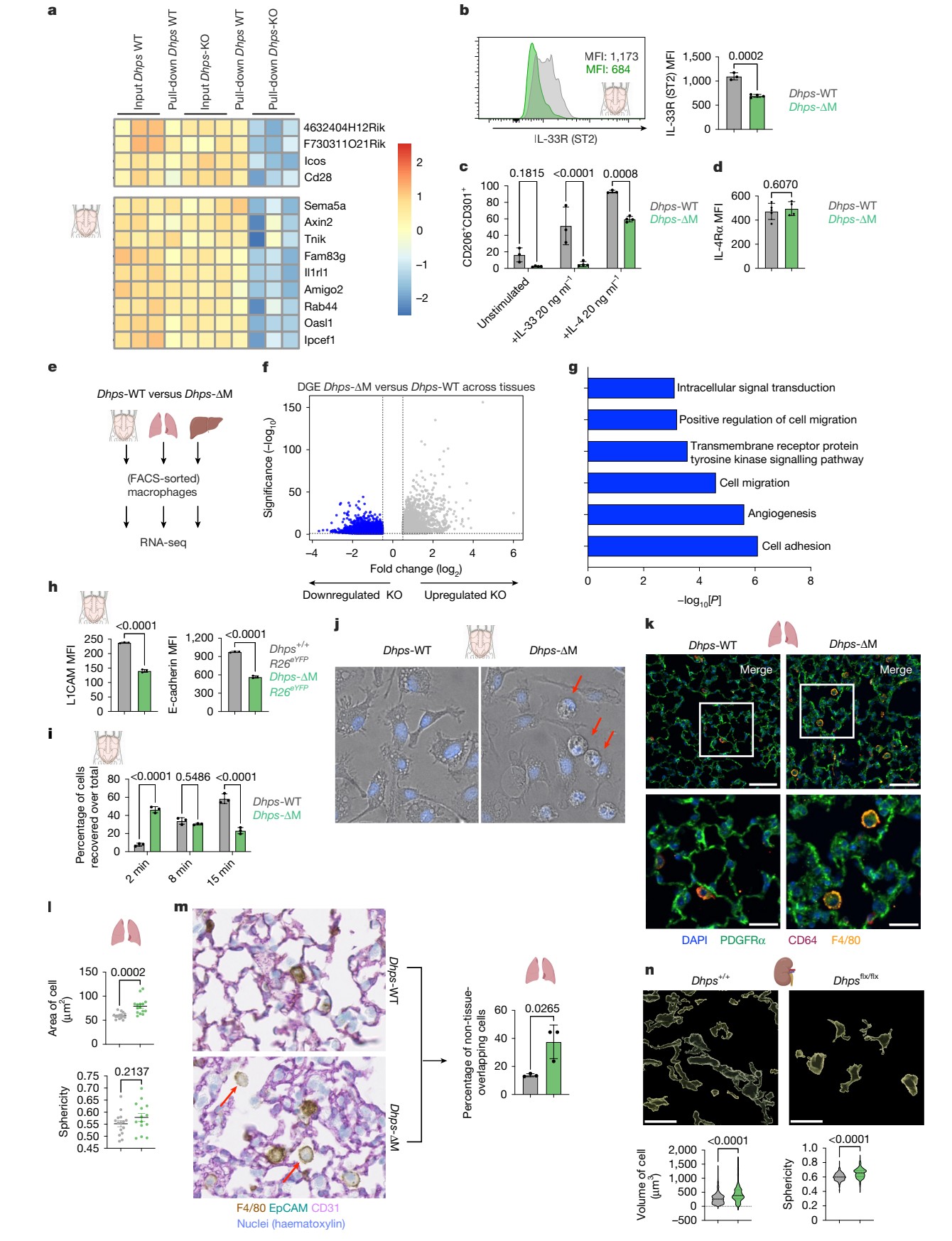

**Fig. 4 | See next page for caption.**

**Fig. 4 | Cell adhesion and signalling pathways critical for macrophage tissue-residency are DHPS-dependent. a**, Heat map of stable transcripts reduced on ribosomes in peritoneal macrophages (pMacs) (*Dhps*-ΔM). **b–d**, pMacs analysed for IL33R (ST2) (**b**); cultured with IL-33 (20 ng ml$^{-1}$) for 72 h or IL-4 (20 ng ml$^{-1}$) for 24 h and analysed (**c**); or analysed for IL-4R (**d**). **e**, Bulk RNA-seq scheme. **f**, Volcano plot of shared DEGs. **g**, DAVID pathway analysis of downregulated genes in *Dhps*-deficient macrophages. **h**, L1CAM and E-cadherin, flow cytometry; pMacs from *Dhps*$^{+/+}$ Rosa26$^{eYFP}$ and *Dhps*-ΔM Rosa26$^{eYFP}$ mice. **i**, Percentage cell recovery by time after EDTA; pMacs were cultured for 24 h. **j**,**k**, Confocal images of cultured pMacs with cell structure by brightfield microscopy (**j**) (arrows indicate rounded cells) and lungs stained for macrophages and stromal cells (F4/80, CD64, and PDGFRa, respectively) from *Dhps*-WT and *Dhps*-ΔM mice (**k**). Nuclei (DAPI); box indicates region of interest. **l**, Area and sphericity of F4/80$^+$CD64$^+$ macrophages from **k**. Each data point is the average of five random fields quantified per biological replicate.

**m**, Lungs from *Dhps*-WT and *Dhps*-ΔM mice stained for macrophages (F4/80, brown), vasculature (CD31, purple), epithelial cells (EpCAM, teal) and nuclei (haematoxylin, indigo). Frequency of F4/80$^+$ cells not positioned on tissue (non-overlapping) was quantified from five random fields in three biological replicates per genotype. **n**, Confocal images of YFP$^+$ kidney macrophages from *Dhps*$^{+/+}$ (control) and *Dhps*$^{flx/flx}$ CX3CR1–ERT2cre-Rosa26$^{eYFP}$ mice 5 weeks post-tamoxifen. Cell volume and sphericity were quantified by Imaris (10.0.1, Bitplane); five random fields in five biological replicates per genotype. All data represent two or more experiments, except in **a** and **e–g**, in which data represent one experiment with three biological replicates. Data are mean ± s.d. For **b** and **c**, $n = 3$, *Dhps*-WT; and $n = 4$, *Dhps*-ΔM. For **d**, $n = 5$, *Dhps*-WT; and $n = 4$, *Dhps*-ΔM. For **h**, **i**, **l** and **m**, $n = 3$. For **n**, $n = 5$. $n$ represents biological replicates. Statistical analyses; two-tailed t-tests and two-way analyses of variance; *P* values are shown. Scale bars, 50 μm (**k**, top), 20 μm (**k**, bottom); 20 μm (**n**). Illustrations in **a**, **b**, **e** and **h–n** were created using BioRender (https://biorender.com).

Supplementary Tables 3 and 4). Many studies have outlined the importance of metabolic remodelling in immune cells[38], including RTMs[39]. Likewise, cell adhesion and signalling are critical facets of RTM biology. A lack of robust expression of components in any of these pathways could lead to a block in RTM differentiation. As hypusinated eIF5A is important for translation, we investigated active translation in WT and DHPS-deficient macrophages by sequencing ribosome-engaged transcripts.

We crossed *Dhps*$^{+/+}$ LysM–Cre mice and *Dhps*-ΔM (*Dhps*$^{flx/flx}$ LysM–Cre) mice with RiboTag mice to create a model in which ribosomes become HA-tagged upon LysM–Cre expression, enabling us to efficiently immunoprecipitate ribosomes from macrophages ex vivo and sequence ribosome-associated transcripts (Supplementary Fig. 16a). We used peritoneal macrophages as they are more abundant and their isolation does not require tissue digestion. Before sequencing, we confirmed that of the HA-tagged cells in the peritoneal cavity, 98% of *Dhps* 'Ribo'-WT cells) and 93% of *Dhps* 'Ribo'-ΔM cells were F4/80$^+$CD11b$^+$ macrophages (Supplementary Fig. 16b). We then sequenced transcripts in all peritoneal exudate cells (input: approximately 60% F4/80$^+$CD11b$^+$HA-TAG$^+$ cells in *Dhps* 'Ribo'-WT mice and approximately 50% F4/80$^+$CD11b$^+$HA-TAG$^+$ cells in *Dhps* 'Ribo'-ΔM mice; Supplementary Fig. 16b), and all transcripts bound to ribosomes (immunoprecipitation of HA-tagged ribosomes) for each genotype. Although this approach meant that the input included some non-macrophage cells, the pull-down was exclusively HA-tagged LysM-expressing cells and reduced manipulation of the samples.

Hypusinated eIF5A promotes translation of sequences that stall the ribosome, and ribosome stalling can lead to transcript degradation[10,11]. Therefore, we analysed all transcripts that were stably expressed in the 'inputs' for both genotypes and then filtered for transcripts that were differentially enriched on ribosomes between control and DHPS-deficient macrophages (Supplementary Fig. 16c). We aimed to find genes that were significantly decreased on ribosomes in DHPS-deficient macrophages compared with controls but not owing to extraordinary differences in total transcript abundance. This analysis identified 13 genes that were significantly reduced on ribosomes in DHPS-deficient macrophages (Fig. 4a), most of which were involved in cell adhesion or interaction, signalling and apoptosis; these genes included *Icos*[40], *Cd28* (ref. 41), *Axin2* (ref. 42), *Tnik*[43,44], *Amigo2* (ref. 45), *Fam83g*[46], *Il1rl1* (refs. 47–49), *Rab44* (ref. 50) and *Oasl1* (ref. 51). These data suggest that the proteins encoded by these genes might be involved in RTM differentiation, and that their expression is decreased in DHPS-deficient macrophages.

*Il1rl1* codes for the IL-33 receptor (ST2) and is involved in differentiation of bone-marrow-derived monocytes and macrophages to RTMs[47,52] and in the self-renewal and tissue repair and maintenance functions of RTMs[48,53]. Consistent with our ribosome sequencing data, ST2 was decreased on peritoneal macrophages from *Dhps*-ΔM mice (Fig. 4b), and, functionally, the ability of these macrophages to alternatively activate in response to IL-33 in vitro was impaired[47,48] (Fig. 4c). By contrast, DHPS-deficient macrophages expressed IL-4R and partially responded to IL-4 in vitro by inducing alternative activation markers, although not to control cell levels (Fig. 4c,d), consistent with their defective response to IL-4c in vivo (Extended Data Fig. 5h–k) and our previous findings that hypusinated eIF5A directs macrophage alternative activation[14]. These data suggest that ST2 is among a subset of ribosome-engaged transcripts that depend on hypusinated eIF5A for efficient translation and that its reduced protein expression might contribute to the monocyte-to-RTM transition defect in the peritoneal cavity in *Dhps*-ΔM mice. However, although decreased ST2 might contribute to this defect in DHPS-deficient peritoneal macrophages, it may not be important for the RTM transition in other tissues. Also, multiple genes, rather than any one gene, probably contribute to RTM differentiation.

As *Dhps*-ΔM mice manifested a defect in all RTMs, we sought to understand which pathways might be critical for the RTM transition across tissues. We performed bulk RNA-seq on F4/80$^+$ macrophages (Supplementary Fig. 17a–c) isolated from the lung, liver and peritoneal cavity of *Dhps*-WT and *Dhps*-ΔM mice (Fig. 4e,f). Pathway analysis of DEGs that were common among all DHPS-deficient macrophages from each tissue identified genes enriched in pathways of cell adhesion, signalling and migration as the most significantly downregulated (Fig. 4g and Supplementary Tables 5 and 6), with increased expression of those enriched in inflammatory response and apoptotic pathways (Supplementary Fig. 17d and Supplementary Tables 7 and 8). Notably, our sequencing of ribosome-engaged transcripts identified several genes involved in cell adhesion (Fig. 4a). For example, *Tnik* augments expression of target genes in the Wnt/β-catenin pathway, which controls cell adhesion[43,44]; as such, β-catenin-deficient macrophages lack cell adherence capacity[54]. The genes encoding cell adhesion molecules L1CAM and E-cadherin (*l1cam* and *Cdh1*, respectively) were downregulated in the analysis of DHPS-deficient macrophages isolated from three distinct tissues (Supplementary Table 6); this was consistent with the finding of decreased protein expression (Fig. 4h) and with earlier work showing that E-cadherin expression is polyamine-dependent in alternatively activated macrophages[55]. To investigate cell adhesion functionally, we cultured peritoneal macrophages for 24 h in the presence of CSF-1 to allow adherence. Cell adhesion and integrin binding are Ca$^{2+}$-dependent. We quantified the sensitivity of cultured cells to detach with EDTA over time. DHPS-deficient macrophages were significantly less adherent, with approximately 50% of the cells detaching after 2 min, compared with fewer than 10% of control cells, which took 15 min for 50% detachment (Fig. 4i). We also observed that DHPS-deficient peritoneal macrophages had rounded morphology compared with control cells (Fig. 4j). Taken together, these results indicate that cell adhesion and signalling are critical for RTM development across tissues, and that this program is controlled by DHPS.

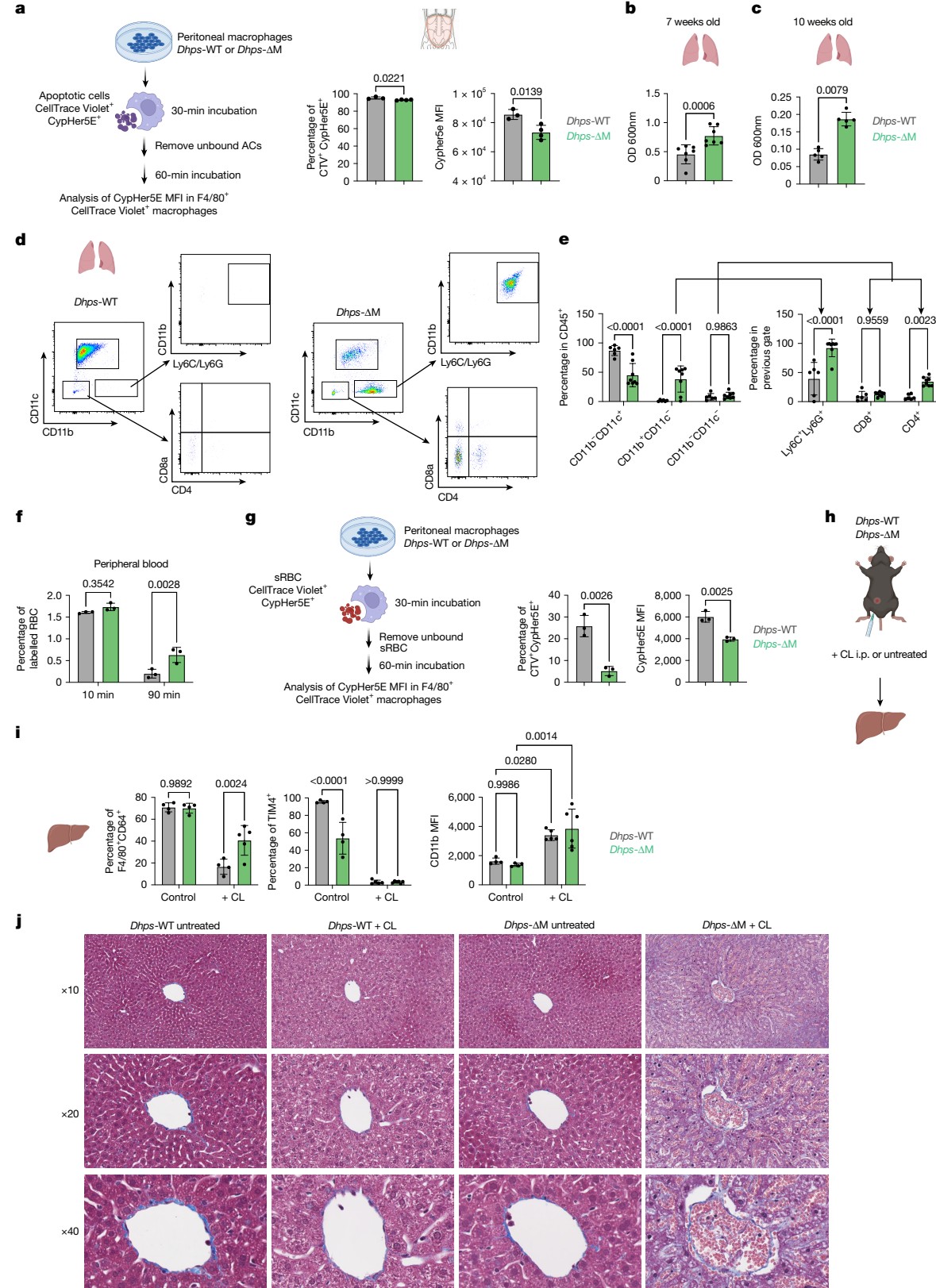

**Fig. 5** | See next page for caption.

## Altered morphology and tissue interactions

Our proteomics and transcriptomics analyses suggested that DHPS was critical for enacting the program of RTMs by influencing the expression of signalling molecules, of which ST2 is an example, and cell adhesion

proteins. If this were true, we reasoned that defects in cell adhesion and signalling would influence how DHPS-deficient macrophages interacted in tissues. We visualized macrophages in the lungs of *Dhps*-WT and *Dhps*-ΔM mice and found that DHPS-deficient macrophages exhibited larger size with a different pattern of localization in the

**Fig. 5 | DHPS-deficient macrophages are defective in critical homeostatic functions of RTMs. a**, Peritoneal macrophages from *Dhps*-WT and Dhps-ΔM mice were cultured with labelled apoptotic cells and measured by flow cytometry. **b,c**, Lung proteinosis measured by optical density at 600 nm of BAL supernatants from 7-week old (**b**) or 10-week-old (**c**) *Dhps*-WT and *Dhps*-ΔM mice. **d**, Gating strategy for CD45⁺ cells from adult *Dhps*-WT and *Dhps*-ΔM in BAL fluid. **e**, Immunophenotyping and quantification of CD45⁺ subpopulations in BAL from *Dhps*-WT and *Dhps*-ΔM mice, by flow cytometry. Frequencies of monocytes and neutrophils (Ly6C⁺Ly6G⁺) were gated from CD11b⁺CD11c⁻ cells, and CD4⁺ and CD8⁺ T cells from CD11b⁻CD11c⁻ cells within CD45⁺ cells. **f**, Labelled sRBCs were intravenously injected into *Dhps*-WT and *Dhps*-ΔM mice. After 10 min and 90 min, the frequency of CTV⁺Cypher5e⁺ cells in peripheral blood was measured by flow cytometry. **g**, Labelled sRBCs were cultured with peritoneal macrophages for 30 min, and efferocytosis was measured by flow cytometry after 60 min. **h**, Scheme of i.p. injection of CL and liver harvesting in *Dhps*-WT and *Dhps*-ΔM mice. **i**, Frequency of liver macrophages (F4/80⁺CD64⁺), TIM-4⁺ Kupffer cells, and expression of CD11b in F4/80⁺CD64⁺ cells in CL-treated or control (untreated) *Dhps*-WT and *Dhps*-ΔM after CL. **j**, Stained liver sections from CL-treated or control (untreated) *Dhps*-WT and *Dhps*-ΔM mice 6 days after CL. Representative plots and graphs summarize results of two or more experiments with at least three biological replicates. Data are mean ± s.d. For **a**, *n* = 3, *Dhps*-WT; and *n* = 4, *Dhps*-ΔM. For **b**, *n* = 7; for **c**, *n* = 5; and for **e**, *n* = 6, *Dhps*-WT; and *n* = 8, *Dhps*-ΔM. For **f** and **g**, *n* = 3. For **i**, *n* = 4, *Dhps*-WT (untreated and CL); and *n* = 5 (untreated) and *n* = 6 (CL), *Dhps*-ΔM. All *n* indicate biological replicates. Statistical analyses: two-tailed *t*-tests and two-way analyses of variance; *P* values are shown. Illustrations in **a**–**d** and **g**–**i** were created using BioRender (https://biorender.com).

alveoli, that is, less stromal cell overlap (Fig. 4k–m and Supplementary Fig. 18a,b). We also assessed the morphology of kidney macrophages after acute DHPS deletion in *Dhps*^flx/flx^ CX3CR1–ERT2cre-Rosa26^eYFP^ mice. YFP⁺ DHPS-deficient kidney macrophages were larger and more spherical (Fig. 4n and Supplementary Fig. 18c). In the same mice, we assessed microglia and observed changes in cell size, morphology and branching after DHPS deletion (Supplementary Fig. 18d,e). These data support the concept that cell shape and adhesion differences can lead to altered tissue interactions, which could negatively affect macrophage survival and transition of DHPS-deficient cells to RTMs.

## Defects in homeostatic functions

The defect in RTMs in the absence of DHPS led us to question whether the F4/80⁺ macrophages present in young *Dhps*-ΔM mice possessed RTM functions. A main function of RTMs is to clear cellular debris[3]. DHPS-deficient peritoneal macrophages cocultured with apoptotic cells labelled with CTV and Cypher5E (which becomes fluorescent inside early phagolysosomes) exhibited reduced dead cell uptake and Cypher5E fluorescence, suggesting a defect in their ability to internalize and break down dead cells (corpses) (Fig. 5a). In vivo, alveolar macrophages clear surfactants; without this critical function, alveolar proteinosis develops[56]. Assessment of bronchoalveolar lavage (BAL) fluid from *Dhps*-ΔM mice demonstrated alveolar proteinosis that worsened over time (Fig. 5b,c) and increased presence of CD45⁺ cells, including CD4⁺ T cells and monocytes and neutrophils (CD11b⁺Ly6C⁺Ly6G⁺) (Fig. 5d,e). These results indicated developing inflammation in the presence of DHPS-deficient macrophages, probably driven by perturbed tissue homeostasis, which, consistent with our transcriptomics and proteomics data, promoted a proinflammatory environment with recruitment of other immune cells to the tissue.

To test efferocytosis systemically, we measured the ability of control and *Dhps*-ΔM mice to remove dead and dying RBCs from the bloodstream[57]. We intravenously injected Cypher5e- and CTV-labelled stressed RBCs (sRBCs) into control and *Dhps*-ΔM mice. Although both genotypes had similar frequencies of labelled RBCs circulating after 10 min, *Dhps*-ΔM mice had significantly more sRBCs in the circulation after 90 min, indicating a decreased ability to clear these cells (Fig. 5f). To test whether this systemic defect in clearing sRBCs was correlated with a macrophage-specific defect in efferocytosis, we isolated peritoneal macrophages from control and *Dhps*-ΔM mice (too few liver RTMs in *Dhps*-ΔM mice made isolation difficult) and cultured them with labelled sRBCs. Both the frequency of Cypher5e⁺CTV⁺ cells and the mean fluorescence intensity (MFI) of Cypher5e were decreased in DHPS-deficient macrophages, indicating significantly less acquisition of efferocytic cargo (Fig. 5g). These data demonstrate that DHPS-deficient macrophages are defective in clearance of apoptotic cells and debris, an important homeostatic RTM function.

A key function of RTMs is to maintain organ homeostasis. We observed significant lung immune cell infiltration in *Dhps*-ΔM mice

(Fig. 5d,e), overtly indicating disrupted tissue homeostasis in the absence of myeloid DHPS expression and defective alveolar macrophages in these mice. To acutely investigate RTM function in maintaining tissue homeostasis in another tissue, and to demonstrate that myeloid expression of DHPS is required for this critical RTM function, we i.p. injected CL in *Dhps*-WT and *Dhps*-ΔM mice to broadly deplete myeloid cells in the peritoneal cavity and in other organs in that space (Fig. 5h). We focused on the liver as it has been shown that upon RTM depletion, bone-marrow-derived monocytes enter the tissue and differentiate into monocyte-derived macrophages and, over time, into RTMs, replenishing the liver RTM niche[20]. In both *Dhps*-WT and *Dhps*-ΔM mice, 5 days after CL, F4/80⁺CD64⁺TIM-4⁺ macrophages were depleted, and monocyte-derived macrophages were detected in the tissue by increased expression of CD11b within the F4/80⁺CD64⁺ gate (Fig. 5i). In the control group, liver sections stained with haematoxylin and eosin (H&E) and Masson trichrome (MAS) appeared identical before and 6 days after CL, whereas livers of *Dhps*-ΔM mice exhibited congestion (Fig. 5j and Supplementary Fig. 19a), abnormal sinusoids, endothelial lining detached from liver cell plates, central veins lifted from the parenchyma (Supplementary Fig. 19b,c) and extensive necrosis (Supplementary Fig. 19d). These data show that myeloid cell DHPS expression is critical for maintenance and/or restoration of liver tissue homeostasis after CL.

## Discussion

Our data support a model in which monocyte interaction within a tissue is a primary and critical driver of RTM differentiation[58]. We show that the polyamine–hypusine axis is an important cell-intrinsic regulatory node directing this process, at least in part by supporting expression of cell adhesion and signalling molecules required for the monocyte–macrophage tissue relationship. Through studying macrophage-specific models of DHPS deficiency, we observed that monocytes are unable to fully differentiate into RTMs in tissues and instead remain as immature monocyte-derived macrophages that would normally develop only during damage[3]. Accordingly, DHPS-deficient macrophages lack homeostatic functions such as clearing of dead cells and cell debris that are normally assumed by RTMs and are thus unable to maintain tissue homeostasis.

Hypusine modification of eIF5A has been shown to be important for translation of sequences such as di- or polyproline stretches that can stall the ribosome and lead to transcript degradation[11]. We assessed protein expression, transcriptional output by bulk and scRNA-seq, as well as ribosome-engaged transcripts in macrophages to best resolve how DHPS, and thus hypusinated eIF5A, controls RTMs. Whereas TNIK is proline-rich and ST2 has diproline motifs, which potentially confer hypusine dependency, exactly which motifs in transcripts require hypusinated eIF5A for efficient translation is not completely clear. Further, which transcripts in a macrophage are dependent on hypusinated eIF5A and are responsible for driving this complex in vivo tissue behaviour are

challenging to resolve, as transcriptional programs between distinct populations of RTMs across tissues vary considerably. There are significant transcriptional differences between control and DHPS-deficient cells; thus, it is difficult to pinpoint how decreased gene expression affects translation of proteins, that is, a lower number of transcripts versus whether a specific transcript is directly eIF5A dependent. Moreover, it is possible that specific proteins such as transcription factors or chromatin-modifying factors are dependent on hypusinated eIF5A, such that the transcriptional program required for RTM differentiation, which includes cell adhesion and signalling proteins, is never enacted in DHPS-deficient cells. Although historical evidence suggests that eIF5A is the only hypusinated protein, a recent study indicated that spermidine reduces RIPK1 activation in a DHPS-dependent manner[59]. Future research will be needed to determine whether other proteins are modified by the polyamine–hypusine axis. Precise molecular mechanisms notwithstanding, our data demonstrate that DHPS controls a cellular program that affects cell adhesion and signalling critical for RTM differentiation across tissues.

Another question raised by our findings is that of how polyamine levels in a cell influence hypusine and affect translation in diverse biological contexts. Whether or not a cell uses spermidine to hypusinate eIF5A could be influenced by the cell's engagement of polyamine biosynthesis and the requirement (and hence the pull) of polyamines towards other processes, as well as whether the cell is able to acquire polyamines, which are provided by the diet, microbiota and even dying cells[60]. Investigation of extracellular polyamine levels in tissues and intracellular polyamine levels and hypusine synthesis in health and disease may shed light on these questions. Our results show that DHPS deletion permits initial macrophage development but prevents these cells from taking up residence in tissues. Our data are consistent with a model of RTM development whereby CSF1R signalling drives the initial macrophage program, and then the polyamine–hypusine pathway determines subsequent tissue occupancy and acquisition of tissue-specific programs. Owing to this, our findings have important implications for how systemic tissue homeostasis is maintained, how macrophages may be targeted therapeutically and how these cells are deleteriously remodelled with age.

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

# Methods

## Mice

WT C57BL/6 and CD45.1 SJL mice, mice expressing Cre recombinase (LysM–Cre) under the control of the lysozyme (*LysM*) promoter, *CX3CR1–ERT2* inducible Cre, *Rosa26-eYFP* (R26$^{YFP}$) mice and RiboTag mice[61] harbouring a modified allele of *Rpl22* (*Rpl22-HA*) induced by the action of Cre recombinase were all purchased from Jackson Laboratories. *Dhps* floxed mice were a gift from S. Balabanov, Zurich. Mice were bred and maintained under specific-pathogen-free conditions, and experiments were performed under protocols approved by the Regierungspräsidium Freiburg, the Animal Welfare Committee of the Max Planck Institute of Immunobiology and Epigenetics, or the Institutional Animal Care and Use Committee of Johns Hopkins University School of Medicine, in accordance with the Guide for the Care and Use of Animals. Mice used for all experiments were littermates and matched for age and sex; both male and female mice were used. Mice of all strains were typically 6–12 weeks of age unless otherwise specified. Experiments were not performed blinded. Parabiosis experiments were performed in the laboratory of F.G., with Institutional Animal Care and Use Committee approval from the Biological Resource Center (A*STAR) in Singapore, as follows: the animals were monitored daily for evidence of fighting and were exposed to a gel food diet supplement at least two times, 2–6 weeks before performance of parabiotic attachment. Animals were randomly assigned a pair member, with the appropriate transgene or CD45 congenic allotype, within their cohoused cohort. Pairs established for parabiotic attachment were 'cage paired', meaning they were housed only with their future parabiotic pair member, 2 weeks before the parabiosis procedure, and the mice were observed again for aggressive behaviour. Aggressive and injured mice were removed from the study. With this husbandry protocol, aggression was highly uncommon.

## Tissue processing for RTM isolation

Lungs, liver, kidney, brain and heart were harvested after perfusion with ice-cold phosphate-buffered saline (PBS) through the heart; organs were placed in 5-ml tubes containing 2 ml digestion solution with 2 mg ml$^{-1}$ collagenase IV (Thermo Fisher) and 0.1 mg ml$^{-1}$ DNAse I (Sigma) and cut into small pieces with scissors. The solution was incubated in a shaker at 37 °C for 1 h. Tissues were further homogenized using a 20 G × 1″ needle and a 5-ml syringe, and homogenates were filtered through a 100-µm cell strainer. RBCs from lungs, liver, kidney and heart were lysed with ACK for 2 min, washed with PBS 1% fetal calf serum (FCS), 2 mM EDTA (FACS buffer) and kept on ice until flow cytometry staining. For splenic macrophages, no digestion solution was used. Cells were isolated by mechanical disaggregation of spleen on a 70-µm strainer, RBCs were lysed and cells were resuspended in FACS buffer. Microglia were isolated postdigestion using a 35–70% Percoll gradient with centrifugation for 20 min at 500*g* and room temperature without brakes. The leucocyte layer was transferred to tubes containing FACS buffer and kept on ice until flow cytometry staining. When indicated, alveolar macrophages were isolated by BAL and centrifuged to isolate lavage fluid for optical density measurement, and the pelleted cells were resuspended in FACS buffer for flow cytometry staining. Absolute numbers of macrophages were calculated using 123count eBeads Counting Beads (Thermo Fisher) following the manufacturer's instructions. For solid organs, the weight of each tissue in milligrams was recorded after harvesting, and the number of macrophages per sample was normalized by this value to obtain the total cells per milligram tissue.

## Peritoneal macrophages in vitro culture

Peritoneal macrophages were obtained by peritoneal lavage and either cultured in complete medium (RPMI 1640 medium supplemented with 10% FCS, 2 mM L-glutamine and 100 U ml$^{-1}$ penicillin/streptomycin) with 20 ng ml$^{-1}$ macrophage colony stimulating factor (PeproTech) on a six-well-plate, or transferred to a 96-well-plate and used directly for flow cytometry staining. For in vitro activation, peritoneal macrophages were cultured with 20 ng ml$^{-1}$ IL-4 (PeproTech) overnight to generate alternatively activated macrophages. For IL-33 (R&D Systems) in vitro treatment, 20 ng ml$^{-1}$ was used in all experiments unless otherwise specified, with culture for 72 h.

## Parabiosis

Parabiotic mice were generated as reported[62] from age- and weight-matched CD45.2$^+$ (*Dhps$^{flx/flx}$* LysM–Cre negative (*Dhps*-WT) or *Dhps$^{flx/flx}$* LysM–Cre positive (*Dhps*-ΔM) and CD45.1$^+$ (C57BL/6) mice (WT) between 6 and 8 weeks old.

## Bone marrow chimera

CD45.1$^+$ (C57BL/6) mice (WT) were irradiated (1,200 cGy, 2 split doses, 3 h apart) in a Cesium Mark 1 irradiator (JL Shepperd & Associates) and then infused with a mix of 5 × 10$^6$ bone marrow cells from *Dhps$^{flx/flx}$* LysM–Cre negative (*Dhps*-WT) or *Dhps$^{flx/flx}$* LysM–Cre positive (*Dhps*-ΔM) CD45.2$^+$ cells mixed with 5 × 10$^6$ bone marrow cells from CD45.1$^+$ mice (1:1 ratio). Mice were harvested, and the frequency of CD45.1$^+$/CD45.2$^+$ RTMs were analysed at 12 weeks after bone marrow transfer.

## In vivo proliferation of peritoneal macrophages by IL-4c administration

For long-acting IL-4 treatment, a mixture of 5 mg recombinant mouse IL-4 (Peprotech) and 25 mg anti-IL-4 mAb (clone 11B11; BioXcell) was incubated for 5 min on ice to form an IL-4:anti-IL-4 complex (IL-4c). IL-4c enables sustained and slow IL-4 release. Mice were then injected i.p. with 100ul of IL-4c (containing 5 µg IL-4 and 25 µg anti-IL-4), or PBS vehicle control. The peritoneal lavage was analysed at day 4 post-injection. To measure proliferation, EdU (0.5 mg) was injected i.p. at day 0 and day 2 in IL-4c treated mice and analysed at day 4. To assess proliferation of adoptively transferred peritoneal macrophages, ex vivo isolated peritoneal macrophages from *Dhps*-WT or *Dhps*-ΔM CD45.2 mice were stained with CTV (Life Technologies) following the manufacturer's instructions and assessed for purity by flow cytometry; then, 0.5 × 10$^6$ to 1.0 × 10$^6$ cells per mouse were transferred by injection i.p. to CD45.1 WT receptor mice. Immediately after transfer, IL-4c or PBS was injected i.p., and macrophage proliferation was measured at day 4 by CTV dilution with flow cytometry.

## RTM in vivo depletion by CL treatment

Mice were injected with 200 µl CL (Liposoma BV) i.p. or PBS to deplete RTMs in the peritoneal cavity and liver. Peritoneal lavage and tissues were collected at the indicated time points for flow cytometry or tissue histology.

For depletion of alveolar macrophages, adult male or female mice were treated with intratracheal delivery of CL at days −2 and 0 after the following procedure on each day: mice were anaesthetized with a mixture of ketamine (75 mg kg$^{-1}$) and xylazine (10 mg kg$^{-1}$) injected i.p. Their necks were cleaned with alternating scrubs of 70% ethanol and betadine. A vertical incision was then made in the neck, and the trachea was exposed. Orotracheal intubation was performed using a 20 g intravenous catheter under direct visualization of the trachea. The mice were briefly ventilated using a MiniVent Model 845 (Harvard Apparatus) with a respiratory rate of 150 breaths per min and tidal volume of 200 µl. The mice were then disconnected from the ventilator, and 70 µl of liposomes were delivered each day. The mice were again briefly ventilated and then extubated and allowed to breathe spontaneously. The neck incision was closed with glue. The mice were kept on a warming blanket until they awakened from anaesthesia and then returned to their cages. BAL was collected at different time points posteuthanasia for flow cytometry analysis. Control mice received PBS instead of CL.

## In vivo Cre induction by tamoxifen administration

Tamoxifen (Sigma) was dissolved in corn oil to a final concentration of 10 mg ml$^{-1}$ and stored at −20 °C. Adult Cx3cr1CreERT2-Rosa26$^{YFP}$ mice were given 200 µl (2 mg) tamoxifen solution once per day for 4 consecutive days, injected i.p. Tissues were harvested at the indicated time points after tamoxifen injection.

## In vitro efferocytosis of apoptotic cells by ex vivo peritoneal macrophages

Peritoneal macrophages were plated on 24-well uncoated tissue culture plates at a density of $0.1 \times 10^6$ cells per well overnight before addition of apoptotic cells. Apoptotic cells were prepared from total lymphocytes isolated from lymph nodes and treated with staurosporin at 0.5 µM overnight at a density of $5 \times 10^6$ ml$^{-1}$ to induce apoptosis. Apoptotic cells were then washed with culture medium and stained with CypHer5E NHS Ester (Cytiva) and CTV (Thermo Fisher), each at 1 µM in PBS. Labelled apoptotic cells were washed three times in culture medium (containing 10% FBS), pelleted and resuspended for counting by trypan blue exclusion.

Apoptosis was confirmed by Apotracker and live/dead viability staining with flow cytometry. Apoptotic cells were added to macrophages in a 1:5 ratio (macrophages to apoptotic cells). To increase the contact between macrophages and apoptotic cells, 30 s centrifugation at 300g was performed, and cells were left at 37 °C in an incubator for 30 min. After this time, apoptotic cells were washed three times with culture medium, and macrophages were left for another 60 min at 37 °C to allow progression of efferocytosis. To analyse the rate of efferocytosis, macrophages were collected and stained for flow cytometry. Cargo-positive cells were gated as F4/80$^+$CTV$^+$, and levels of efferocytosis were analysed on the basis of MFI of cypher5e in the AF647/APC channel in cargo-positive cells.

## Mouse sRBC preparation for in vitro and in vivo transfer

Fresh RBCs were prepared by isolation from mouse peripheral blood and depletion of leukocytes by two steps. First, a 35–70% Percoll gradient was used to pellet RBCs and deplete leukocytes (positioned in the interphase). The second step involved incubating the RBC pellet with mouse CD45 MojoSort beads (BioLegend) following the manufacturer's protocol to deplete any remaining contaminating CD45$^+$ cells. After isolation, RBCs were stressed by heat shock at 48 °C for 20 min under constant agitation. The generated sRBCs were spun down for 10 min at 400g in PBS to remove free haemoglobin and colabelled with CTV and Cypher5e following the manufacturer's instructions. Before use, sRBCs were washed more than three times with cold PBS to remove free CTV and Cypher5e probes.

## In vitro efferocytosis of sRBCs by ex vivo peritoneal macrophages

Peritoneal macrophages were plated on 24-well uncoated tissue culture plates at a density of $0.1 \times 10^6$ cells per well overnight before addition of sRBC. To increase the contact between macrophages and sRBCs, a 30-s centrifugation at 300g was performed, and cells were left at 37 °C in an incubator for 30 min. After this time, sRBCs were washed three times with culture media, and macrophages were left for another 60 min at 37 °C to allow progression of efferocytosis. For analysis of the rate of efferocytosis, macrophages were collected and stained for flow cytometry. The percentage of macrophages in active efferocytosis was gated as CTV$^+$Cypher5$^+$ in the F4/80$^+$CD11b$^+$ gate, and the progression of efferocytosis was analysed by MFI of cypher5e in the AF647/APC channel within this gate.

## In vivo removal of transferred stressed sRBCs from peripheral blood

CTV and Cypher5e colabelled sRBCs were prepared as described above and injected into mice through the tail vein. The prepared sRBCs (isolated from one mouse) were injected into ten recipient mice. Each recipient mouse received a total volume of sRBCs resuspended in 200 µl of PBS. At 10 min (initial sRBC frequency in circulation) and 90 min (end time point) after injection, mice were euthanized, and whole blood was isolated. The frequency of remaining labelled sRBCs was measured by flow cytometry by gating in CD45$^-$CTV$^+$Cypher5e$^+$ cells in peripheral blood.

## BAL and ex vivo lung proteinosis analysis

A vein catheter (BD) was inserted into the trachea of mice, and the first wash was performed with 1 ml of prewarmed PBS followed by four extra washes to collect BAL cells. The first BAL wash with PBS was centrifuged for 5 min at 500g, and the supernatant was used for measurement of optical density at 600 nm using a spectral photometer. The cell pellet was merged with the rest of the BAL washes for analysis by flow cytometry.

## Flow cytometry protocol and cell sorting

For flow cytometry staining, cells were washed in cold PBS, and non-specific antibody binding to cells was blocked by incubation with an anti-CD16/32 antibody (clone 2G8; BD Biosciences) at 4 °C for 10 min in the presence of LIVE/DEAD viability dye (Thermo). Then, cells were stained with fluorophore-conjugated antibodies (Supplementary Table 9) at 4 °C for 25 min in PBS 1% FCS, 2 mM EDTA (FACS buffer). Cells were maintained at 4 °C and analysed on a BD Fortessa X20, Celesta or Symphony (BD Biosciences). For intracellular staining of active caspase-3 and Ki-67 (Supplementary Table 9), BD Cytofix/Cytoperm Fixation/Permeabilization Kit was used following manufacturer recommendations and the cells stained overnight and analysed on a BD Fortessa X20, Celesta or Symphony (BD Biosciences). Data were analysed in FlowJo (FlowJo LLC). For cell sorting non-specific antibody binding to cells was blocked by incubating cells with an anti-CD16/32 antibody (clone 2.4G2; BD Biosciences) at 4 °C for 10 min. The cells were then stained with fluorophore-conjugated antibodies (Supplementary Table 9) at 4 °C for 25 min. FACS was performed on a BD FACS Aria III (BD Biosciences) to achieve greater than 95% purity. Dead cells were excluded by staining with LIVE/DEAD viability dye (Thermo).

## Western blot protocol

For western blot analysis, cells were washed with ice-cold PBS and lysed in 1× Cell Signaling lysis buffer (20 mM Tris-HCl, (pH 7.5), 150 mM NaCl, 1 mM Na2EDTA, 1 mM EGTA, 1% Triton X-100, 2.5 mM sodium pyrophosphate, 1 mM β-glycerophosphate, 1 mM Na$_3$VO$_4$, 1 µg ml$^{-1}$ leupeptin (Cell Signaling Technologies), supplemented with 1 mM phenylmethylsulfonyl fluoride. Samples were frozen and thawed 3 times, followed by centrifugation at 20,000g for 10 min at 4 °C. Cleared protein lysate was denatured with LDS loading buffer for 10 min at 70 °C and loaded on precast 4% to 12% Bis-Tris protein gels (Life Technologies). Proteins were transferred on to nitrocellulose membranes using an iBLOT 2 system (Life Technologies) following the manufacturer's protocols. Membranes were blocked with 5% w/v milk and 0.1% Tween-20 in Tris-buffered saline (TBS) and incubated with the appropriate antibodies in 5% w/v bovine serum albumin in TBS with 0.1% Tween-20 overnight at 4 °C. All primary antibody incubations were followed by incubation with secondary HRP-conjugated antibody (Pierce) in 5% milk and 0.1% Tween-20 in TBS and visualized using SuperSignal West Pico or Femto Chemiluminescent Substrate (Pierce) on Biomax MR film (Kodak) or a Bio-Rad ChemiDoc. The following antibodies were used: anti-DHPS (Abcam), anti-GAPDH, anti-EIF5A (BD Bioscience), anti-EIF5A-hypusine (Millipore). All antibodies were used at a dilution of 1:1,000.

## Mouse tissue histology and histopathologic staining

Perfused lungs, liver, kidney, heart, spleen and brain were fixed overnight in 10% formalin at 4 °C. Lungs were inflated with 1 ml of formalin before harvesting. All organs were cut into 4-µm-thick formalin-fixed paraffin-embedded (FFPE) tissue sections. Liver sections were stained

with H&E or MAS. All sectioning and staining were performed by The Johns Hopkins University Oncology Tissue and Imaging Services Core Laboratory. H&E- and MAS-stained liver slides from CL-treated mice and control images were observed and pathologically described by Y.Z. Immunohistochemical staining was performed by the Johns Hopkins University Oncology Tissue and Imaging Services Core Laboratory. Lung sections were stained with an antibody against F4/80 (brown) and lung tissues using antibodies against CD31 for vasculature (purple), EpCAM for epithelial cells (teal) and haematoxylin (indigo-like blue) for nuclei. H&E, MAS and immunohistochemically stained slides were whole-slide scanned using Hamamatsu NanoZoomer XR and NDP software (Hamamatsu Photonics, NDP.scan). NDP.view2 (Hamamatsu Photonics) was used for image analysis.

## Immunofluorescence staining and image acquisition of FFPE tissue sections

For immunofluorescence staining of lung, liver, kidney, heart, spleen and brain with FFPE sections of mouse tissues, heat-induced epitope retrieval with Diva Decloaker (Biocare Medical, DV2004MX) was performed after deparaffinization and rehydration. Then, slides were blocked with 2.5% normal donkey serum (Jackson ImmunoResearch, 017-000-121) for 30 min at room temperature and incubated with the cocktail of primary antibodies for 16 h at 4 °C. Slides were washed with PBST (Dulbecco's phosphate-buffered saline with 0.05% Tween-20), then incubated with the cocktail of secondary antibodies conjugated with fluorochrome for 2 h at room temperature. A Streptavidin/Biotin Blocking Kit (VectorLabs, SP-2002) was used before the incubation with antibodies when endogenous biotin blocking was needed. Slides were cover-slipped with EverBrite TrueBlack Hardset Mounting Medium with DAPI (Biotium, 23018) or ProLong Gold Antifade Mountant (Invitrogen, P36930) after nuclear staining with 1 μg ml$^{-1}$ DAPI (Thermo Scientific, 62248) for 10 min at room temperature. Images were acquired with Yokogawa Spinning Disk Field Scanning Confocal Systems (Nikon, CSU-W1 SoRa), ZEISS Axioscan 7 or Vectra Polaris (Akoya Biosciences). Images were analysed using NIS Elements (Nikon), ImageJ (v.1.54f, National Institutes of Health), ZEN (ZEN lite, v.3.9.101, ZEISS) and QuPath-0.5.0-x64. For lung sections, five image fields per sample were selected to be analysed.

For the measurement of macrophages of lung (cell counts, area and circularity), five image fields (202,536 μm$^2$ for each field) per sample were randomly selected from whole-slide scanned images using ZEISS Axioscan 7 and ZEN software. Random fields were at least 800 μm away from each other; this was confirmed after the selection. The formula (circularity) = $4\pi \times$ (area)/(perimeter)$^2$ was used to calculate sphericity, with a value close to 1 indicating a round object (sphericity = 1 for an exact circle). ImageJ was used for measurement. The following primary antibodies were used: anti-PDGFRα (R&D Systems, AF1062, polyclonal, 1:200 dilution), anti-F4/80 (Bio-Rad, MCA497GA or MCA497B, clone Cl:A3-1, 1:300 or 1:50), anti-CD64 (Invitrogen, MA5-29706, clone 027, 1:300), anti-GFP (Abcam, for YFP detection, ab5450, polyclonal, 1:400), anti-IBA1 (Synaptic Systems, HS-234 308, clone Gp311H9. 1:300) and anti-TIM-4 (BioLegend, 130002, clone RMT4-54, 1:400). Streptavidin conjugated with Alexa Fluor 488 (Invitrogen, S32354) or Alexa Fluor 594 (Invitrogen, S32356) was used for biotin-conjugated primary antibody detection. The following secondary antibodies conjugated with fluorochrome were used: Alexa Fluor Plus 488 (Invitrogen, A32814 and A32790), Alexa Fluor Plus 555 (Invitrogen, A48270 and A32794), Alexa Fluor 594 (Jackson ImmunoResearch, 706-585-148) and Alexa Fluor Plus 647 (Invitrogen, A32795 and A32849).

## Immunofluorescence staining of frozen sections from mouse kidney and brain

After transcardial perfusion with PBS, kidneys and brains were harvested and fixed in 4% paraformaldehyde (Electron Microscopy Sciences, catalogue no. 15710) at 4 °C for at least 24 h. Fixed tissues were dehydrated in 30% sucrose solution (Sigma, S1888) for at least 48 h and embedded in Tissue-Plus O.C.T. compound (Fisher Scientific, 23-730-571). Cryosections, prepared by The Johns Hopkins University Oncology Tissue and Imaging Services Core Laboratory, were obtained at a thickness of 10 μm (kidney) or 15 μm (brain). Cryosections were air-dried and blocked for 1 h with 2.5% normal donkey serum (Jackson Immuno-Research, 017-000-121) and 0.3% Triton X-100 (Sigma) in PBS, followed by incubation with anti-GFP (Abcam, ab5450, 1:500), anti-IBA1 (Invitrogen, PA5-27436, 1:500) diluted in blocking buffer at 4 °C overnight. For anti-cleaved caspase-3 (CST, 9661, polyclonal, 1:300) staining, epitope retrieval was performed using Antigen Retrieval Citra Plus (Biogenex, HK080-9K). After washing, sections were incubated with secondary antibodies (1:500) conjugated with Alexa Fluor Plus 647 (Invitrogen, A32849) and Alexa Fluor 568 (Invitrogen, A10042,) at room temperature for 1 h. Slides were cover-slipped with mounting medium (Vector Laboratories, H-1900-10), after staining of nuclei with 1 μg ml$^{-1}$ DAPI (Sigma, D8417) for 3 min at room temperature. Images were captured using a confocal microscope (Zeiss LSM880 Airyscan). For kidney, at least four fields per sample were randomly captured and analysed using Imaris software (10.0.1, Bitplane) to measure the volume and sphericity of YFP-expressing macrophages. For cleaved caspase-3 analysis, whole-slide scanning was performed with a Vectra Polaris (Akoya Biosciences), and images were analysed using QuPath-0.5.0-x64. For brain, images of the hippocampal region were captured and denoised using ImageJ (v.1.54g, National Institutes of Health). The volume and sphericity of microglia were analysed using Imaris (10.0.1, Bitplane).

## Proteomics procedure and analysis

Protein samples were prepared with $5 \times 10^6$ cells using an iST 96X kit (PreOmics), according to the manufacturer's recommendations. All samples used for data-dependent acquisition (DDA) and data-independent acquisition (DIA) analyses were spiked with index retention time kit peptides (Biognosys), according to the manufacturer's instructions. DIA spectral libraries were generated with Spectronaut (Biognosys) v.10.0 using MaxQuant (https://www.maxquant.org/) results as an input[63]. For the latter, DDA runs (using two or three biological replicates from each biological conditions) were acquired using a Q Exactive Plus instrument, and data were searched using MaxQuant (v.1.6.1.0). The spectral library was constructed using a false discovery rate cutoff of 1% and minimum and maximum of 3 and 6 fragment ions, respectively, and protein grouping was performed according to the MaxQuant search results.

For mass spectrometric acquisition, the general nanoscale liquid chromatography mass spectrometry setup was similar to that previously described[63] with minor modifications. A Q Exactive Plus mass spectrometer (Thermo Fisher) and an Easy nanoLC-1200 (Thermo Fisher) were used for both DDA and DIA experiments. For the chromatographic separation of peptides, 4 μg peptide digest was analysed at 50 °C (controlled by a Sonation column oven) on a 50-cm in-house-packed fused-silica emitter microcolumn (75 μm inner diameter, 360 μm outer diameter, 8 μm tapered open end; SilicaTip PicoTip; New Objective) packed with 1.9-μm reverse-phase ReproSilPur C18-AQ (120 Å) beads (Dr Maisch). Peptides were separated using a binary solvent system consisting of 0.1% (v/v) formic acid (solvent A) and 80% (v/v) acetonitrile/0.1% (v/v) formic acid (solvent B). Peptides were loaded at 400 nl min$^{-1}$ at 0% B and then separated by the following linear gradients: from 2% to 10% solvent B over 5 min at 250 nl min$^{-1}$, from 10% to 35% over 180 min at 250 nl min$^{-1}$, from 35% to 50% over 26 min at 250 nl min$^{-1}$, and from 50% to 80% over 6 min at 250 nl min$^{-1}$, and kept at 80% for 10 min at 250 nl min$^{-1}$, followed by an inverse gradient from 80% to 5% over 6 min and a re-equilibration step over 5 min at 5% B (300 nl min$^{-1}$).

For data analysis, MS2-based label-free quantification was carried out by processing DIA raw data using Spectronaut (v.10.0) software with default parameters as previously described[63] with minor modifications.

In brief, the decoy method was set to 'mutated', data extraction and extraction window were set to 'dynamic' with correction factor 1, identification was set to 'normal distribution P-value estimator' with $q$-value cutoff of 0.1 and the profiling strategy was set to 'iRT profiling' with $q$-value cutoff of 0.01. Ultimately, protein quantity was set to 'average precursor quantity' and the smallest quantitative unit was set to 'precursor ion' (summed fragment ions). For statistical testing and identification of deregulated proteins in all approaches, a two-sample Student's $t$-test was used to identify differentially expressed proteins filtered to 1% false discovery rate.

## scRNA-seq procedure and analysis

For peritoneal macrophage sequencing, single cells were isolated from the peritoneal cavities of the indicated animals with 3–4 biological replicates per genotype (*Dhps*-WT and *Dhps*-ΔM) and prepared for scRNA-seq using the GemCode Single Cell Platform with GemCode Gel Beads, Chip and Library Kits (v.3) and 10x Genomics Chromium Controller following the manufacturer's protocol. Peritoneal lavage libraries were sequenced on a NovaSeq 6000 (Illumina). For lung macrophage sequencing, single cells were isolated from digested lungs as described before, and LIVE CD45$^+$YFP$^+$ cells were FACS sorted from three biological replicates per genotype for *Dhps*$^{+/+}$ LysM–Cre Rosa26$^{eYFP}$ (control) and *Dhps*-ΔM Rosa26$^{eYFP}$ mice. Cells were submitted to sequencing company OMAPiX for 10x Genomics library preparation using the GEM-X 3′ v.4 gene expression kit. Lung libraries were sequenced on a NovaSeq X Plus (Illumina).

Samples were demultiplexed, quality checked, filtered and aligned with genome build GRCm38 using pre-established pipelines implemented in snakePipes[64] with STARsolo v.2.7.4a[65], deeptools v.3.3.2, seqtk v.1.3, pigz v.2.3.4, snpsplit v.0.3.4, samtools v.1.10, fastqc v.0.11.9, cutadapt v.2.8, trim-galore v.0.6.5, multiqc v.1.8, fastp v.0.20.0, umi_tools v.1.0.1 and star v.2.7.4a. For lung macrophage libraries, CellRanger v.9.0.1 was used to obtain count matrices. Resulting raw read count matrices of barcodes corresponding to cells and features corresponding to detected genes were processed, analysed and visualized in R v.4.3.1 (ref. 66) using Seurat v.4 (ref. 67) with default parameters for all functions unless otherwise specified. Poor-quality cells with low total unique molecular identifier counts and high percentages of mitochondrial gene expression were excluded. Filtered samples were normalized using a regularized negative binomial regression (SCTransform)[67] and integrated with the reciprocal principal component analysis approach followed by mutual nearest neighbours, using 50 principal components. Integrated gene expression matrices were visualized with UMAP[68] as a dimensionality reduction approach. Resolution for cell clustering was determined by evaluating hierarchical clustering trees at a range of resolutions (0–1.2) with Clustree[69], selecting a value that induced minimal cluster instability. Datasets were subsetted to include only macrophages, on the basis of expression of key macrophage markers (*Adgre1*, *Csf1r*, *H2-Ab1*, *Cd68*, *Lyz2*, *Itgam* and *Mertk*). Macrophage-only datasets were then split along conditions and processed anew as described above. DEGs between clusters were identified as those expressed in at least 25% of cells with a log fold change greater than +1 and an adjusted $P$ value of less than 0.01, using the FindMarkers function in Seurat v.4 with all other parameters set to default. Ribosomal protein genes were excluded from the results. Cluster-specific genes were explored for pathway enrichment using StringDB[70]. DEGs (adjusted $P < 0.05$, $\log_2$ fold change > 0.5) across clusters were subjected to gene ontology pathway enrichment analysis using DAVID[71] (v.2016 and v.2021). Gene set scores were calculated using UCell with default parameters[72].

**Bulk RNA-seq procedure and analysis.** RNA-seq data were obtained by sorting 5,000–100,000 cells from each population directly into RLT buffer (Qiagen) containing 1% 2-ME and submitting them to the Emory Integrated Genomics Core at Emory University. Total RNA was isolated using a Quick-RNA Microprep Kit (Zymo Research), and cDNA was generated using a SMART-Seq v.4 Ultra Low Input RNA Kit for Sequencing (Takara Bio) and used to generate sequencing libraries with a Nextera XT kit (Illumina). Libraries were pooled at equimolar ratios and sequenced on a NovaSeq 6000 at approximately 100M reads per sample using 100PE reads. Sequenced libraries were processed for analysis with deepTools[73] v.2.0, using STAR[65] v.2.7.10 for trimming and mapping and feature Counts[74] v.2.0.3 to quantify mapped reads. Raw mapped reads were processed in R (Lucent Technologies)[66], using DESeq2 (ref. 75) v.1.36 to generate normalized read counts for visualization as heatmaps with Morpheus (Broad Institute) and to determine DEGs with fold change greater than 1.4 and adjusted $P$ value less than 0.01. Gene ontology analysis was performed with DAVID[71] (v.2016 and v.2021).

## Ribo-seq pulldown, sequencing and analysis

Freshly isolated cells from total peritoneal lavage were processed as described in ref. 76 with the following modifications. One millilitre of homogenization buffer was used per mouse, and dithiothreitol was removed from the washing buffer to avoid uncoupling of conjugated anti-HA antibodies from magnetic beads. Total lysate (100 μl) was used as an input. The remaining lysate was used for immunoprecipitation of polysomes. Twenty-five microlitres of anti-HA.11 (Pierce) were used per lysate with rotation at 4 °C for 4 h. Beads were then washed three times with 500 μl of wash buffer. RNA was extracted from magnetic beads with TRIzol reagent (Invitrogen), and total RNA was isolated with a Direct-zol RNA Microprep Kit (Zymo Research) according to the manufacturer's instructions. Total RNA was submitted to Admera Health for RNA-seq. Isolated RNA sample quality was assessed with an RNA TapeStation (Agilent Technologies) and quantified by AccuBlue Broad Range RNA Quantitation assay (Biotium). Paramagnetic beads coupled with oligo d(T)25 were combined with total RNA to isolate poly(A)+ transcripts in a process based on the NEBNext Poly(A) mRNA Magnetic Isolation Module manual (New England Biolabs). Before first strand synthesis, samples were randomly primed (5′ d(N6), 3′ [N = A,C,G,T]) and fragmented on the basis of the manufacturer's recommendations. The first strand was synthesized with Protoscript II Reverse Transcriptase with a longer extension period, approximately 40 min at 42 °C. All remaining steps for library construction were performed according to the NEBNext Ultra II Directional RNA Library Prep Kit for Illumina (New England Biolabs). Final library quantity was assessed with Qubit 2.0 (Thermo Fisher), and quality was assessed with TapeStation D1000 ScreenTape (Agilent Technologies). The final library size was about 430 bp with an insert size of about 300 bp. Illumina 8-nt dual-indices were used. Equimolar pooling of libraries was performed on the basis of quality control values, and libraries were sequenced on an Illumina NovaSeq X Plus platform with a read length configuration of 150 paired ends for 40 million paired-end reads per sample (20 million in each direction). For analysis, sequenced libraries were processed as described above for bulk RNA-seq analysis, with the differential expression tests and filters specified in the Results section.

## Quantification and statistical analysis

Prism 7 software (GraphPad) was used for statistical analysis, and results are presented as the mean ± s.d., unless otherwise indicated. Comparisons for two groups were performed using unpaired two-tailed Student's $t$-tests, and comparisons of more than two groups used one-way analysis of variance with Bonferroni's multiple comparison tests. Exact $P$ values and details of statistical testing can be found in the figure legends and in the source data file. Unless otherwise specified, $n$ represents the number of individual biological replicates and is represented in graphs as one dot per sample. Flow cytometry plots are representative of at least three replicates, immunoblots of at least two independent experiments, and confocal and immunohistochemistry images of at least three independent biological replicates. No statistical method was used to predetermine sample size, but a minimum of three

samples were used per experimental group and condition. Experiments were not randomized.

## Reporting summary

Further information on research design is available in the Nature Portfolio Reporting Summary linked to this article.

## Data availability

Materials and protocols will be made available to qualified researchers upon request. Raw sequencing data have been deposited to GEO with accession codes as follows: Ribo-seq (GSE290459), scRNA-seq of peritoneal macrophages (GSE290571) and lung macrophages (GSE312873), bulk RNA-seq (GSE290686). Proteomics data have been deposited in PRIDE (PXD054670). Source data are provided with this paper.

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

**Acknowledgements** We thank the members of the Pearce laboratories and R. Green, and acknowledge use of Core facilities at the MPI-IE, the High Parameter Flow Core, and Oncology Tissue and Imaging Services (OTS) supported by CA006973, and the Animal Facilities at SKCCC (Johns Hopkins). This study was supported in part by the Emory Integrated Genomics Core (EIGC) (RRID: SCR_023529) and the Georgia Clinical & Translational Science Alliance of the National Institutes of Health, no. UL1TR002378. This work was also supported in part by two Bloomberg Distinguished Professorships from Johns Hopkins University (to E.L.P. and E.J.P.) and by the National Institutes of Health (AI170599 to E.L.P. and AI177282 to E.J.P.). The study was supported in part by the Singapore BMRC Use-inspired Basic Research (UIBR) Award (F.G.).

**Author contributions** G.E.C., D.J.P. and E.L.P. designed the study; G.E.C., P.L., S.H.L., K.S., C.B., M.C., L.S., Z.S., N.v.T.B., K.M.G., B.K., E.J.P., F.G., D.E.S., D.J.P. and E.L.P. designed and/or performed experiments; G.E.C., D.E.S., K.S., K.L.S., G.M., E.J.P., F.G., Y.Z. and N.S. provided conceptual input; G.E.C., P.L., C.B., L.S., Y.M., Y.Z., S.H.L., D.E.S., D.J.P. and E.L.P. analysed data; and G.E.C., D.J.P. and E.L.P. wrote the manuscript.

**Competing interests** E.L.P. is a SAB member of ImmunoMet Therapeutics and Cour Therapeutics, and E.L.P. and E.J.P. are Scientific Advisors to Remedy Plan Therapeutics.

**Additional information**
**Correspondence and requests for materials** should be addressed to Daniel J. Puleston or Erika L. Pearce.

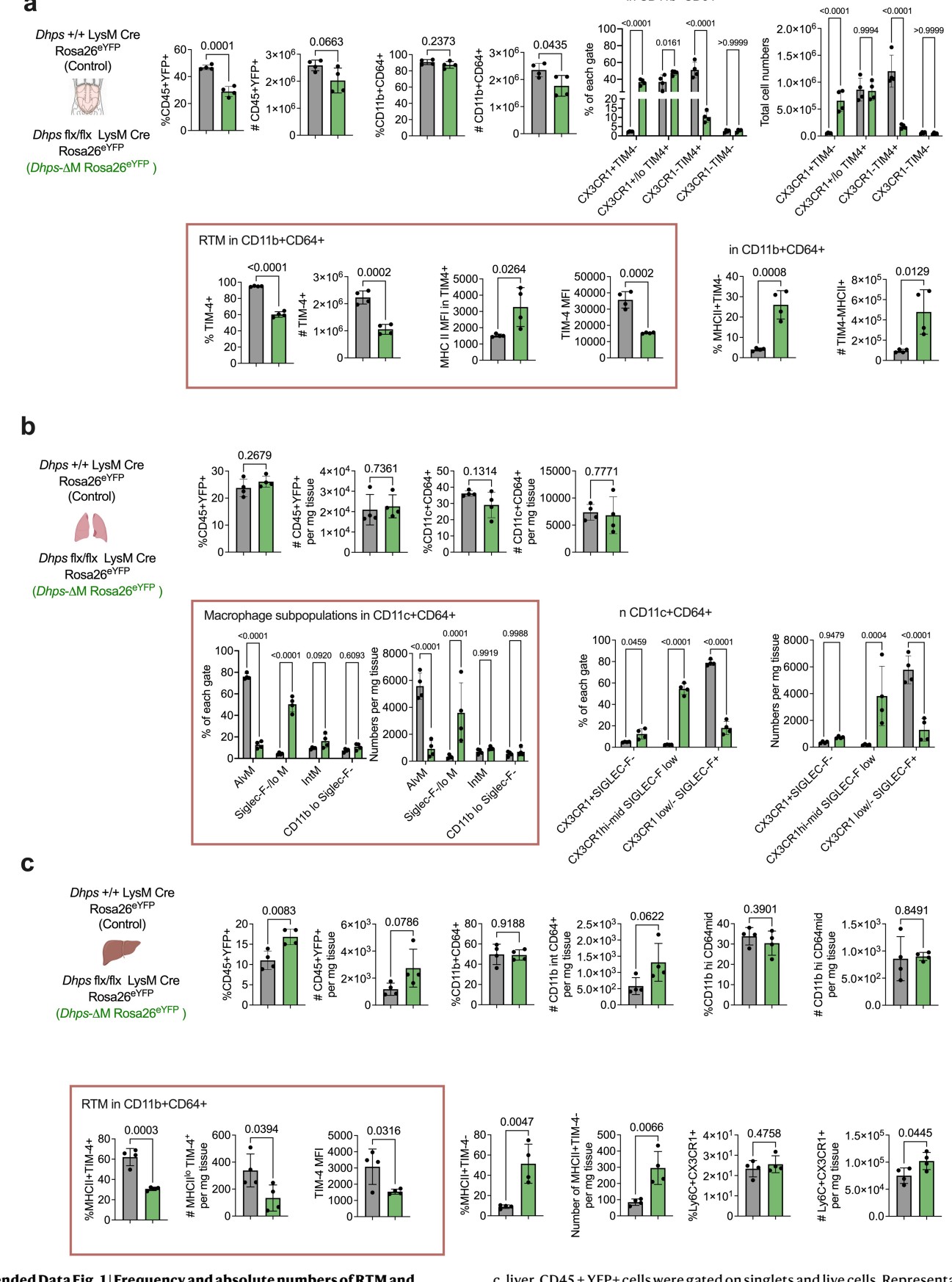

**Extended Data Fig. 1 | Frequency and absolute numbers of RTM and macrophage subpopulations in peritoneum, lung and liver.** Macrophage subpopulations from *Dhps*[+/+] Rosa26[eYFP] (control) or *Dhps*-ΔM Rosa26[eYFP] reporter mice were measured by flow cytometry in a, peritoneum, b, lung, and c, liver. CD45 + YFP+ cells were gated on singlets and live cells. Representative plots and graphs summarize results of at least two independent experiments. Data are mean ± s.d. Exact P-values are indicated. Illustrations in **a–c** were created using BioRender (https://biorender.com).

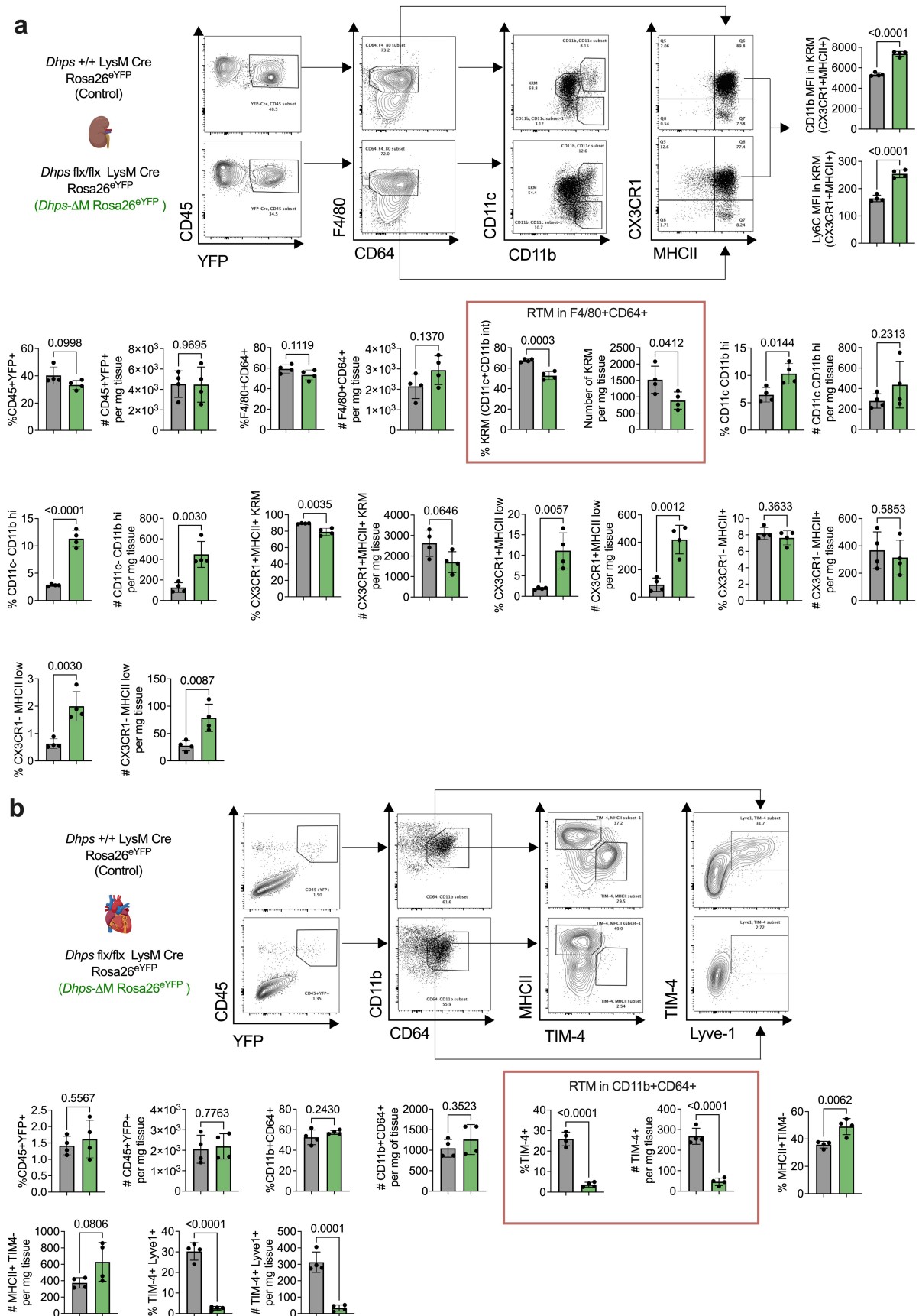

**Extended Data Fig. 2 | Frequency and absolute numbers of RTM and macrophage subpopulations in kidney and heart.** Macrophage subpopulations from *Dhps*[+/+] Rosa26[eYFP] (control) or *Dhps*-ΔM Rosa26[eYFP] reporter mice were measured by flow cytometry in a, kidney, and b, heart. CD45 + YFP+ cells were gated on singlets and live cells. Representative plots and graphs summarize results of at least two independent experiments. Data are mean ± s.d. Exact P-values are indicated. Red boxes indicate RTM populations. Illustrations in **a** and **b** were created using BioRender (https://biorender.com).

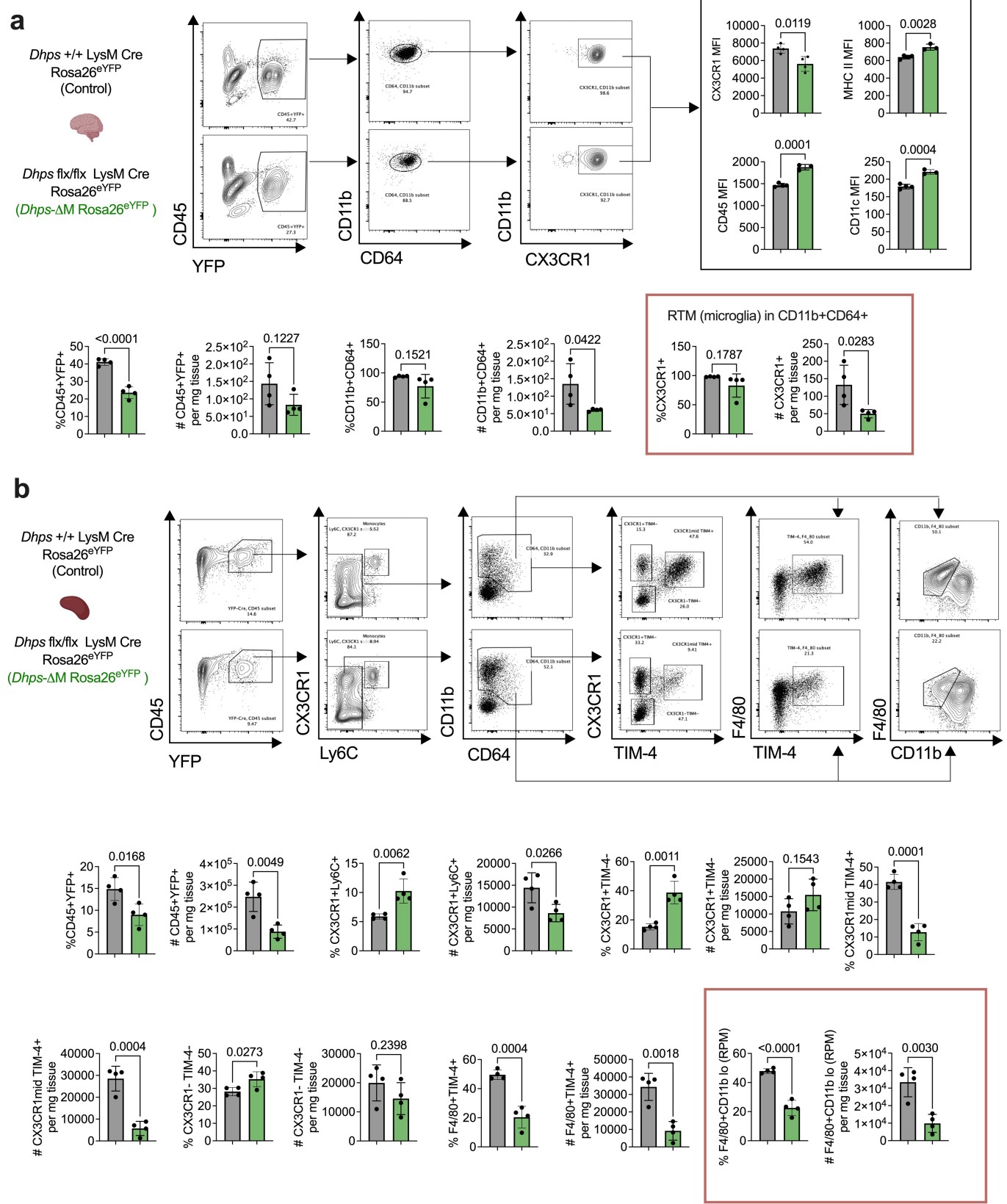

**Extended Data Fig. 3 | Frequency and absolute numbers of RTM and macrophage subpopulations in brain and spleen.** Macrophage subpopulations from *Dhps*[+/+] Rosa26[eYFP] (control) or *Dhps*-ΔM Rosa26[eYFP] reporter mice were measured by flow cytometry in a, brain, and b, spleen. CD45 + YFP+ cells were gated on singlets and live cells. Representative plots and graphs summarize results of at least two independent experiments. Data are mean ± s.d. Exact P-values are indicated. Red boxes indicate RTM populations. Illustrations in **a** and **b** were created using BioRender (https://biorender.com).

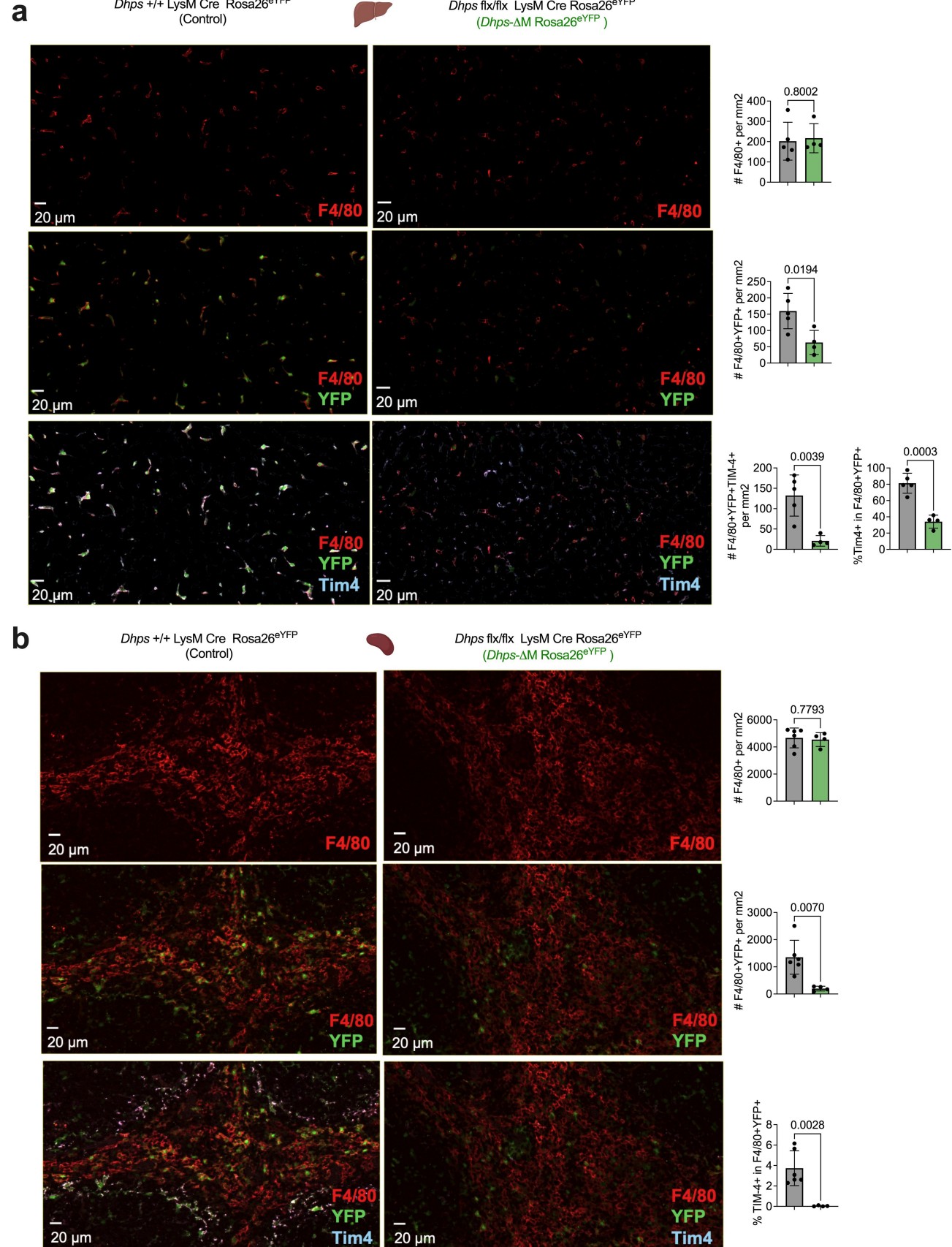

**Extended Data Fig. 4** | See next page for caption.

**Extended Data Fig. 4 | *Dhps*-ΔM Rosa26^eYFP mice show decreased numbers of TIM-4 + YFP+ macrophages in liver and decreased numbers of F4/80 + YFP + red pulp macrophages and F4/80 + YFP + TIM-4+ macrophages in spleen.** Representative immunofluorescence images of a, liver, and b, spleen from adult *Dhps*^+/+ Rosa26^eYFP and *Dhps*-ΔM Rosa26^eYFP mice. Total macrophages were stained with fluorescently-tagged antibodies against F4/80 for liver and spleen sections, and TIM-4 to detect Kupffer cells in liver sections and TIM-4 expressing cells in spleen. Anti-GFP was used to detect YFP⁺ cells (LysM Cre), and nuclei were visualized by DAPI staining. Quantification of F4/80⁺, F4/80⁺ YFP⁺ and F4/80⁺YFP⁺TIM-4⁺cells per unit area (mm²) in *Dhps*^+/+ Rosa26^eYFP and *Dhps*-ΔM Rosa26^eYFP mice is shown for each tissue. The percentage of TIM-4⁺ cells within the total F4/80⁺YFP⁺ population was calculated for both tissues. All images and graphs are representative of 4–5 mice per group. Quantification for liver was performed across the entire tissue section and expressed as cells per unit area. Data are mean ± s.d. Exact P-values are indicated. Red boxes indicate RTM populations. Illustrations in **a** and **b** were created using BioRender (https://biorender.com).

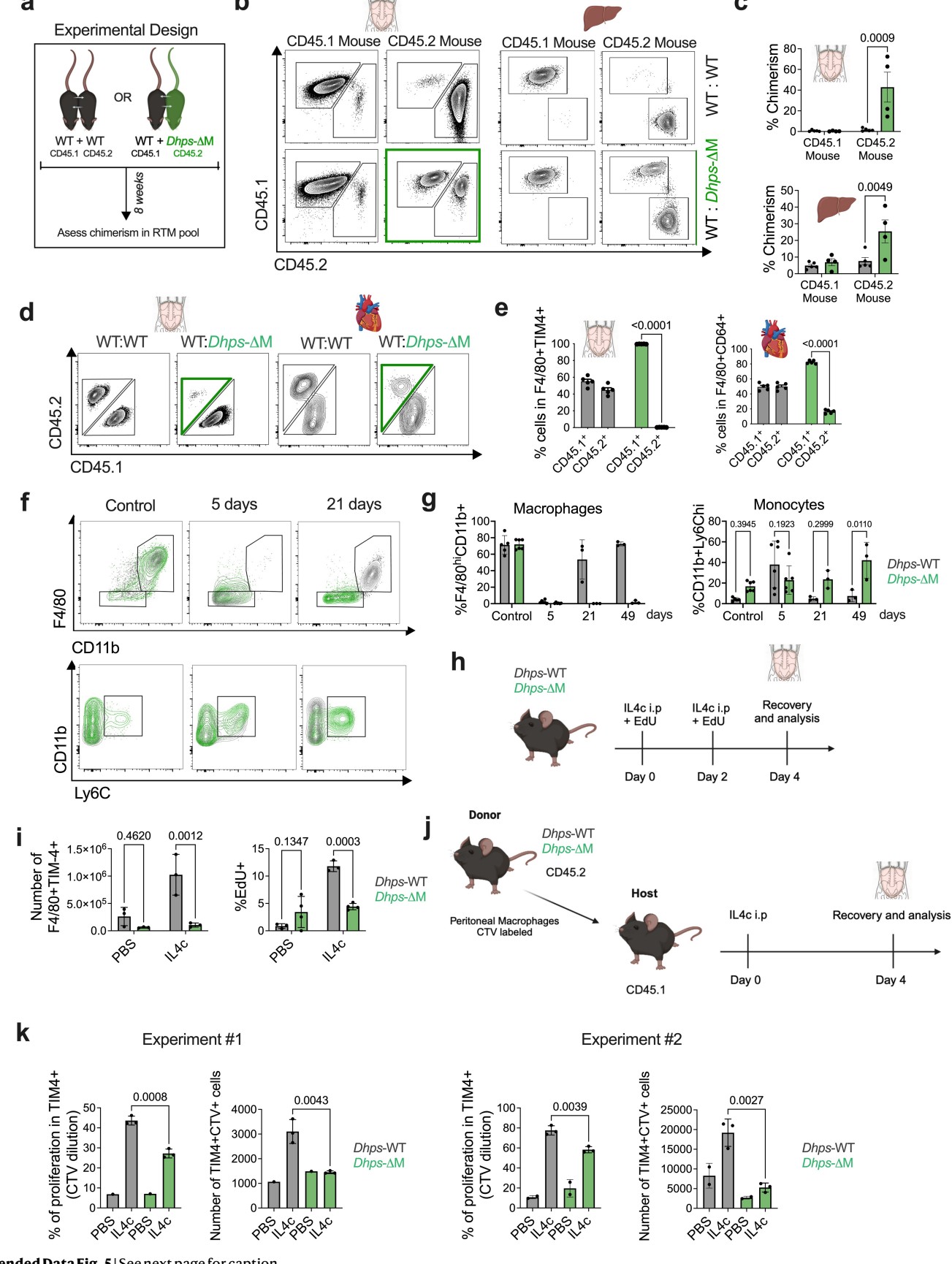

**Extended Data Fig. 5** | See next page for caption.

**Extended Data Fig. 5 | DHPS-deficient macrophages have defects in replenishing the RTM pool and self-renewal. a**, Parabiosis experimental design (CD45.2+ *Dhps*-ΔM mice were joined via the circulation to CD45.1 + WT mice, and as controls, CD45.2+ *Dhps*-WT mice were joined to the blood supply of WT CD45.1+ mice). **b**, Representative flow cytometry plots, and **c**, quantification of CD45.1:CD45.2 chimeric frequencies after 8 weeks of parabiosis in RTM in peritoneum (F4/80 + CD11b + TIM4) and liver (F4/80 + CD64 + TIM4). **d**, Representative flow cytometry plots from bone marrow chimeras, and **e**, quantification of CD45.1:CD45.2 chimeric frequencies of RTM in peritoneum and heart assessed at 12 weeks after BM transfer. **f, g**, Frequency of peritoneal macrophages and recruited monocytes in the peritoneal cavity of *Dhps*-WT and *Dhps*-ΔM mice at indicated times after i.p. clodronate liposome administration. **h**, Experimental design of IL-4 complex (IL4c) and EdU administration in *Dhps*-WT and *Dhps*-ΔM mice. **i**) Absolute numbers of F4/80 + TIM4+ peritoneal macrophages and percentage of EdU incorporation after IL4c injection in *Dhps*-WT and *Dhps*-ΔM mice. **j**, Adoptive transfer scheme: CD45.2+ cell trace violet labeled (CTV) peritoneal macrophages were injected into CD45.1+ recipient mice. Recipient mice were challenged with IL4c or PBS and after 4 days, proliferation was measured. **k**, Percentage of proliferation and total numbers of CD45.2 + CTV+ peritoneal macrophages 4 days after IL4c administration from two independent experiments. Representative plots and graphs summarize results of at least two independent experiments. Data are mean ± s.d. Exact P-values are indicated. Illustrations in **a**–**e**, **h** and **j** were created using BioRender (https://biorender.com).

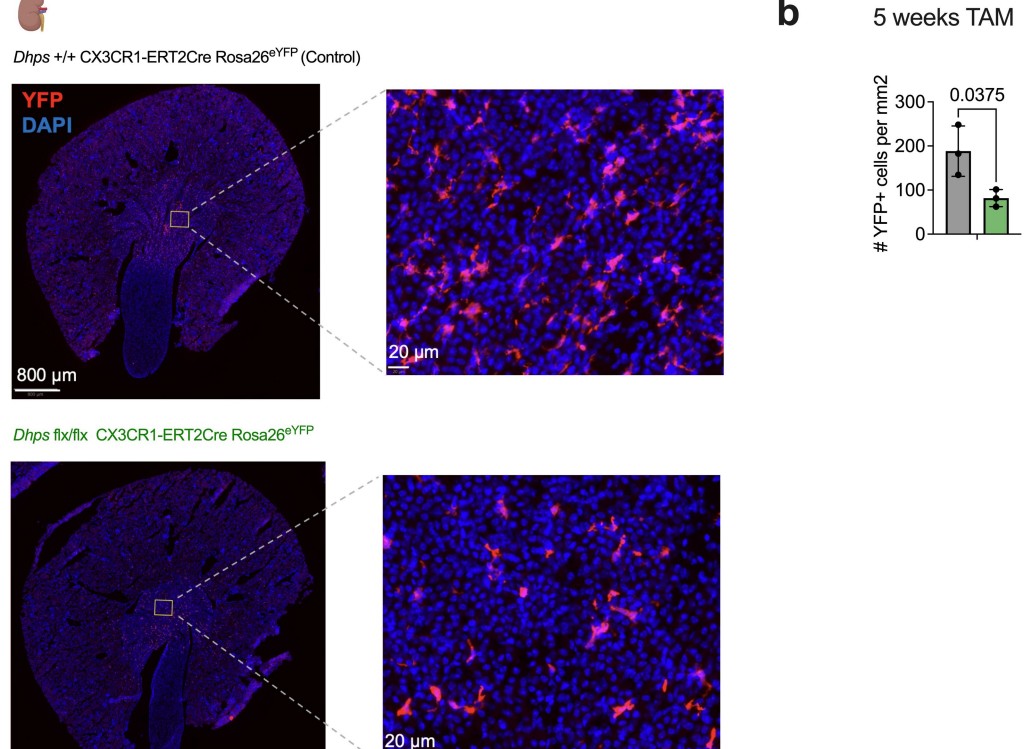

**a** *Dhps* +/+ CX3CR1-ERT2Cre Rosa26$^{eYFP}$ (Control)

YFP
DAPI

800 μm

20 μm

*Dhps* flx/flx CX3CR1-ERT2Cre Rosa26$^{eYFP}$

800 μm

20 μm

**b** 5 weeks TAM

# YFP+ cells per mm2

0.0375

**Extended Data Fig. 6 | Decreased numbers of YFP+ kidney resident macrophages after acute deletion of DHPS by in vivo tamoxifen administration in *Dhps* flox/flox CX3CR1-ERT2 Cre Rosa26$^{eYFP}$ vs *Dhps* +/+ (control) CX3CR1-ERT2 Cre Rosa26$^{eYFP}$ mice. a**, Representative immunofluorescence images of kidney from adult *Dhps* +/+ CX3CR1-ERT2 Cre Rosa26$^{eYFP}$ (control) and *Dhps* flox/flox CX3CR1-ERT2 Cre Rosa26$^{eYFP}$ mice 5 weeks after in vivo tamoxifen administration. CX3CR1-ERT2 Cre-targeted kidney resident macrophages were stained with fluorescently-tagged antibodies against GFP to detect YFP$^+$ cells (CX3CR1-ERT2Cre). Nuclei were visualized by DAPI staining. **b**, Quantification of YFP$^+$ kidney macrophages per unit area (mm$^2$) in *Dhps* +/+ (control) and *Dhps*$^{flox/flox}$ CX3CR1-ERT2 Cre Rosa26$^{eYFP}$ mice are shown. All images and graphs are representative of 3 mice per group. Quantification was performed across the entire tissue section and expressed as YFP+ cells per unit area. Data are mean ± s.d. Exact P-values are indicated. Illustration in **a** was created using BioRender (https://biorender.com).

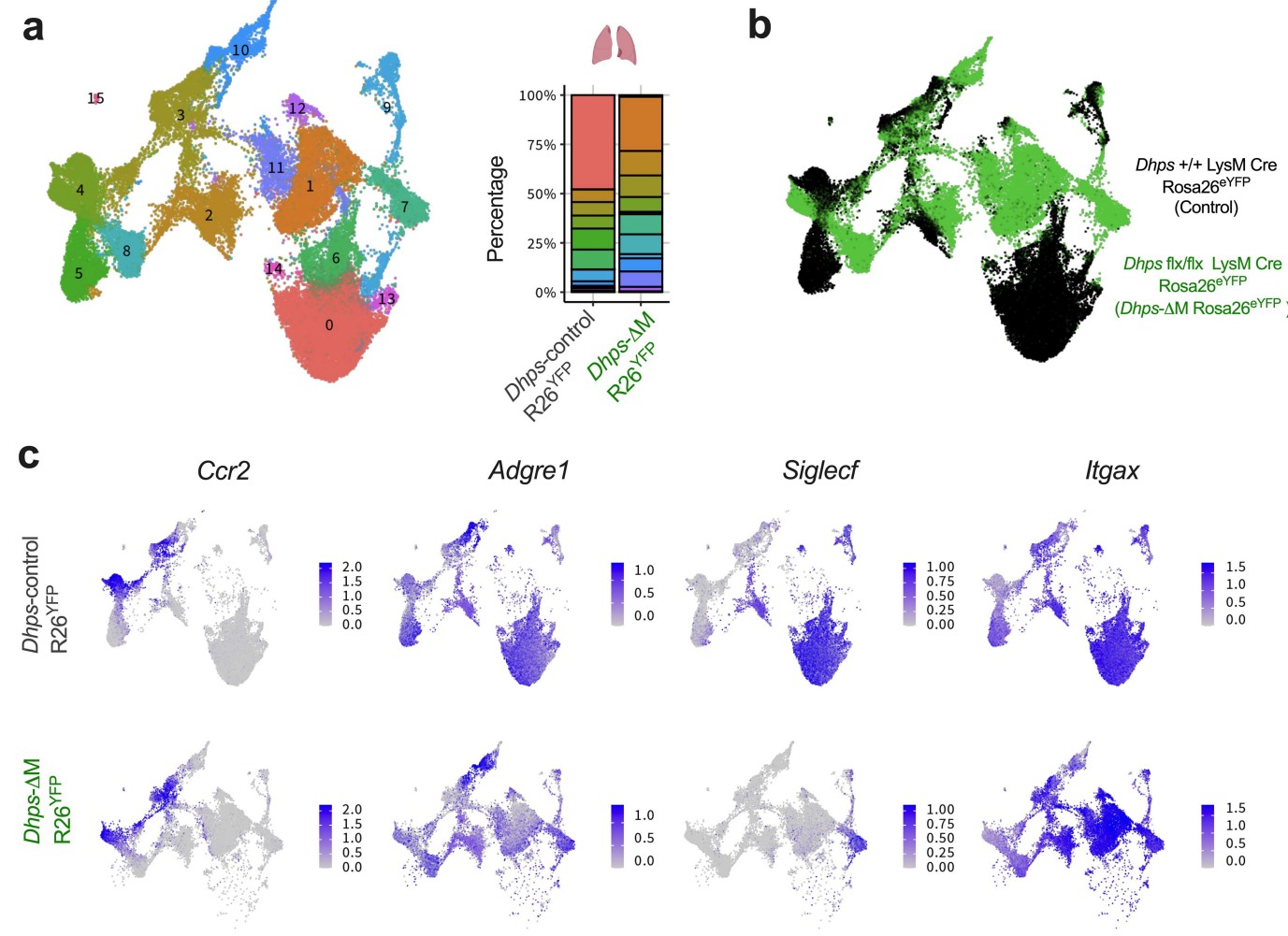

**Extended Data Fig. 7 | Single cell transcriptional analysis indicates a block in the maturation of DHPS-deficient macrophages to RTM in the lung.** **a**, scRNAseq clustering analysis of macrophages from the CD45 + YFP+ sorted population in lung of *Dhps*-Control Rosa26[eYFP] and *Dhps*-ΔM Rosa26[eYFP] mice. The proportion of each cluster within conditions are represented as a percentage. **b**, Overlapped clustering distribution between macrophages from the CD45 + YFP+ sorted population in lung of *Dhps*-Control Rosa26[eYFP] and *Dhps*-ΔM Rosa26[eYFP] mice. **c**, *Ccr2*, *Adgre1* (F4/80), *Siglecf* and *Itgax* (CD11c) expression in macrophages from the CD45 + YFP+ sorted population in lung of *Dhps*-Control Rosa26[eYFP] and *Dhps*-ΔM Rosa26[eYFP] mice. scRNAseq data represent one experiment with 3 biological replicates per condition. Illustration in **a** was created using BioRender (https://biorender.com).

**a**

Peritoneal macrophage data set

Lung macrophage data set

Cluster 2 signature →

**b**

List of DEGs in cluster 2 (peritoneal macrophages)

| | | | | | | | |
|---|---|---|---|---|---|---|---|
| Nrp2 | Tns1 | Rgl1 | Ly9 | Mrc1 | Fcna | Ptges | Acp2 |
| AA467197 | Tgm2 | Sdc4 | Mmp9 | Cp | Tlr2 | Rhoc | Wwp1 |
| Abca1 | Marcksl1 | Ptafr | Fam213b | Cd38 | Cxcl2 | Cxcl9 | Gna12 |
| Clec5a | Frmd4b | C3ar1 | Clec4d | Clec4e | Emp1 | C5ar1 | Klk8 |
| Fcgrt | Tmem86a | Anpep | P2ry6 | Folr2 | Lyve1 | Gprc5b | Nupr1 |
| Sash1 | Mfsd12 | Igf1 | Slc35e3 | Cd63 | Slc7a2 | Hmox1 | Maf |
| Stab1 | Ctsb | Irg1 | Icam1 | Fxyd2 | Pmp22 | Gas7 | Cxcl16 |
| Adap2 | Ccl3 | Ccl4 | Igfbp4 | Vat1 | Trim47 | Timp2 | Cbr2 |
| Aoah | Serpinb6a | F13a1 | Ninj1 | Slc12a7 | Galc | Lgmn | Tnfaip2 |

**Extended Data Fig. 8 | Cluster 2 signature from single-cell transcriptional analysis of peritoneal macrophages identifies unique macrophage clusters in the lung dataset that are not RTMs. a**, Mapping of top upregulated DEGs from cluster 2 in peritoneal macrophage scRNA-seq data (from Fig. 3a–e) onto the lung scRNA-seq dataset. **b**, Gene list of the Cluster 2 signature used to identify "Cluster 2-like" cells in lung scRNA-seq transcriptional analysis.

| | |
|---|---|

# Reporting Summary

## Statistics

For all statistical analyses, confirm that the following items are present in the figure legend, table legend, main text, or Methods section.

| n/a | Confirmed | |
|---|---|---|
| ☐ | ☒ | The exact sample size (*n*) for each experimental group/condition, given as a discrete number and unit of measurement |
| ☒ | ☐ | A statement on whether measurements were taken from distinct samples or whether the same sample was measured repeatedly |
| ☐ | ☒ | The statistical test(s) used AND whether they are one- or two-sided<br>*Only common tests should be described solely by name; describe more complex techniques in the Methods section.* |
| ☐ | ☒ | A description of all covariates tested |
| ☒ | ☐ | A description of any assumptions or corrections, such as tests of normality and adjustment for multiple comparisons |
| ☐ | ☒ | A full description of the statistical parameters including central tendency (e.g. means) or other basic estimates (e.g. regression coefficient) AND variation (e.g. standard deviation) or associated estimates of uncertainty (e.g. confidence intervals) |
| ☐ | ☒ | For null hypothesis testing, the test statistic (e.g. *F*, *t*, *r*) with confidence intervals, effect sizes, degrees of freedom and *P* value noted<br>*Give P values as exact values whenever suitable.* |
| ☒ | ☐ | For Bayesian analysis, information on the choice of priors and Markov chain Monte Carlo settings |
| ☒ | ☐ | For hierarchical and complex designs, identification of the appropriate level for tests and full reporting of outcomes |
| ☒ | ☐ | Estimates of effect sizes (e.g. Cohen's *d*, Pearson's *r*), indicating how they were calculated |

*Our web collection on statistics for biologists contains articles on many of the points above.*

## Software and code

Policy information about availability of computer code

| Data collection | No softwares or codes were used to perform data collection. |
|---|---|
| Data analysis | For references, use the list provided in the manuscript.<br><br>Flow cytometry raw data were analyzed with FlowJo (BD).<br><br>Single cell RNA seq:<br>Samples were demultiplexed, quality checked, filtered and aligned genome build GRCm38 using pre-established pipelines implemented via snakePipes 77 with STARsolo v. 78, deeptools v3.3.2, seqtk v1.3, pigz v2.3.4, snpsplit v0.3.4, samtools v1.10, fastqc v0.11.9, cutadapt v2.8, trim-galore v0.6.5, multiqc v1.8, fastp v0.20.0, umi_tools v1.0.1 and star v2.7.4a. For lung macrophage libraries, CellRanger v.9.0.1 was used to obtained count matrices. Resulting raw read count matrix of barcodes corresponding to cells and features corresponding to detected genes were processed, analyzed and visualized in R v. 4.3.1 79 using Seurat v.4 80 with default parameters in all functions, unless specified. Poor quality cells, with low total unique molecular identifier (UMI) counts and high percent mitochondrial gene expression, were excluded. Filtered samples were normalized using a regularized negative binomial regression (SCTransform) 80 and integrated with the reciprocal principal component analysis (rpca) approach followed by mutual nearest neighbors, using 50 principal components. Integrated gene expression matrices were visualized with a UMAP 81 as a dimensionality reduction approach. Resolution for cell clustering was determined by evaluating hierarchical clustering trees at a range of resolutions (0 - 1.2) with Clustree 82, selecting a value inducing minimal cluster instability. Datasets were subsetted to include only macrophages, based on the expression of key macrophage markers (Adgre1, Csf1r, H2-Ab1, Cd68, Lyz2, Itgam, Mertk). Macrophage only datasets were then split along conditions, and processed anew as described above. Differentially expressed genes between clusters were identified as those expressed in at least 25% of cells with a greater than +1 log fold change and an adjusted p value of less than 0.01, using the FindMarkers function in Seurat v.4 with all other parameters set to default. Ribosomal protein genes were excluded |

from results. Cluster specific genes were explored for pathway enrichment using StringDB 83. Differential expressed genes (pdj value<0.05, Log2Fold>0.5) across clusters were used to perform pathway enrichment analysis by Gene ontology analysis using DAVID 84 (v_2016 and v_2021). Gene set scores were calculated using the UCell with default parameters

Western blot:
Optical density of the signals on the film was quantified using grayscale measurements in ImageJ software (NIH) and converted to fold change, normalized to the loading control or total protein (non-hypusinated).

Fluorescence microscopy:
Lung Images were analyzed using NIS Elements (Nikon), ImageJ (v1.54f, National Institutes of Health, NIH), and ZEN (ZEN lite, v3.9.101, ZEISS). For the measurement of macrophages (cell counts, area, and circularity), 5 image fields (202,536 μm2 for each field) per sample were randomly selected from whole-slide scanned images by ZEISS Axioscan 7 and ZEN software. Random fields are at least 800 μm away from each other, which was confirmed after the selection. The formula: (Circularity) = 4π×(area)÷(perimeter)2 was used for circularity calculation, which represents the object is round when the value is close to 1 (circularity = 1 for the exact circle). ImageJ was used for the measurement. For kidney, at least 4 fields per sample were randomly captured and analyzed using Imaris software (10.0.1, Bitplane) to measure the volume and sphericity of YFP expressing macrophages. For Cleaved Caspase-3 analysis, whole-slide scanning was performed by Vectra Polaris (Akoya Biosciences), and images were analyzed using QuPath-0.5.0-x64. For brain, images of hippocampal region were captured and denoised using ImageJ (v1.54g, NIH).The volume and sphericity of microglia were analyzed using Imaris software (10.0.1, Bitplane).

Bulk RNAseq:
Sequenced libraries were processed with deepTools 10 v_2.0, using STAR 4 v_2.7.10, for trimming and mapping, and feature Counts 11 v_2.0.3 to quantify mapped reads. Raw mapped reads were processed in R (Lucent Technologies) 5 with DESeq2 12 v_1.36 to generate normalized read counts to visualize as heatmaps using Morpheus (Broad Institute) and determine differentially expressed genes with greater than 1.4-fold change and lower than 0.01 adjusted P-value. Gene ontology analysis was performed with DAVID 13 (v_2016 and v_2021)

Proteomics:
The MS2-based label-free quantification was carried out by processing DIA raw data using Spectronaut (version 10.0) software using default parameters as previously described 2 with minor modifications. In brief, the decoy method was set to 'mutated', data extraction and extraction window were set to 'dynamic' with correction factor 1, identification was set to 'normal distribution P-value estimator' with q-value cut-off of 0.1, and the profiling strategy was set to 'iRT profiling' with q-value cut-off of 0.01. Ultimately, protein quantity was set to 'average precursor quantity' and smallest quantitative unit was set to 'precursor ion' (summed fragment ions). For statistical testing and identification of deregulated proteins in all approaches, a two-sample Student's t-test was employed to identify differentially expressed proteins filtered to 1% FDR.

Ribosome associated transcripts sequencing:
Sequenced libraries were processed as described above for Bulk RNAseq analysis and with differential expression tests and filters specified in the results section.

Statistical analysis:
Statistical analysis was performed using Graphpad Prism 7 Software.

For manuscripts utilizing custom algorithms or software that are central to the research but not yet described in published literature, software must be made available to editors and reviewers. We strongly encourage code deposition in a community repository (e.g. GitHub). See the Nature Portfolio guidelines for submitting code & software for further information.

# Data

Policy information about availability of data

All manuscripts must include a data availability statement. This statement should provide the following information, where applicable:
- Accession codes, unique identifiers, or web links for publicly available datasets
- A description of any restrictions on data availability
- For clinical datasets or third party data, please ensure that the statement adheres to our policy

Data that support the findings of this study have been deposited under the following accession numbers: Peritoneal macrophages scRNAseq (GSE290571), Riboseq (GSE290459), RNAseq (GSE290686) and proteomics (PXD054670). Access codes to peer reviewers are available upon request.
Data from scRNAseq lung macrophages will be deposited as soon as GEO (NIH) repository is working again due to the current US government shutdown. Access codes to peer reviewer will be available by then.

# Research involving human participants, their data, or biological material

Policy information about studies with human participants or human data. See also policy information about sex, gender (identity/presentation), and sexual orientation and race, ethnicity and racism.

Reporting on sex and gender

*Use the terms sex (biological attribute) and gender (shaped by social and cultural circumstances) carefully in order to avoid confusing both terms. Indicate if findings apply to only one sex or gender; describe whether sex and gender were considered in study design; whether sex and/or gender was determined based on self-reporting or assigned and methods used.*
*Provide in the source data disaggregated sex and gender data, where this information has been collected, and if consent has been obtained for sharing of individual-level data; provide overall numbers in this Reporting Summary. Please state if this information has not been collected.*
*Report sex- and gender-based analyses where performed, justify reasons for lack of sex- and gender-based analysis.*

Reporting on race, ethnicity, or

*Please specify the socially constructed or socially relevant categorization variable(s) used in your manuscript and explain why*

| Reporting on race, ethnicity, or other socially relevant groupings | *they were used. Please note that such variables should not be used as proxies for other socially constructed/relevant variables (for example, race or ethnicity should not be used as a proxy for socioeconomic status).*<br>*Provide clear definitions of the relevant terms used, how they were provided (by the participants/respondents, the researchers, or third parties), and the method(s) used to classify people into the different categories (e.g. self-report, census or administrative data, social media data, etc.)*<br>*Please provide details about how you controlled for confounding variables in your analyses.* |
|---|---|
| Population characteristics | *Describe the covariate-relevant population characteristics of the human research participants (e.g. age, genotypic information, past and current diagnosis and treatment categories). If you filled out the behavioural & social sciences study design questions and have nothing to add here, write "See above."* |
| Recruitment | *Describe how participants were recruited. Outline any potential self-selection bias or other biases that may be present and how these are likely to impact results.* |
| Ethics oversight | *Identify the organization(s) that approved the study protocol.* |

Note that full information on the approval of the study protocol must also be provided in the manuscript.

# Field-specific reporting

Please select the one below that is the best fit for your research. If you are not sure, read the appropriate sections before making your selection.

☒ Life sciences       ☐ Behavioural & social sciences       ☐ Ecological, evolutionary & environmental sciences

For a reference copy of the document with all sections, see nature.com/documents/nr-reporting-summary-flat.pdf

# Life sciences study design

All studies must disclose on these points even when the disclosure is negative.

| Sample size | No statistical method was used to predetermine sample size, but a minimum of three samples were used per experimental group and condition. Sample size for single cell RNA and bulk RNA sequencing was chosen according to personal experience (n = 3 biological replicates) and in view of financial constraints. |
|---|---|
| Data exclusions | In the single cell analysis data were curated to exclude obvious contaminant cell populations. No data were excluded elsewhere throughout the manuscript. |
| Replication | Majority of the experiments were repeated 3 or more times. Some experiments were repeated only twice<br>or one time, if the outcome was not essential for the validation of our hypothesis. The replicate number is always clearly stated in the legends. |
| Randomization | No conscious biases were used to assign experimental groups. We thus believe that test samples were randomly assigned. |
| Blinding | No blinding was used to assign and constitute experimental groups. Our readouts consist of objective measurements (such as flow cytometry and quantification of western blots) that do not require blinding for unbiased data analysis. |

# Reporting for specific materials, systems and methods

We require information from authors about some types of materials, experimental systems and methods used in many studies. Here, indicate whether each material, system or method listed is relevant to your study. If you are not sure if a list item applies to your research, read the appropriate section before selecting a response.

### Materials & experimental systems

| n/a | Involved in the study |
|---|---|
| ☐ | ☒ Antibodies |
| ☐ | ☒ Eukaryotic cell lines |
| ☒ | ☐ Palaeontology and archaeology |
| ☐ | ☒ Animals and other organisms |
| ☒ | ☐ Clinical data |
| ☒ | ☐ Dual use research of concern |
| ☒ | ☐ Plants |

### Methods

| n/a | Involved in the study |
|---|---|
| ☒ | ☐ ChIP-seq |
| ☐ | ☒ Flow cytometry |
| ☒ | ☐ MRI-based neuroimaging |

## Antibodies

| Antibodies used | Flow cytometry:<br>Complete list of antibodies and clones used for this studies can be found in Supplementary Table 9 from our Supplementary information excel file. |
|---|---|

Western blot:
anti-DHPS (Abcam), anti-GAPDH, anti-EIF5A (BD Bioscience), anti-hypusine (Millipore). All antibodies were used at a dilution of 1:1,000.

Immunofluorescence:
Listed primary antibodies were used: anti-PDGFRα (R&D Systems, #AF1062, polyclonal, 1:200 dilution), anti-F4/80 (Bio-rad, #MCA497GA or #MCA497B, clone Cl:A3-1, 1:300 or 1:50), anti-CD64 (Invitrogen, #MA5-29706, clone 027, 1:300), anti-GFP (Abcam, for YFP detection, #ab5450, polyclonal, 1:400), anti-IBA1 (Synaptic Systems, # HS-234 308, clone Gp311H9. 1:300), anti-IBA1 (Invitrogen, #PA5-27436, 1:500), anti-TIM4 (Biolegend, #130002, clone RMT4-54, 1:400) and anti-Cleaved Caspase-3 (CST, #9661, polyclonal, 1:300). Streptavidin conjugated with Alexa Fluor 488 (Invitrogen, #S32354) or Alexa Fluor 594 (Invitrogen, #S32356) were used for biotin-conjugated primary antibody detection.
Listed secondary antibodies conjugated with fluorochrome were used: Alexa Fluor Plus 488 (Invitrogen, #A32814 and #A32790), Alexa Fluor Plus 555 (Invitrogen, #A48270 and #A32794), Alexa Fluor 594 (Jackson ImmunoResearch, #706-585-148), Alexa Fluor Plus 647 (Invitrogen, #A32795 and #A32849) and Alexa Fluor 568 (Invitrogen, #A10042,).
For nuclei DAPI (Sigma, #D8417) was used.

| Validation | Antibodies were all sourced commercially with independent validations and citations. |
|---|---|

## Eukaryotic cell lines

Policy information about cell lines and Sex and Gender in Research

| Cell line source(s) | *State the source of each cell line used and the sex of all primary cell lines and cells derived from human participants or vertebrate models.* |
|---|---|
| Authentication | *Describe the authentication procedures for each cell line used OR declare that none of the cell lines used were authenticated.* |
| Mycoplasma contamination | *Confirm that all cell lines tested negative for mycoplasma contamination OR describe the results of the testing for mycoplasma contamination OR declare that the cell lines were not tested for mycoplasma contamination.* |
| Commonly misidentified lines (See ICLAC register) | *Name any commonly misidentified cell lines used in the study and provide a rationale for their use.* |

## Animals and other research organisms

Policy information about studies involving animals; ARRIVE guidelines recommended for reporting animal research, and Sex and Gender in Research

| Laboratory animals | For references, use the list provided in the manuscript.<br>Wild-type C57BL/6 , CD45.1 SJL, and mice expressing Cre recombinase (LysM Cre) under the control of the Lysozyme (LysM) promoter, CX3CR1-ERT2 inducible Cre, Rosa26e-YFP (R26YFP) mice and RiboTag mice harboring a modified allele of Rpl22 (Rpl22-HA) that is induced by the action of Cre recombinase, were all purchased from Jackson Laboratories. Dhps conditional floxed mice were a gift from Stefan Balabanov, Zurich. All mice were bred and maintained under specific pathogen free conditions under protocols approved by the Animal Welfare Committee of the Max Planck Institute of Immunobiology and Epigenetics, Freiburg, Germany, and the Institutional Animal Care and Use Committee of Johns Hopkins, Baltimore, USA, in accordance with the Guide for the Care and Use of Animals. Mice used for all experiments were littermates and matched for age and sex (both male and female mice were used). Mice for all strains were typically 6-12 weeks of age unless specified. |
|---|---|
| Wild animals | No wild animals were used in this study. |
| Reporting on sex | Male and female mice have been used in our research, with no differences observed between sexes in any of the described phenotypes. |
| Field-collected samples | No field collected samples were used in this study. |
| Ethics oversight | All mice were maintained in the animal facilities at the Max Planck Institute for Immunobiology and Epigenetics or the Jonhs Hopkins animal facility (CRB1), under specific-pathogen free (SPF) conditions and following institutional animal use and care guidelines. Euthanasia and animal procedures were conducted in compliance to § 4, paragraph 3 of the German Animal Protection Act, animal licenses 35-9185.81/G-20/107 and 35-9185.81/G-20/101, approved by the Regierungspräsidium Freiburg or in compliance with the Institutional Animal Care and Use Committee guidelines of Johns Hopkins. |

Note that full information on the approval of the study protocol must also be provided in the manuscript.

# Plants

| | |
|---|---|
| Seed stocks | *Report on the source of all seed stocks or other plant material used. If applicable, state the seed stock centre and catalogue number. If plant specimens were collected from the field, describe the collection location, date and sampling procedures.* |
| Novel plant genotypes | *Describe the methods by which all novel plant genotypes were produced. This includes those generated by transgenic approaches, gene editing, chemical/radiation-based mutagenesis and hybridization. For transgenic lines, describe the transformation method, the number of independent lines analyzed and the generation upon which experiments were performed. For gene-edited lines, describe the editor used, the endogenous sequence targeted for editing, the targeting guide RNA sequence (if applicable) and how the editor was applied.* |
| Authentication | *Describe any authentication procedures for each seed stock used or novel genotype generated. Describe any experiments used to assess the effect of a mutation and, where applicable, how potential secondary effects (e.g. second site T-DNA insertions, mosiacism, off-target gene editing) were examined.* |

# Flow Cytometry

## Plots

Confirm that:

☐ The axis labels state the marker and fluorochrome used (e.g. CD4-FITC).

☐ The axis scales are clearly visible. Include numbers along axes only for bottom left plot of group (a 'group' is an analysis of identical markers).

☒ All plots are contour plots with outliers or pseudocolor plots.

☒ A numerical value for number of cells or percentage (with statistics) is provided.

## Methodology

| | |
|---|---|
| Sample preparation | Please, see the Methods section for a detailed description of sample preparation before flow cytometry. |
| Instrument | Cells were acquired on BD Fortessa X20, Celesta or Symphony (BD Biosciences), or they were sorted using BD FACS Aria III (BD Biosciences). |
| Software | Raw data were analyzed with FlowJo (BD). |
| Cell population abundance | Cell numbers are stated in the Methods section. Post-sort purity was assessed by re-analyzing the sorted population and was always above 99% pure. |
| Gating strategy | The initial gating for each RTM population consisted in FSC/SSC, singlets (FSC-A/FSC-H), live/dead negative and CD45+ (with YFP in reporter mice), lineage negative (T, B, NK, Ly6G).<br>Each RTM population was furthered characterized by the following markers:<br>Peritoneum: F4/80+CD11bhi or CD64+TIM-4+; Lung: F4/80+CD64+CD11c+Siglec-F+; Liver: F4/80+CD11bmidCD64+TIM-4+; Heart: F4/80+CD64+CD11bmidTIM4+ Brain: CD45midCD11b+; Kidney: F4/80+CD64+CD11c+CD11blow; Spleen: F4/80+CD11blowCD11clow<br>Further description of gating strategy for specific markers within each RTM population can be found in the figure legends. |

☒ Tick this box to confirm that a figure exemplifying the gating strategy is provided in the Supplementary Information.

