## [Peer Review File · Nature]

The monocyte to tissue resident macrophage transition requires DHPS

Corresponding Author: Dr Erika Pearce

Version 0:

Reviewer comments:

Referee #1

(Remarks to the Author)

In this study, Carrizo et al. explore how macrophages attain tissue residency and the cell intrinsic factors that govern monocyte differentiation into tissue resident macrophages (RTM). The authors focused on deoxyhypusine synthase (DHPS), an enzyme involved in polyamine biosynthesis via hypusination of eIF5A which is involved in transcription of mRNA that stall the ribosome. Myeloid deletion of DHPS using a constitutive LysM-cre system revealed reduced tissue resident markers of macrophages and a decline in RTM in numerous tissues. Lack of DHPS results in an open MF niche, allowing for increased monocyte influx, which had reduced differentiation into mature RTM because they are also targeted. Proteomics analyses revealed reduced cell adhesion pathways, which was corroborated by sequencing of ribosome-engaged transcripts using Ribo-Tag mice, as well as altered morphology in at least one macrophage subtype. Macrophages lacking DHPS were also functionally different, such as increased lung proteinosis in homeostasis. In contrast, the lack of DHPS was also beneficial in the context of cancer and infection due to increased inflammation. Therefore, this study reveals that DHPS is essential for the differentiation of RTM.

Overall, this is an interesting study that could be valuable and of broad interest to macrophage biologists, however key gaps exist. My two major gaps are focused on 1) the lack of mechanism that defines how DHPS (or downstream integrin signaling) drives MF differentiation, and 2) Given the paper is focused on broad RTM biology, the study reads premature - with some tissues MFs assessed in one assay or genetic system, but not in a complementary (and required) separate system, creating many gaps and caveats, and a lot of unknowns. As a master regulator of both MF lifecycle and differentiation from monocytes, a more thorough understanding of this process is needed.

Specific Comments

Much of the paper appears to be a single experiment (n=3) - these all needs to be repeated before firm conclusions can be made, and I would reserve judgement until that data is seen. The authors include statistics for n=2 and at times n=1. One example is figure 2, where n=2 is present in the groups, the parabiosis experiment appears to a representative flow plot (no groups or stats). Many other examples are present.

A key deficiency is that data across organs and tissue macrophage subtypes is absent. For example, lung alveolar vs different populations of known lung interstitial, all the subpopulations of liver MFs (kupffer vs capsule MFs), cardiac macrophage subpopulations, each organ has these different populations. Each of these needs to be assessed in each system used in the paper (LysM-cre, Cx3cr1-CreERT2, parabiosis, functional experiments). The authors aspire to shown a broad relevance of across all RTM subsets -thus, the vast majority of MFs in the mouse need to be studied. We all appreciate different MF microenvironments have different requirements - thus is the need for DHPS required in all?

The more detailed studies were done in peritoneal macrophages - but most other tissues are not serosal (ie cells are surrounded by other cells tightly.). Thus, what applies in the peritoneum may or may not apply at other sites.

Most (if not all) the figures rely on % of cells and ratios. What is missing is the absolute abundance of macrophages, including the different subpopulations of MFs. What are the cells /mg in the various gates? Are additional cells trafficking in and diluting out populations? Are ratios changing in each tissue for the same reason?

The post-natal depletion using CX3CR1-ERT2cre-Rosa26 mice is critical data in the paper. This deletes well in many tissues, however this was only assessed (unclear why) in the kidney. This will delete well in brain macrophage subtypes (microglia and different border associated populations), interstitial lung macrophages, cardiac macrophages, and capsular liver macrophages. All of these need to be assessed. The manuscript is about resident tissue macrophages, a more concerted effort to include a large number of tissues is required to make claims about RTMs. Currently, a relatively small subset of tissues is tested, and it is inconsistently tested across experiments.

The link between cell adhesion, which is mentioned numerous times in multiple pieces of data, and DHPS and ultimately macrophage differentiation is not clear, thus a mechanism is not clearly described. The authors state the challenge of performing gain-of-function experiments in this system due to translational block in DHPS-deficient macrophages. Overall the mechanism of how adhesion leads to differentiation isn't clear though, which is the focus of the work. The cell sorting and RNA-seq experiments across tissues is an important study (between control and KO animals). However, there are different macrophage subtypes in those tissues (ie those that express TIMD4 and those that do not, but express other markers), etc. However, in these sorting experiment, all the MFs were pooled - and thus the different subtype information is lost. These MF subset differentially rely on monocyte input, so pooling them together may confound observations. This story has some similarity to the deletion of *cmaf* and *mafb* in interstitial lung macrophages, where monocytes gain entry. Sorting all populations together (monocyte-derived cells) and more differentiated macrophage may be a large issue. It would be critical to show the gates that were sorted (monocytes, monocyte-derived MFs) and more resident populations (Even if they are transient / different). In that way different subsets can be preserved. A large single cell study could also help resolve this issue - but this would require replicates and detailed analysis. Lastly, I do not really understand how the analyses were done here? All the tissues were pooled? which DEGs are changed in each tissue? This experiment and associated figure are difficult to interpret

- Additional questions about Figure 4: scRNA-seq data presented are not sufficient as it stands and require deeper analyses to be of value to readers. There are 11 clusters shown in Figure 4a but it is never stated what these clusters are and what they express. (A) Are all clusters macrophages or are there other immune cells or monocytes included too? Please include canonical macrophage markers. (B) There should be more unbiased representation of the data with appropriate visualization of DEGs in each subset (heatmaps). The authors introduce this section as an opportunity to assess RTM in an unbiased manner, but this is not fully taken advantage of. (C) How do TIMD4+ RTM transcriptionally differ in WT vs DHPS-KO mice? Are the reduced RTM similar to WT just less in number, or do molecular differences exist (this could be in tissue resident macrophage genes but also other pathways)? Direct comparisons should be included, this is the type of data that cannot be acquired with flow cytometry easily but was not well-explored in the manuscript. If there are no differences, this is also important to report.

The altered morphology of tissue macrophages was only assessed in the lung alveolar compartment. As noted above, a multiple tissues need examined. For a pan RTM paper, making a single observation in a single tissue MF subset reduces enthusiasm for the study. Was surfactant itself measured?

Figure 2e: In this analysis, tissue resident macrophages were first gated prior to assess CD45.1 vs CD45.2. It's possible that there could be donor contribution to non TIMD4 macrophages to compensate for the reduced resident pool. The authors should report chimerism in other macrophage subsets as well.

Figure 3c: Was there a difference in CSF1 in the tissue? CSF1R on macrophages may be the same but with less CSF1 present in the environment.

If you inject clodronate liposomes and deplete tissue MFs, that would trigger an inflammatory response with monocyte entry. If you were to then infect or transfer malignant cells into that environment, likely neither would take hold. I understand the functional consequences of impaired MF differentiation leads to more monocytes, but its unclear how it is relevant to challenge such a model. I think more focus on RTM vs monocyte biology in this system, and more detailed mechanistic exploration is needed, rather than functional consequences in a model where homeostasis is so profoundly disturbed, in as the authors state, a multi-factorial fashion.

Referee #2

(Remarks to the Author)

This study from Carrizo et al. seeks to implicate hypusination in the homeostatic process associated with differentiation of macrophages. In particular, data are presented that suggest that the DHPS/hypusinated EIF5A axis places a central role in generation of tissue resident macrophages. The implication of the work is that potentially, modulation/inhibition of hypusination could be used to enhance antimicrobial or antitumoral host defense since data are presented in which knockout of *Dhps* led to increase activation of monocyte-derived macrophages.

There is originality related to the concept of the role of hypusination in macrophage differentiation. However, there are other studies about macrophage function. In particular the study referenced as #55, Gobert et al., Cell Reports, 2020, is not accurately considered. Specifically the current authors state in the Discussion on p. 12 "Previous studies showed that DHPS-deficient macrophages are less prone to cause inflammation in adipose tissue 54, and in a mouse model of

Helicobacter pylori 55." However, Gobert et al. (ref 55) in fact showed exactly the opposite of what is written here in this sentence. In that prior report, the authors showed in mice with myeloid-specific deletion of Dhps (the same approach as in the current manuscript) there was actually increased (not less) gastric inflammation with H. pylori infection, and that this was associated with increased H. pylori bacterial burden (Fig. 5 in the Cell reports paper); moreover Fig. 6 in that same paper showed that for another gut infection, Citrobacter rodentium, a colonic pathogen, there was increased colonic inflammation and increased C. rodentium bacterial load in the colon. These findings were shown to be associated with loss of macrophage immune effector molecules in that study, which goes against the conclusion of the current manuscript that interference with hypusination would enhance host defense against pathogens.

In terms of statistics, the methods overall seem appropriate, but there also seem to be a substantial number of experiments performed in small numbers of mice. Not all of the bar graphs tell us how many animals were used.

In terms of conclusions there are additional concerns. There is no data availability section and the actual list of transcripts in the RNAseq and proteins in the proteomics is not provided, a major omission.

Other specific comments: Fig. 1f thought significant, some of the differences are fairly small and the mouse numbers are low. Are these siblings, what sex, etc. Fig. 2g, can the authors provide more information about the monocyte differentiation, such as with the "waterfall" assay?

Fig 3d, CX3CR1 is not specific to kidney.

Fig. 4 speculates about a pro-inflammatory phenotype, but this should be validated. Fig 4e, f, g has pathways but the actual proteins lists should be in the extended data but are not. How many proteins are up or down, Venn diagrams? Same question for the RiboTag proteins.

Fig 5a, if possible data should be confirmed at the protein level to prevent over-interpretation. Fig. 5 in vitro data about adherence; how can this be validated in vivo?

Fig. 6g: the severity of liver pathology is hard to follow as to how this is all due to a monocyte effect. It is as if these mice have right-sided congestive heart failure because this is what happens in patients with this situation. Or simply some type of acute liver injury like with drug induced necrosis. Perhaps there is more going on in these mice that needs to be worked out.

Fig. 6K is at odds with ref 55 as stated above. It may be that Gram-positive organisms are very different (as studied here) compared to Gram-negative organisms, as in Ref 55. What were the proteins affected by Dhps deletion in lung macrophages? How was there gain of function if less macrophages and less translation?

Fig 6N, this finding may not be true for solid tumors. This should be studied or at least acknowledged to prevent over-statement of the data.

In terms of the conclusions, one major issue is that this study has no human data. Can the authors show that hypusination is required for tissue resident macrophage differentiation in humans? Also there are multiple studies showing that spermidine is beneficial in humans; how do the authors relate that to their conclusion that hypusination, which derives from spermidine, is a target to be blocked?

Referee #3

(Remarks to the Author)

Carrizo et al. investigate the role of DHPS in the differentiation and maintenance of tissue-resident macrophages. Using various mouse models, the authors explore how DHPS-mediated hypusination of eIF5A, a key translation factor, influences macrophage development. They find that mice with myeloid-specific deletion of DHPS (LysM-Cre Dhps-KO mice) exhibit abnormal expression of tissue-resident macrophage markers such as Tim4 and a decline in macrophage populations with age. Parabiosis experiments reveal that this decline creates an open tissue niche, leading to persistent monocyte influx. Further analysis using scRNA seq shows that DHPS-deficient peritoneal macrophages are blocked in their transition from monocytes to mature RTMs. Proteome analysis reveals a decrease in the expression of cell adhesion molecules. Functionally, DHPS-deficient macrophages display impaired efferocytosis and increased lung proteinosis at steady state. However, the authors show that these KO macrophages exhibit enhanced inflammatory functions, resulting in improved protective immunity in models of cancer and infection.

Together, the study characterizes a pathway mediated by the polyamine-hypusine axis, which seems essential for macrophage maintenance, differentiation and function. These findings are interesting, but the manuscript lacks originality in terms of unambiguously defining a novel function for a specific gene that is essential for macrophage biology. It may well be that the inhibition of a key translation factor simply changes everything in a cell, which will impact all its functions.

Major points:

1. It is well known that LysM-Cre is not a good targeting strategy for some macrophage populations. The authors show themselves that the spleen, and especially the brain (microglia are not targeted at all by LysM-Cre, this probably represents only autofluorescence). However, they see clear changes in the analyzed populations. This is hard to explain. Do the authors have an idea what exactly is happening? All tissue macrophages should also be quantified via immunofluorescence and via flow showing cells per gram of tissue and not %.
2. Since LysM-Cre is also active in monocytes, how do the authors explain the decreased expression of YFP in KO animals (Fig. 1f)? This is contradictory to the hypothesis that monocytes replenish the empty niche.
3. Experiments with n=1-2, especially showing p values, is not acceptable.

4. The analysis of the Cx3cr1-CreERT models is somewhat distracting as it is looking at kidney, although before the focus was on the peritoneum, lung and liver. The manuscript would have profited from a holistic approach looking at all organs in all models to describe the differences of affects versus not-affected macrophage populations.

5. Extended Data Fig. 4b: the KO clearly has not only F4/80 macrophages, therefore any result for this RNA-seq analysis may stem from the change of cell composition from 100% F4/80+ cells to only 90%

6. Along these lines, the scRNA-seq shows that macrophages remain somewhat immature in KO animals. This means that for all RNAseq data, the authors should be careful regarding the DEG, especially if it comes to cell adhesion. It could be simply due to the immature state of monocyte-derived macrophages. An in-depth analysis of all clusters in the scRNA-seq may help, and if the cell adhesion is really affected, it would show up in the bona fide macrophage clusters.

7. The lung, with highly motile alveolar macrophages, seems not the best choice to show a lack of adherence. Here, rather the spleen and liver should be analysed. Could the lack of adherence be correlated with the reduced expression of ST2 in these tissues?

8. It is highly unlikely that KO mice die due to the lack of KC and replenishment by Dhps after clodronate treatment as the normal KO without treatment has a normal liver morphology. Did the authors look at the gut? It may be that the gut barrier integrity is not functional any more, which could also have a secondary effect on the liver. The conclusion that the observed phenotype is liver-specific should be removed.

9. Is efferocytosis and phagocytosis in alveolar macrophages and other populations also affected? There are some good in vivo efferocytosis assays. As many macrophage populations are affected, they could be performed instead of looking only at the OD of BAL. Further, aged mice should be analyzed for progression to lung fibrosis or similar to make the point that Dhps is essential for macrophage functions.

10. The point of inflammatory capacity is somewhat weak and contradicts the reduced phagocytic capacity, especially in the Streptococcus model. Also, IL6 production was reduced upon stimulation.

11. The inoculation strategy with 1:1 mix of macrophages is elegant, but in this case has the caveat that KO macrophages are more prone to apoptosis as shown by the cleaved caspase assessment. Therefore, it could be simple cell death that affects tumor growth.

12. The maturation process could be controlled via the altered metabolism of cells. Do they differ e.g. in their oxidative phosphorylation/glycolysis capacity?

13. Do these cells undergo ER stress or other types of stress? What is the cause of increased apoptosis?

Minor points:

1. Fig. 1b-c: why are 4 samples shown per genotype in the WB but only 3 quantified?
2. Gating strategies for representative WT and KO samples used for bulk RNA-seq should be shown

Version 2:

Reviewer comments:

Referee #1

(Remarks to the Author)

The authors have performed a substantial number of new experiments, in particular expanding to other tissue subpopulations beyond the peritoneum. The data indicate a generalized disruption of the macrophage differentiation when DHPS is targeted.

One critical issue was the hypothesis that differentiation was blocked in the DHPS KO, and they show this most clearly in peritoneum with scRNA_seq and many other parallel approaches. The peritoneum is such a different compartment than the interstitium. It would solidify the conclusions and provide more generalizability if scRNA_seq was also performed in another tissue, such as the lung. I appreciate the authors provide flow cytometric evidence, but this is less convincing.

Referee #2

(Remarks to the Author)

The revised manuscript is greatly improved and much more focused.

The main findings are centered on the essential role of hypusination in the differentiation of macrophages into tissue resident cells.

The statistics, data, presentation, and conclusions are improved, as is the clarity of presentation.

I only have a couple of minor suggestions:

All protein names should be all caps, see line 236 where "encodes for Slc7a2" should be written as SLC7A2. All genes should be italicized, see line 231 where "Folr2" should be corrected.

I do not see any method description for the active cascade-3 assays; the antibody information and method is needed; I assume this was by flow cytometry but I do not see where this is mentioned.

Keith T. Wilson

Referee #3

(Remarks to the Author)

The authors have made a substantial effort to address many of the reviewer concerns raised in the first version of the manuscript. They provide additional experimental data, extend their analyses across more tissues, and perform deeper transcriptomic and proteomic assessments. However, they also remove functional data. Further, several key concerns remain insufficiently addressed, and fundamental conceptual issues persist regarding the interpretation of the data and the strength of the conclusions.

1. "These data illustrate that Dhps-dM mice maintain open RTM niches in adulthood." This is a very bold statement that is only deduced from flow cytometry data. The authors should provide counts of Iba1+ macrophages stained in situ across all tissues to support this statement. A lower expression of e.g. Tim4 does not necessarily mean that these macrophages are not tissue resident.
2. "Overall these data show a collapse in the tissue-residency potential": is it now an empty niche or the lack of self maintenance that KO macrophages have? All approaches, including the clodronate and parabiosis experiments, should be supported by IF stainings for Iba1+/F4/80+/CD64+ macrophages. It would also be advisable to combine this with apoptosis markers, as the isolation of macrophages for flow cytometry, as done in the manuscript, can artificially increase apoptosis, particularly affecting more vulnerable populations such as KO macrophages in this study
3. The authors now provide a few IF pictures, in the kidney only 3D rendered cells (include marker used in the figure). However, from these pictures (especially in the Extended data) I cannot see a reduction of macrophages (i.e. "an empty niche"). Especially in the lung there seems to be even more and enlarged alveolar macrophages. Microglia are certainly less ramified (please quantify morphometry across all tissues). But if cell bodies would be counted, there wouldn't be much less in the KO than in the WT from the picture that has been provided. Please provide large tissue sections in the supplements showing Iba1 expression (e.g. sagittal or coronal brain section, whole liver and whole lung lobe, sagittal kidney section etc.).
4. "there are few liver RTMs in Dhps-dM mice": please provide IF pictures of Iba1/F4/80 to be able to conclude this.
5. Since LysM-Cre labels also monocytes, neutrophils and dendritic cells, the authors should correct their sentence "the pulldown was exclusively HA-tagged macrophages." This would also mean that even 5–10% of transcripts coming from other cell types would have a big impact on DEGs (see point 3.6). Therefore, to convince the reader that the few DEGs are indeed macrophage-specific, one would need to show WT and KO gating strategy, including the labelling of other myeloid cells, before showing the histogram for ST2 expression.
6. Supplementary tables should contain the list of all genes and their normalized expression. This is also true for all other RNAseq experiments, where for now only genes from certain GO terms are shown (in line with the reviewer #2 point 2.6). The reviewers should have a chance to assess the raw data in Excel Tables, including all genes and GO terms, not only the once chosen for the figures. It would be useful to plot heatmaps/dotplots/violin plots to show expression of relevant genes across all experiments, instead of only showing GO terms.
7. If ST2 is indeed a DHPS-dependent molecule required for macrophage maturation, its expression should be also shown across all tissues analysed (both flow cytometry and IF).
8. To my knowledge, ST2 KO mice do not have reduced macrophage numbers. Do the authors have an explanation for their phenotype beyond ST2? (see also point 3.8, which in my opinion should be addressed in the context of the current study)
9. The liver phenotype after clodronate could also be due to impaired neutrophils, which are both targeted by the LysM-cre and the clodronate. As the authors state, this is likely completely KC independent, so I am not sure how the authors deduce last paragraph of the results part. Regarding point 1) : this is a well-known fact, and has not been shown by this study. Regarding point 2) : adoptive transfer (WT BM into Dhps-dM mice) could support this statement. But as outlined above, in the manuscript and the rebuttal letter, it remains unclear if this is at all liver macrophage-specific. Models such as Clec4f-Cre will help to delineate this, at least for the liver.

Version 3:

Reviewer comments:

Referee #1

(Remarks to the Author)

The authors have sufficiently addressed my concerns.

Referee #3

(Remarks to the Author)

I appreciate the additional work and clarifications provided in this revised version. The manuscript has certainly improved, and several of the earlier points have been addressed. However, I still have a major conceptual disagreement with the

authors' interpretation of an "empty macrophage niche," which remains a central claim in the paper.

I would strongly urge the authors to reconsider this interpretation, as the data presented do not support the idea of an "empty" niche. The notion that a macrophage niche becomes "empty" implies that the tissue lacks macrophages altogether, which is clearly not the case here. The authors show that Iba1⁺ and F4/80⁺ macrophage numbers remain unchanged between wild-type and knockout mice. This demonstrates that macrophages continue to occupy the tissue space, and therefore the niche is not empty. The absence of Tim4 expression does not define the absence of a macrophage niche—it merely reflects that these macrophages have not acquired the full tissue-resident Kupffer cell-like program.

Furthermore, the authors' interpretation misrepresents the concept of "open" and "closed" tissues. As discussed extensively in the field, this dichotomy has evolved significantly since the 2016 review the authors cite in the rebuttal letter. We now know that even tissues previously considered "open," such as the gut, contain a substantial fraction of long-lived, yolk sac-derived macrophages. The current understanding is therefore that macrophage niches are dynamic but continuously occupied; they may change in cellular composition or turnover rate, but they are not "empty."

I recommend that the authors revise this conceptual framing throughout the manuscript. The phenotype they observe can be more accurately explained as a failure of maintenance or maturation, likely due to impaired DHPS-dependent longevity or differentiation capacity. The macrophages that occupy the Kupffer cell niche in the knockout mice are thus distinct in ontogeny or turnover dynamics, but they are still macrophages. Describing this situation as an "empty niche" risks confusing the reader and contradicts the current consensus in macrophage biology.

In summary, I encourage the authors to:

- Remove the use of the term "empty niche" and related phrasing implying complete absence of macrophages.
- Reframe the interpretation in terms of impaired maintenance, reduced longevity, or altered differentiation within an occupied macrophage niche.
- Clarify that the tissue remains populated by macrophages (Iba1⁺/F4/80⁺), albeit with different identity and stability.

Addressing this conceptual issue will make the paper more accurate and much clearer for the field.

Referees' comments:

First, we would like to thank all the reviewers for their insightful comments on our work. We have done our best to address each point-by-point, specific to each reviewer, below. We feel that the changes we made as a result of reviewers comments have made our revised manuscript much stronger.

Referee #1 (Remarks to the Author):

In this study, Carrizo et al. explore how macrophages attain tissue residency and the cell intrinsic factors that govern monocyte differentiation into tissue resident macrophages (RTM). The authors focused on deoxyhypusine synthase (DHPS), an enzyme involved in polyamine biosynthesis via hypusination of eIF5A which is involved in transcription of mRNA that stall the ribosome. Myeloid deletion of DHPS using a constitutive LysM-cre system revealed reduced tissue resident markers of macrophages and a decline in RTM in numerous tissues. Lack of DHPS results in an open MF niche, allowing for increased monocyte influx, which had reduced differentiation into mature RTM because they are also targeted. Proteomics analyses revealed reduced cell adhesion pathways, which was corroborated by sequencing of ribosome-engaged transcripts using Ribo-Tag mice, as well as altered morphology in at least one macrophage subtype. Macrophages lacking DHPS were also functionally different, such as increased lung proteinosis in homeostasis. In contrast, the lack of DHPS was also beneficial in the context of cancer and infection due to increased inflammation. Therefore, this study reveals that DHPS is essential for the differentiation of RTM.

Overall, this is an interesting study that could be valuable and of broad interest to macrophage biologists, however key gaps exist. My two major gaps are focused on 1) the lack of mechanism that defines how DHPS (or downstream integrin signaling) drives MF differentiation, and 2) Given the paper is focused on broad RTM biology, the study reads premature - with some tissues MFs assessed in one assay or genetic system, but not in a complementary (and required) separate system, creating many gaps and caveats, and a lot of unknowns. As a master regulator of both MF lifecycle and differentiation from monocytes, a more thorough understanding of this process is needed.

Specific Comments

1.1. *Much of the paper appears to be a single experiment (n=3) - these all needs to be repeated before firm conclusions can be made, and I would reserve judgement until that data is seen. The authors include statistics for n=2 and at times n=1. One example is figure 2, were n=2 is present in the groups, the parabiosis experiment appears to a representative flow plot (no groups or stats). Many other examples are present.*

This is an important point and the number of experiments is now clearly mentioned in all Figure Legends. All data in our revised manuscript now represent at least 2 independent experiments with many experiments repeated 3 or more times. The exceptions to this include the 'omics experiments (Fig. 3 scRNA-seq and proteomics; Fig. 4a Ribo-seq; Fig. 4e-g, bulk RNA-seq), where data represented one experiment with 3 biological replicates per condition; and the study where we assessed RTMs longitudinally out to 34 weeks (Fig. 2b), which represents one experiment with 2-4 biological replicates per group, depending on time point, and new Fig 2e, f, and g which represents one experiment with 3 biological replicates per group and time point.

The quantification for the parabiosis and bone marrow chimera experiments was previously in Extended Data Fig. 2a, b. In this revised version, we now show one representative flow plot of the tissue

together with its quantification (Fig. 2c, d). Quantifications and representative flow plots of all other tissues appear in Extended Data Fig. 6a-e.

1.2. *A key deficiency is that data across organs and tissue macrophage subtypes is absent. For example, lung alveolar vs different populations of known lung interstitial, all the subpopulations of liver MFs (kupffer vs capsule MFs), cardiac macrophage subpopulations, each organ has these different populations. Each of these needs to be assessed in each system used in the paper (LysM-cre, Cx3cr1-CreERT2, parabiosis, functional experiments). The authors aspire to show a broad relevance of across all RTM subsets -thus, the vast majority of MFs in the mouse need to be studied. We all appreciate different MF microenvironments have different requirements - thus is the need for DHPS required in all?*

We thank the reviewer for this comment. In our revised manuscript we repeated our experiments across organs in LysM-Cre Rosa26YFP reporter mice using a flow cytometry panel that allows for granular quantification of macrophage subpopulations. We focused the analysis on RTMs for the Main figures (Fig. 1), but Extended Data Figs. 2-4 now show abundance of all macrophage subpopulations (e.g. liver capsule, lung interstitial, etc.). The conclusion from this analysis is that RTMs depend on DHPS in the LysM-Cre model.

Further to this, we also repeated our experiments in the CX₃CR₁-CreERT₂ inducible deletion model and characterized the populations of macrophages targeted by this Cre (i.e. kidney and brain), using Rosa26YFP as a reporter for Cre recombination. Data from this model, presented in Fig. 2j-o, also support the central hypothesis that DHPS is essential for macrophage tissue-residency across tissues.

Employing our CX₃CR₁-ERT₂Cre YFP reporter strain, we also observed small populations of macrophages in both the lung and heart that were YFP⁺ in WT mice, but these populations were absent in Dhps CX₃CR₁-ERT₂Cre mice, supporting the notion that DHPS is vital tissue-occupancy across macrophage populations (Reviewer only Fig. 1 and 2).

In terms of the question of whether DHPS is needed for all tissue macrophages, we can compare results in our 2 mouse models. LysM-Cre Rosa26YFP will continuously label monocyte-derived macrophages in the tissue, therefore we observe no difference in interstitial macrophages in the lungs of these mice when we gate on CD64⁺CD11c⁺CD11b⁺SIGLEC-F⁻ (Extended Data Fig. 2). Does this mean that interstitial macrophages do not require DHPS? We do not think this is the case, and rather, the numbers of cells look the same as WT because in the Lys M-Cre DHPS KO mouse they are being constantly replenished by monocytes. We can resolve this question by looking at the same population in the inducible CX₃CR₁-ERT₂Cre YFP fate mapping strain, where we first label, and then follow, YFP⁺ cells. Here, we can see that the YFP⁺CD64⁺CD11c⁺CD11b⁺SIGLEC-F cells disappear after tamoxifen-induced DHPS deletion (Reviewer only Fig. 2).

1.3. *The more detailed studies were done in peritoneal macrophages - but most other tissues are not serosal (ie cells are surrounded by other cells tightly.). Thus, what applies in the peritoneum may or may not apply at other sites.*

We have now extended many key aspects of the paper to other tissues including our models of macrophage depletion/monocyte tissue entry (lung) (Fig. 2e-g) and cell morphology by confocal imaging in kidney (Fig. 4n) and brain (Extended Data Fig. 13d).

1.4. Most (if not all) the figures rely on % of cells and ratios. What is missing is the absolute abundance of macrophages, including the different subpopulations of MFs. What are the cells /mg in the various gates? Are additional cells trafficking in and diluting out populations? Are ratios changing in each tissue for the same reason?

We understand what an important point this is. To comprehensively address this issue, we now report abundance of cells/mg tissue across a wide range of organs in both our LysM-Cre and CX₃CR₁-CreERT₂ Rosa26YFP models. These data are shown in Figs. 1 and Extended Data Fig. 2-4 for LysM-Cre and Fig. 2j-o for the CX₃CR₁-CreERT₂ model.

1.5. The post-natal depletion using CX₃CR₁-ERT₂cre-Rosa26 mice is critical data in the paper. This deletes well in many tissues, however this was only assessed (unclear why) in the kidney. This will delete well in brain macrophage subtypes (microglia and different border associated populations), interstitial lung macrophages, cardiac macrophages, and capsular liver macrophages. All of these need to be assessed. The manuscript is about resident tissue macrophages, a more concerted effort to include a large number of tissues is required to make claims about RTMs. Currently, a relatively small subset of tissues is tested, and it is inconsistently tested across experiments.

In addition to kidney RTM, we have now characterized cardiac RTM and microglia in our CX₃CR₁-ERT₂Cre model by percentage and absolute numbers per mg/tissue. These data are in Fig. 2j-o. We also assessed lung interstitial macrophages and capsular liver macrophages and at 8 weeks post-tamoxifen, and observe a significant decrease in the number and frequency of these cells in the DHPS KO mice (Reviewer only Fig. 2 and 3).

1.6. The link between cell adhesion, which is mentioned numerous times in multiple pieces of data, and DHPS and ultimately macrophage differentiation is not clear, thus a mechanism is not clearly described. The authors state the challenge of performing gain-of-function experiments in this system due to translational block in DHPS-deficient macrophages. Overall the mechanism of how adhesion leads to differentiation isn't clear though, which is the focus of the work.

We appreciate this point, and in this section we have done our best to clarify the mechanism of how DHPS deficiency impedes RTM differentiation, which is the focus of our study, and have employed a number of orthogonal approaches to do so.

Transcriptional and Translational analysis of cell adhesion

Since eIF5A is a translation factor that helps translate 'hard to translate' mRNAs that pause the ribosome (ribosome pausing can lead to transcript degradation), we performed sequencing of ribosome engaged transcripts, analyzing in such a way that we would discern those that differ between WT and DHPS KO peritoneal macrophages, but not because they are transcribed to different levels in each. In other words, we asked which mRNAs are robustly expressed in both genotypes, but are specifically missing from DHPS KO ribosomes. Analyzing the data in this way, we found 13 genes that are specifically lost on ribosomes in DHPS-deficient peritoneal macrophages, suggesting that these genes are highly depending on hypusinated-eIF5A for their translation. Importantly, these genes are associated with cell adhesion, as well as signaling and apoptosis.

Text from our new submission regarding Riboseq analysis of Dhps^{-/-} RTMs:

"This analysis revealed 13 genes that were significantly reduced on ribosomes in DHPS-deficient peritoneal macrophages (Fig. 4a). We investigated the ascribed functions for each gene and found that most were involved in cell adhesion/interaction, signaling, and apoptosis, such as Icos⁴⁴, Cd28⁴⁵, Axin2⁴⁶, Tnik^{47,48}, Amigo2⁴⁹, Fam83g⁵⁰, Il1rl1⁵¹⁻⁵³, Rab44^{54,55}, and Oasl1⁵⁶. These data suggest that the proteins encoded by these genes are involved in promoting RTM differentiation, but that expression of these proteins is lacking in peritoneal DHPS-deficient macrophages."

We reasoned that the processes identified in our orthogonal approaches to be most affected by the loss of DHPS (i.e. cell adhesion/interaction, signaling, and apoptosis) would be vitally important for RTM differentiation. Although it is possible that one or more of these processes underwrite the requirement for DHPS in RTMs, we honed in on ST2 for 'signaling' (Fig. 4b, c), as this has been shown before to be important in macrophage differentiation (e.g. Lu Y. et al, 2020 May 19;52(5):782-793.e5; Dagher R. et al Nat Comm 2020 Sep 22;11(1):4786; Faas M. et al Immunity 2021 Nov 9;54(11):2531-2546.e5; Yang Y, J Neurosci 2017 May 3;37(18):4692-4704). We think it likely that these identified factors could promote cell adhesion directly through an unspecified mechanism (for proteins like CD28 or ICOS), or lead to the expression of other genes that promote cell interactions – for example, TIMd4. We feel that defining exactly how these 13 proteins may contribute to remodeling the transcriptome to promote cell adhesion during macrophage differentiation is a large undertaking beyond the scope of this study.

However, in exploring this in more detail, we further reasoned that if the modules identified (cell adhesion/signaling etc) underscored the defects in RTM differentiation, perturbations in these pathways would be seen in Dhps^{-/-} RTMs across tissues. To explore this idea, and to assess what was happening in these cells more broadly, transcriptionally, we performed bulk RNA-Seq on macrophages from 3 tissues. From our new submission about bulk-RNAseq:

"We performed bulk RNA-sequencing on F4/80+ macrophages (gating strategies in Extended Data Fig. 12a-c) isolated from 3 distinct tissues in Dhps WT and Dhps Δ M mice: the lung, liver, and the peritoneal cavity (Fig. 4e, 4f). Pathway analysis of DEGs that were common amongst all DHPS-deficient macrophages from each tissue revealed cell adhesion, signaling, and migration pathways as the most significantly downregulated (Fig. 4g and Supplementary Table 5), with inflammatory response and apoptotic pathways increased (Extended Data Fig. 12d and Supplementary Table 6). Notably, our sequencing of ribosome transcripts revealed several genes implicated in cell adhesion (Fig. 4a). For example, Tnik is critical for augmenting expression of target genes in the Wnt/ β -catenin pathway that controls cell adhesion^{47,48}, and as such, β -catenin-deficient macrophages lack cell adherence capacity⁶⁰."

To this point, pathway analyses of our proteomics and bulk RNAseq data highlight cell adhesion and signaling pathways as perturbed in DHPS KO RTMs. Further, our Ribo-Seq and scRNA-seq data show genes involved in cell adhesion and signaling as downregulated. We have now included volcano plots and pathway analysis for all 11 clusters and what differences are found between DHPS WT vs KO (Extended Data Fig. 9 and 10). Indeed, we found fewer TIMD4⁺ cells in the KO by scRNA-Seq. Importantly, when directly comparing TIMD4⁺ cells in WT and DHPS KO, we observe significant transcriptional differences. For example, in cluster 3, the expression of *Lyz2*, *CD63*, *Tmem17*, *Wfdc17* are among the most significantly decreased, supporting that DHPS KO macrophages are perturbed for genes related to macrophage identity, cell adhesion, and signaling, respectively. In cluster 4 (as well as cluster 2) we noted significantly reduced expression of *Wnt2* and *Tgfb2*, important molecules for cell-cell adhesion and matrix interaction. GO Pathway analysis for clusters 3 and 4 also reveal translation, cell division, and chromatin remodeling as downregulated processes in DHPS KO, which might align

with the fact that DHPS regulates translation via eIF5A, and that homeostatic renewal of RTM is critical to their survival and differentiation. Together, these data indicate that these processes are important for RTM differentiation.

Analysis of cell adhesion molecules on protein level in Dhps KO RTM

In our new submission, we further show that cell adhesion molecules are not robustly expressed at the protein level in Dhps KO:

"The genes encoding for the cell adhesion molecules L1CAM (l1cam) and E-Cadherin (cdh1) were downregulated in the pathway analysis of DHPS-deficient macrophages isolated from 3 distinct tissues (Table 5), which correlated with decreased protein expression (Fig. 4h), and with earlier work showing that E-Cadherin expression is polyamine dependent in alternatively activated macrophages."

Defects in cell morphology and cell-to-cell interactions by imaging

In Fig. 4i-n and Extended Data Fig. 13 we show data depicting defects in cell morphology in Dhps KO macrophages and their perturbed interaction with the stroma across lung, kidney, and brain.

Functional assessments of cell adhesion

Importantly, we present functional data showing significant defects in adherence in Dhps-deficient macrophages:

"To investigate cell adhesion functionally, we cultured peritoneal macrophages for 24 hours in the presence of CSF-1 to allow adherence. Cell adhesion and integrin binding are Ca²⁺ dependent and we quantified the sensitivity of cultured cells to detach in the presence of a calcium chelator (EDTA) over time. DHPS-deficient macrophages were significantly less adherent, with ~50% of the cells detaching after 2 minutes in EDTA, compared to <10% of control cells, which took 15 minutes for 50% detachment (Fig. 4i)."

Overall, we believe our manuscript contributes two significant findings – that DHPS is a universal regulator of RTM differentiation and tissue residency, and that by investigating the mechanistic basis for this discovery we reveal that cell adherence and interaction with the tissue is critical for the ability of macrophages to take up occupancy in the tissues. We acknowledge that deciphering how the physical interaction of developing RTMs with cells of the tissue regulates their development is key to understand, but would require a substantive body of work and additional mouse models that we feel goes beyond the scope of the findings presented here. It is worth noting that the idea that cell adhesion of monocyte precursors entering tissue has been discussed before. For example, a paper from Manfred Kopf's lab (Okreglicka et al, 2021, J Ex Med) indicates that monocytes that enter the spleen to differentiate into red pulp macrophages undergo a transcriptional change that includes increased VCAM-1 and integrin alphaD expression. They speculate here that this may be required for progenitor retention in the tissue because it allows access to niche-related signals that permit RTM differentiation. Understanding how this works is the focus of our future work.

1.7. *The cell sorting and RNA-seq experiments across tissues is an important study (between control and KO animals). However, there are different macrophage subtypes in those tissues (ie those that express TIMD4 and those that do not, but express other markers), etc. However, in these sorting experiment, all the MFs were pooled - and thus the different subtype information is lost. These MF subset differentially rely on monocyte input, so pooling them together may confound observations. This story has some similarity to*

*the deletion of *cmaf* and *maf*b in interstitial lung macrophages, where monocytes gain entry. Sorting all populations together (monocyte-derived cells) and more differentiated macrophage may be a large issue. It would be critical to show the gates that were sorted (monocytes, monocyte-derived MFs) and more resident populations (Even if they are transient / different). In that way different subsets can be preserved. A large single cell study could also help resolve this issue - but this would require replicates and detailed analysis.*

This is a fair point. Our rationale for sorting total macrophages was that *Dhps*^{-/-} RTM often lack markers of tissue-residency or time-in-the-tissue (e.g. TIM4) so this makes sorting comparable populations challenging. Furthermore, given that our parabiosis studies suggest that the tissues of DHPS LysmCre mice are likely a mix of both monocyte-derived and embryonically derived macrophages (although probably dominated by monocyte-derived cells, unlike in WT mice at a young age), we thought the fairest comparison we could make for this particular experiment would be the sorting of total macrophages from these tissues. Unlike the scSeq experiment in Fig. 3 using peritoneal macrophages, here we hoped to gain some insight into common programs across several tissues. The previous submission was lacking the gating strategy flow plots, which is critically important to show (so the reader is clear that we are comparing heterogeneous populations in terms of macrophage subtypes) and we apologize for this oversight. New plots in Extended Data Fig. 12a-c show how we gated out lymphocytes, monocytes, and neutrophils to only sort macrophages.

1.8. *Lastly, I do not really understand how the analyses were done here? All the tissues were pooled? which DEGs are changed in each tissue? This experiment and associated figure are difficult to interpret*

We realize that our schematic figure conveyed the idea that the tissues were pooled, but they were not and we have now fixed the figure (Fig. 4e). In order to study what pathways were downregulated across tissues by the deletion of DHPS, we compared DHPS KO lung, peritoneum, and liver vs DHPS WT F4/80+ macrophages. We sequenced bulk RNAs from each of these populations separately. Then, to analyze, we used a stringent filter to only look at genes that were highly down or upregulated, speculating that this would minimize tissue specific changes in transcription, and allow focus on what cell intrinsic programs were impacted in these 3 macrophage populations across tissues. The raw data from all the -omics experiments have been deposited and access tokens can be provided to the reviewers.

1.9. *Additional questions about Figure 4: scRNA-seq data presented are not sufficient as it stands and require deeper analyses to be of value to readers. There are 11 clusters shown in Figure 4a but it is never stated what these clusters are and what they express. (A) Are all clusters macrophages or are there other immune cells or monocytes included too? Please include canonical macrophage markers. (B) There should be more unbiased representation of the data with appropriate visualization of DEGs in each subset (heatmaps). The authors introduce this section as an opportunity to assess RTM in an unbiased manner, but this is not fully taken advantage of.*

We thank the reviewer for pointing out this problem, and have now performed a deeper analysis of our scRNAseq data. Even though the scRNA-seq was done from total peritoneal exudates, we excluded non-macrophage cell types and the analysis was done only in macrophages. The complete analysis can be found in our scRNAseq methods section: "*Datasets were subsetted to include only macrophages, based on the expression of key macrophage markers (*Adgre1*, *Csf1r*, *H2-Ab1*, *Cd68*, *Lyz2*, *Itgam*, *Mertk*). Macrophage only datasets were then split along conditions, and processed as new as described above.*"

The top genes expressed in each of the 11 clusters in WT cells are now presented in a heatmap (Extended Data Fig. 8). Extended Data Fig. 9 shows volcano plots of the top up and down regulated genes, and Extended Data Fig. 10 shows the top up and down regulated GO pathways, for each cluster. Fig. 3d, e also show the pathway analysis of cluster 2.

The focus of this experiment was to investigate the monocyte-macrophage transition on a single cell level in Dhps KO cells. The scRNA-seq data identifies cluster 2 as an intermediate population between Ccr2-expressing monocyte-derived macrophages (cluster 1) and Timd4-expressing RTM (cluster 3, and, in part, 4). What is very clear in these data is that WT cells are reduced in clusters 1 and 2, and KO cells are reduced in cluster 3 and 4. For the main figure, we decided to focus on cluster 2, as we speculated that this could be the intermediate population that monocyte-derived macrophages would transition through on their way to becoming RTM. Comparing between WT and KO (Fig. 3d) now shows that the most downregulated pathway in cluster 2 is cell division (proliferation), while Fig. 3e shows that the most upregulated pathways are related to inflammatory responses. These results suggested to us that KO cells were 'blocked' from transitioning to a mature macrophage state, and instead were 'retained' in an immature form, unable to acquire the full local RTM signature that ultimately precludes their ability to take up long-term residency in the tissue. That DHPS KO cells remain in a more 'inflammatory state', as defined by pathway analysis, which we think is reflective of their recent monocytic origin, and in keeping with the idea that monocytes typically infiltrate tissues in response to injury or pathogenic threats.

(C) How do TIMD4+ RTM transcriptionally differ in WT vs DHPS-KO mice? Are the reduced RTM similar to WT just less in number, or do molecular differences exist (this could be in tissue resident macrophage genes but also other pathways)? Direct comparisons should be included, this is the type of data that cannot be acquired with flow cytometry easily but was not well-explored in the manuscript. If there are no differences, this is also important to report.

This is a good point and we have now included volcano plots and pathway analysis for all 11 clusters and what differences are found between DHPS WT vs KO (Extended Data Fig. 9 and 10). Indeed, we found fewer TIMD4+ cells in the KO by scRNA-Seq. Importantly, when directly comparing TIMD4+ cells in WT and DHPS KO, we observe significant transcriptional differences. For example, in cluster 3, the expression of *Lyzz*, *CD63*, *Tmem17*, *Wfdc17* are decreased, supporting that DHPS KO macrophages are perturbed for genes related to macrophage identity, cell adhesion, and signaling, respectively. In cluster 4 (as well as cluster 2) we noted reduced expression of *Wnt2* and *Tgfb2*, important molecules for cell-cell adhesion and matrix interaction. GO Pathway analysis for clusters 3 and 4 also reveal translation, cell division, and chromatin remodeling as downregulated processes in DHPS KO, which might align with the fact that DHPS regulates translation via eIF5A, and that homeostatic renewal of RTM is critical to their survival and differentiation.

1.10. *The altered morphology of tissue macrophages was only assessed in the lung alveolar compartment. As noted above, a multiple tissues need examined. For a pan RTM paper, making a single observation in a single tissue MF subset reduces enthusiasm for the study. Was surfactant itself measured?*

This is a valid point and we have added new data showing confocal images from kidney and brain macrophages in the CX3CR1 inducible deletion model (Fig. 4n and Extended Data Fig. 13d). These data show, as in lung, altered morphology of macrophages in both of these tissues after DHPS depletion.

Surfactants were not directly measured, although the OD method has been used as a standard to assess lung alveolar proteinosis in BAL by several publications, e.g. Wculek, SK et al. *Immunity* vol. 56,3 (2023): 516-530.e9. and Suzuki T, et al. *Nature* vol. 514,7523 (2014): 450-4. doi:10.1038/nature13807.

1.11. *Figure 2e: In this analysis, tissue resident macrophages were first gated prior to assess CD45.1 vs CD45.2. It's possible that there could be donor contribution to non TIMD4 macrophages to compensate for the reduced resident pool. The authors should report chimerism in other macrophage subsets as well.*

We acknowledge the possibility that BM progenitors may contribute to RTM in our bone marrow chimera experiment. To address this question, we have prepared Reviewer only Fig. 4 examining the lung from the same experiment. When we perform the analysis by looking at the bulk macrophage gate (F4/80⁺CD64⁺) instead of pre gating specifically on alveolar macrophages (ie CD11b^{low}CD11c^{hi}SIGLEC-F⁺), we find that a small percentage of F4/80⁺CD64⁺ macrophages are DHPS KO donor derived (CD45.2). Further more, the contribution of these cells to the total macrophage pool contribute almost exclusively to the CD11b⁺CD11c^{low} population rather than the CD11b^{low}CD11c^{hi} alveolar macrophage population.

Ultimately, our competitive BM transplant data supports the notion that infiltrating Dhps^{-/-} monocytes are significantly impaired in their ability to differentiate into a stable RTM population. We also explored this in an orthogonal model whereby monocyte recruitment is stimulated via liposome-mediated depletion of alveolar macrophages (Fig. 2e, f). Following alveolar macrophage depletion, Dhps LysmCre⁺ mice could successfully form CD11c⁺ macrophages, but their transition to mature CD11b^{low} Siglec-F⁺ alveolar macrophages was impaired. Furthermore, the presence of multiple waves of incoming monocytes in DHPS LysmCre⁺ mice, not observed in WT controls (Fig. 2g), also supports a failure of KO mice to adequately replenish and maintain the alveolar macrophage pool, something that is sensed by the bone marrow in a futile attempt to repopulate the lung niche.

1.12. *Figure 3c: Was there a difference in CSF1 in the tissue? CSF1R on macrophages may be the same but with less CSF1 present in the environment.*

We have not measured this directly, but we think our bone marrow chimera (Fig. 2c) and parabiosis experiments (Fig. 2d), where there is likely to be normal circulating and tissue CSF-1 levels, suggest a cell-intrinsic defect in RTM development that is independent of local CSF-1 levels. Similarly, our CX3R1-CreERT2 model allow us to delete DHPS in macrophages from a fully formed and healthy niche, supporting a cell-intrinsic mechanism rather than microenvironmental perturbations.

1.13. *If you inject clodronate liposomes and deplete tissue MFs, that would trigger an inflammatory response with monocyte entry. If you were to then infect or transfer malignant cells into that environment, likely neither would take hold. I understand the functional consequences of impaired MF differentiation leads to more monocytes, but its unclear how it is relevant to challenge such a model. I think more focus on RTM vs monocyte biology in this system, and more detailed mechanistic exploration is needed, rather than functional consequences in a model where homeostasis is so profoundly disturbed, in as a the authors state, a multi-factorial fashion.*

We agree entirely with these caveats and believe the major findings of our paper center around RTM and monocyte biology and the consequences of impaired RTM development on tissue homeostasis - in and of itself a major and vital function of RTMs. We have removed the cancer and infection models from our revised paper, as although the data were intriguing, we feel they complicated the central

message of the manuscript and is better as a focus of future work (also explained in more detail response to Reviewer #2 point 1).

In new experiments in lung, and peritoneum, we instead employ clodronate liposomes to further explore the monocyte-to-macrophage transition in naïve “steady state” tissues (Fig. 2e-g and Extended Data Fig. 6f, g). This process is challenging to observe in true physiological conditions as monocyte entry to the tissue is infrequent in healthy tissues. Use of clodronate liposomes to stimulate monocyte entry offers an opportunity to observe macrophage-differentiation in the tissues in the absence of infection or cancer. We do not believe the small amount of inflammation observed with liposomes diminishes the validity of our observations in these models, as in reality most monocyte to macrophage differentiation is likely to occur under some form of inflammation, even if relatively minor in magnitude, as monocytes typically enter tissues in response to damage and/or inflammatory stimuli.

Referee #2 (Remarks to the Author):

This study from Carrizo et al. seeks to implicate hypusination in the homeostatic process associated with differentiation of macrophages. In particular, data are presented that suggest that the DHPS/hypusinated ELF5A axis places a central role in generation of tissue resident macrophages. The implication of the work is that potentially, modulation/inhibition of hypusination could be used to enhance antimicrobial or antitumoral host defense since data are presented in which knockout of Dhps led to increase activation of monocyte-derived macrophages.

2.1. There is originality related to the concept of the role of hypusination in macrophage differentiation. However, there are other studies about macrophage function. In particular the study referenced as #55, Gobert et al., Cell Reports, 2020, is not accurately considered. Specifically the current authors state in the Discussion on p. 12 “Previous studies showed that DHPS-deficient macrophages are less prone to cause inflammation in adipose tissue 54, and in a mouse model of Helicobacter pylori 55.” However, Gobert et al. (ref 55) in fact showed exactly the opposite of what is written here in this sentence. In that prior report, the authors showed in mice with myeloid-specific deletion of Dhps (the same approach as in the current manuscript) there was actually increased (not less) gastric inflammation with H. pylori infection, and that this was associated with increased H. pylori bacterial burden (Fig. 5 in the Cell reports paper); moreover Fig. 6 in that same paper showed that for another gut infection, Citrobacter rodentium, a colonic pathogen, there was increased colonic inflammation and increased C. rodentium bacterial load in the colon. These findings were shown to be associated with loss of macrophage immune effector molecules in that study, which goes against the conclusion of the current manuscript that interference with hypusination would enhance host defense against pathogens.

We thank the reviewer for bringing this to our attention and apologize for miscommunicating and are grateful to have the opportunity to correct this. The first part of the sentence, i.e. “DHPS-deficient macrophages are less prone to cause inflammation in adipose tissue”, was meant to reflect the findings of Anderson-Baucum et al. that showed “DHPS deficiency in myeloid cells of obese mice suppressed M1 macrophage accumulation in adipose tissue and improved glucose tolerance” (Cell Metab. 2021 Sep 7;33(9):1883-1893.e7). In this study, proinflammatory M1 macrophages require DHPS expression. In the second part of the sentence, referring to Gobert et al., 2020, Cell Reports 33, 108510, we meant to convey that DHPS expression in macrophages was also needed for clearance of gut pathogens, and thus, DHPS-deficient macrophages would be *less inflammatory cell-intrinsically*. The sentence as written is misleading as the study found that *overall inflammation* was increased in Citrobacter or

Helicobacter infected mice with myeloid cell DHPS deletions. In our revised manuscript we have now re-written this section and adjusted the discussion of these reports:

"Other studies have investigated the role of DHPS-deficiency in myeloid cells, observing that in obese mice DHPS-deletion suppressed inflammatory macrophage accumulation in adipose tissue and improved glucose tolerance¹⁵ and that myeloid cell DHPS expression is required to clear gastrointestinal pathogens by controlling the translation of antimicrobial factors".

Regarding the role of DHPS in macrophage function, it was suggested by Reviewer 1 that rather than investigating the functional impact of DHPS deletion in infection or cancer models, we should focus more on understanding the mechanisms that govern RTM vs monocyte differentiation. We agree with this, and in conjunction with the comments of Reviewer 2, we removed infection and cancer models from the paper as we feel that our conclusions in disease settings were premature and should be thoroughly investigated as a focus of future studies. For example, it may be that mechanisms of clearance for lung (when we showed Strep pneumo in our first version) vs gut pathogens are very different, and that is why myeloid deletion of DHPS appears to have different effects in these models. Also, while we have now characterized a multitude of macrophage subsets in this revised manuscript, we have not investigated deletions of DHPS in gut macrophages. Gut RTM are made up of mostly monocyte-derived macrophages, compared to other tissues (Park MD Cell et al Cell 2022 Nov 10;185(23):4259-4279), and this could also explain differences in their function in the lung vs gut. Ultimately, we conclude that we have much more work to do in models of disease to understand how RTM engage in these processes and, further, how DHPS-deficiency influences them. For now, these studies are outside the scope of our revised manuscript, which focuses solely on important homeostatic functions of RTM and how they are disturbed in the absence of DHPS, and present data in this regard that we believe are robustly supported by multiple approaches and models.

2.2. *In terms of statistics, the methods overall seem appropriate, but there also seem to be a substantial number of experiments performed in small numbers of mice. Not all of the bar graphs tell us how many animals were used.*

Data in our revised manuscript has been re-formatted to clearly show how many mice per group were used, and experiments were repeated with more mice per group (such as Fig. 5i). Please also see the comments to Reviewer 1 point 1.1 and Reviewer 3 point 3.4 addressing this issue.

2.3. *In terms of conclusions there are additional concerns. There is no data availability section and the actual list of transcripts in the RNAseq and proteins in the proteomics is not provided, a major omission.*

This is a helpful point and we have added gene and protein lists in Supplementary Information (Tables 1-6) as well as pathway analyses with DEGs in the Extended data (scRNAseq volcano plots in Extended Data Fig. 9; Cluster 2 DEG genes in each up or downregulated pathway in Table 1 and 2; significantly altered proteins in Tables 3 and 4; DEGs in up and downregulated pathways from bulkRNAseq in Tables 5 and 6). In addition, we have deposited all sequencing and proteomics data with accession codes available for the reviewers upon request.

2.4. *Other specific comments: Fig. 1f thought significant, some of the differences are fairly small and the mouse numbers are low. Are these siblings, what sex, etc. Fig. 2g, can the authors provide more information about the monocyte differentiation, such as with the "waterfall" assay?*

We have now added as Reviewer only Fig. 5a, b, the quantification of subpopulations of circulating monocytes in DHPS LysM Cre mice WT vs KO. We observed that in circulation, there are no differences in the frequency of CX₃CR₁+Ly6Chi vs CX₃CR₁+Ly6Clow monocytes between genotypes. In our new submission, the liposome depletion experiments either in peritoneum (Extended Data Fig. 6f-g) or lung (Fig. 2e-g) show clearly that there are no significant differences in the frequency of the first wave of incoming Ly6C⁺ monocytes into the affected tissue, but there are differences in their transition to RTM.

In support of this, in Reviewer only Fig. 5c-e, we show that isolated Dhps^{-/-} monocytes exhibit defects in their in vitro differentiation towards macrophages with reduced F₄/80 expression in response to CSF-1.

2.5. *Fig 3d, CX₃CR₁ is not specific to kidney.*

We have now investigated the effects of DHPS deletion in other CX₃CR₁-expressing TRM populations, such as microglia and cardiac macrophages. These new data (Fig. 2n, o), together with those from the kidney (Fig. 2k, l), also reveal a significant reduction in frequency and number of these RTM populations

2.6. *Fig. 4 speculates about a pro-inflammatory phenotype, but this should be validated. Fig 4e, f, g has pathways but the actual proteins lists should be in the extended data but are not. How many proteins are up or down, Venn diagrams? Same question for the RiboTag proteins.*

This is an important point and the list of genes and proteins from all our omics experiments have now been added in Supplementary Tables 1-6.

Fig. 3h of the revised manuscript now shows the upregulated pathways from our proteomics data by DAVID pathway enrichment analysis, and Table S₄ shows the protein list that generated that pathway analysis, which collectively, we term 'inflammatory', due to the pathways identified: i.e. "Inflammatory response, positive regulation of T cell activation, positive regulation of monocyte chemotaxis, positive regulation of T cell migration, cellular response to cytokine stimulus, cellular response to IFNbeta, cellular response to IFNgamma". It is worth noting that inflammation is a very broad term, and proteins identified here as being upregulated in DHPS KO macrophages are not necessarily at odds with those as listed as downregulated in DHPS KO cells in Anderson-Baucum et al. or Gobert et al. studies. We agree with comments from the reviewers that the impactful part of our manuscript is the role of DHPS in macrophage development and tissue-residency. The role of hypusination in immune responses to infection dilutes this message, which is why in this revised manuscript we have removed these aspects.

2.7. *Fig 5a, if possible data should be confirmed at the protein level to prevent over-interpretation.*

Since IL33R (*Il1rl1*), a.k.a. ST₂, has been shown to be implicated in the differentiation of tissue macrophages in many studies (e.g. Lu Y. et al, 2020 2020 May 19;52(5):782-793.e5; Dagher R. et al Nat Comm 2020 Sep 22;11(1):4786; Faas M. et al Immunity 2021 Nov 9;54(11):2531-2546.e5; Yang Y, J Neurosci 2017 May 3;37(18):4692-4704), we were intrigued when our Ribo-seq results indicated that ST₂ is hypusine dependent in peritoneal macrophages and focused here for our validation studies. Importantly, we confirmed that ST₂ protein expression was decreased in DHPS KO peritoneal macrophages (Fig. 4b), and in microglia and alveolar macrophages (Reviewer only Fig. 6). Interestingly, as WT monocytes infiltrate the peritoneal cavity and differentiate into peritoneal macrophages (pMacs), ST₂ expression increases (Reviewer only Fig. 6a). Since it has been shown that CD28 is expressed on macrophages with anti-inflammatory potential (Estrado-Capetillo L et al. Eur J

Immunol 2021 Apr;51(4):824-834), we also assessed CD28 by flow cytometry. DHPS KO peritoneal macrophages express less CD28 than controls (Reviewer only Fig. 6e).

2.8. *Fig. 5 in vitro data about adherence; how can this be validated in vivo?*

This is a good question, although challenging to directly test in vivo. We think our data support the importance of perturbations in adherence for a number of reasons. Given that we see changes in shape, size, tissue placement, and volume of Dhps^{-/-} RTMs in vivo (Fig. 4j-n and Extended Data Fig. 13d, e of the revised manuscript), we think it is likely this would impair cell-to-cell interactions and adherence within the tissue. In support of this, 'cell adhesion' was also one of the top downregulated modules in both our RNA sequencing and proteomics data. We have further validated this by showing that the expression of adhesion proteins (Fig. 4h) are significantly decreased, alongside our in vitro adherence assays (Fig. 4i). Together, we hypothesize that these disruptions in adherence machinery would drive perturbations in local interactions with the tissue niche and will ultimately dictate RTM differentiation and/or survival.

2.9. *Fig. 6g: the severity of liver pathology is hard to follow as to how this is all due to a monocyte effect. It is as if these mice have right-sided congestive heart failure because this is what happens in patients with this situation. Or simply some type of acute liver injury like with drug induced necrosis. Perhaps there is more going on in these mice that needs to be worked out.*

This is an excellent point, and again highlights that our understanding of how RTMs govern tissue homeostasis is incomplete. The observation that clodronate disrupts tissue homeostasis in the liver of Dhps LysMCre mice was a serendipitous one. In WT mice, when liver macrophages are depleted with clodronate liposomes, the liver remains healthy with no histological disturbances in the tissue. We were surprised to see that when this was done in Dhps LysMCre mice, we observed severe liver pathology. These data strongly suggest that in WT mice, monocyte-derived macrophages entering the tissues post-clodronate are sufficient to maintain tissue homeostasis. Although we observe monocyte entry into the liver post-clodronate in Dhps LysM-Cre mice (and the formation of F4/80+ cells), they are unable to perform the critical homeostatic function observed in WT mice, suggesting that DHPS is essential for RTMs to maintain tissue integrity. Although this is not a canonical model of liver damage, we think it is revealing and an important observation supporting the notion that DHPS-deficient RTMs are unable to perform critical homeostatic functions.

Additionally, we think it is interesting that the reviewer brings up the point of right-sided congestive heart failure and liver damage, which is something we were not familiar with. After reading about this, it appears that patients with right-sided heart failure can have backflow of blood from the heart to the liver, and this increased pressure can lead to liver damage. Our results might suggest one of two things: 1) that liver damage and congestion due to macrophage DHPS deletion in our model could lead to problems in the heart, as it seems that increased portal hypertension can lead to right-sided heart failure, or 2) when we peritoneally inject clodronate liposomes we also deplete DHPS in cardiac macrophages and this leads to right-sided heart failure, causing backflow of blood from the heart and leading to liver damage, or 3) both of these things. We have yet to investigate these possibilities, but we are currently collaborating with our colleagues in the lab of David Kass (a cardiologist at Johns Hopkins) to assess what exactly happens to the heart after deletion of DHPS in macrophages as part of separate studies looking into disease processes.

Reviewer #3 also noted that the observed phenotype might not be liver-specific. After considering all of the comments, we amended the text as follows to better reflect why we did this experiment and our interpretation of the data:

“A key function of RTM is to maintain organ homeostasis. We observed significant immune cell infiltration in the lungs of Dhps-ΔM mice (Fig. 5d, e), overtly indicating disrupted tissue homeostasis in the absence of myeloid DHPS expression and defective alveolar macrophages in these mice. To more acutely investigate the function of RTM in maintaining tissue homeostasis in another tissue, and to demonstrate the fact that myeloid expression of DHPS is required for this critical RTM function, we intraperitoneally administered CL in Dhps WT and Dhps ΔM mice to broadly deplete macrophages in the peritoneal cavity and in other organs located in that space (Fig. 5h). We focused on the liver, as it has been shown that upon RTM depletion, bone-marrow derived monocytes enter the tissue and differentiate into monocyte-derived macrophages, and over time into RTM, thereby replenishing the liver RTM niche²¹. In both Dhps WT and Dhps ΔM mice 5 days after CL administration F4/80+CD64+TIM4+ macrophages were depleted, and monocyte-derived macrophages were detected in the tissue by their increased expression of CD11b within the F4/80+CD64+ gate (Fig. 5i). In control mice, liver sections stained with H&E and Masson Trichrome (MAS) appeared identical before and 6 days after CL administration, while the livers of Dhps-ΔM mice exhibited significant congestion (Fig. 5j and Extended Data Fig. 14a), abnormal sinusoidal structures, with the endothelial lining detached from liver cell plates, central veins lifted from the parenchyma (Extended Data Fig. 14b, c), and extensive necrosis (Extended Data Fig. 14d). Although the damage to the liver here may not be specific to the depletion of liver RTM, but rather to a broad depletion of RTM across all organs in the peritoneal cavity, which would result in severely disrupted homeostasis, the data do illustrate two important concepts: 1) the bone-marrow derived monocytes that enter tissues in control mice quickly transition to macrophages with the functional ability to maintain tissue homeostasis after RTM depletion, and 2) DHPS expression in macrophages is critical for restoring tissue homeostasis after perturbation, highlighting its importance in RTM differentiation, function, and maintenance.”

2.10. *Fig. 6K is at odds with ref 55 as stated above. It may be that Gram-positive organisms are very different (as studied here) compared to Gram-negative organisms, as in Ref 55. What were the proteins affected by Dhps deletion in lung macrophages? How was there gain of function if less macrophages and less translation? Fig 6N, this finding may not be true for solid tumors. This should be studied or at least acknowledged to prevent over-statement of the data.*

While these models are no longer included in this revised version, we want to provide some idea of what we were thinking here. We see better protection in the Strep pneumo model (previous version Fig. 6k), with reduced bacterial burden in the LysMcre-DHPS KO mice. We think this might be a consequence of enhanced monocyte-derived macrophage infiltration and more immune cell infiltration (such as T and B cells and granulocytes). We think that even in steady state, DHPS KO monocyte-derived macrophages create a proinflammatory environment that leads to the recruitment of other immune cells, that in the lung, led to more protection from Strep pneumo. For these reasons, it is unclear to us whether on a per cell basis Dhps^{-/-} RTM are more or less protective, and we suspect these observations are the result of a complex interplay of multiple cell types and alterations with the lung niche. For these reasons, it is very possible one might observe different effects depending on the tissue infected. Different RTM dynamics within each tissue might also be important in dictating response. For

instance, in the gut, embryonic derived RTMs are replaced early after birth with monocyte-derived cells, whereas in the lung this takes place slowly over the lifetime of the organism. Therefore, each tissue may be differentially affected in DHPS LysM-Cre mice in its response to pathogenic challenge. Furthermore, one might envisage that antibodies are important for controlling *Citrobacter*, while in the lung, perhaps neutrophils are more important for Strep clearance. As the reviewer indicates, our conclusions for infection and tumors are premature, and they are removed from the revised paper.

2.11. *In terms of the conclusions, one major issue is that this study has no human data. Can the authors show that hypusination is required for tissue resident macrophage differentiation in humans?*

We agree that understanding what role hypusination plays in human RTMs is intriguing, although we feel this is exceptionally difficult to study given that these cells only truly develop and exist in vivo and examining these processes in vivo in humans is largely beyond the capacity of the RTM field. While humans with DHPS mutations do exist, fewer than 10 people have been identified as of 2022 and most are children. In this small cohort of humans with mutations in DHPS, they present with Neurodevelopmental / Cognitive Delays , Epilepsy / Seizure Disorder (with no other underlying genetic cause), Speech and Language Delays , Neuromuscular Delays: Hypermobility, Hypertonia, Spasticity, Growth issues / Failure to Thrive , Immunological problems (associated with low IgA and IgG), and Congenital hip dislocation (<https://www.dhpsfoundation.org/>). RTM numbers in the tissues have not been studied in these patients. Our future studies in the brain will investigate how microglial deficiency in DHPS influences neurodevelopmental and cognitive function in mice, and depending on our results, could hint at the idea that the phenotypes evident in humans with DHPS mutations may be due to, at least in part, its lack of function in microglia.

2.12. *Also there are multiple studies showing that spermidine is beneficial in humans; how do the authors relate that to their conclusion that hypusination, which derives from spermidine, is a target to be blocked?*

We see this is a potential route to target healthy aging and maintain effective tissue homeostatic in advanced age. During aging, there is an attrition of RTMs from the tissues (this is particularly evident in lung and liver), which is suspected to drive altered tissue homeostasis. Hypusination, like polyamines, has also been shown to decrease with age in immune cells and we suspect that falling polyamine and hypusine levels could be responsible for the attrition of RTMs in aging. Given these observations and ideas, and the data we present here in this manuscript, we think that administration of spermidine could be a way to promote RTM levels during aging, thereby promoting healthy aging and protective tissue homeostasis.

Referee #3 (Remarks to the Author):

Carrizo et al. investigate the role of DHPS in the differentiation and maintenance of tissue-resident macrophages. Using various mouse models, the authors explore how DHPS-mediated hypusination of eIF5A, a key translation factor, influences macrophage development. They find that mice with myeloid-specific deletion of DHPS (LysM-Cre Dhps-KO mice) exhibit abnormal expression of tissue-resident macrophage markers such as Tim4 and a decline in macrophage populations with age. Parabiosis experiments reveal that this decline creates an open tissue niche, leading to persistent monocyte influx. Further analysis using scRNA seq shows that DHPS-deficient peritoneal macrophages are blocked in their transition from monocytes to mature RTMs. Proteome analysis reveals a decrease in the expression of cell adhesion molecules. Functionally, DHPS-deficient macrophages display impaired efferocytosis and

increased lung proteinosis at steady state. However, the authors show that these KO macrophages exhibit enhanced inflammatory functions, resulting in improved protective immunity in models of cancer and infection.

Together, the study characterizes a pathway mediated by the polyamine-hypusine axis, which seems essential for macrophage maintenance, differentiation and function.

3.1 *These findings are interesting, but the manuscript lacks originality in terms of unambiguously defining a novel function for a specific gene that is essential for macrophage biology. It may well be that the inhibition of a key translation factor simply changes everything in a cell, which will impact all its functions.*

We agree that interfering with a translation factor has the potential to have multifactorial effects. However, our Ribo-Seq data clearly demonstrate that deletion of DHPS perturbs the translation of a narrow subset of genes in macrophages, and these genes are associated with specific cellular programs (e.g cell adhesion and signaling). The novelty of this finding is also centered on identifying a factor that regulates RTM development and tissue occupancy across tissues, suggesting that this is a program essential for systemic population of the tissues by macrophages. Very few other factors have been identified to regulate macrophage biology in this way, acting as a universal regulator independent of the tissue of occupancy. We also report that DHPS specifically regulates RTMs, it has limited or no effects on circulating monocytes or their production in the bone marrow, or their ability to be recruited to tissues. These data suggest that DHPS acts to specifically govern the monocyte to macrophage transition, and we present evidence here that this may occur through specific perturbation of cell signaling and adherence programs that likely affect important interactions with the tissue upon entry.

Major points:

3.2. *It is well known that LysM-Cre is not a good targeting strategy for some macrophage populations. The authors show themselves that the spleen, and especially the brain (microglia are not targeted at all by LysM-Cre, this probably represents only autofluorescence). However, they see clear changes in the analyzed populations. This is hard to explain. Do the authors have an idea what exactly is happening? All tissue macrophages should also be quantified via immunofluorescence and via flow showing cells per gram of tissue and not %.*

We agree, it is true that LysM-Cre is not effective in all macrophage populations. For this reason, in our revised manuscript, we have crossed in the reporter YFP to our DHPS LysM Cre mice so that we can specifically identify Cre-targeted cells in each of our experiments (Fig. 1 and Extended Data Fig. 2, 3, 4 and 5). Using this reporter system, we find that LysM efficiently targets alveolar, peritoneal, and liver populations, with reduced targeting in kidney, heart, and brain (Extended Data Fig. 5). In our revised manuscript, we use CX₃CR₁-CreERT₂ to target RTM populations not well excised by LysM-Cre, chiefly microglia and kidney macrophages. With this model, in new data, we show that DHPS is critical for RTM maintenance across multiple tissues (Fig. 20), supporting our observations in LysM-Cre mice (Extended Data Fig. 4a). We have now added the YFP expression in microglia and lung in LysM-Cre positive and negative mice as Reviewer only Fig. 7 to show how this YFP expression represents a minor population of microglia targeted by this Cre and doesn't represent autofluorescence.

We thank the reviewer for the comment about quantifying absolute numbers per mg of tissue. This has been done for all tissues in our revised manuscript (Fig. 1f, Fig. 2l, 2n, 2o and Extended Data Fig. 2, 3

and 4)

3.3. *Since LysM-Cre is also active in monocytes, how do the authors explain the decreased expression of YFP in KO animals (Fig. 1f)? This is contradictory to the hypothesis that monocytes replenish the empty niche.*

We apologize for the confusion here. The original Fig. 1f showed the percentage of YFP+ cells in the total macrophage gate, therefore monocytes were already excluded. The % of YFP was meant to convey the proportion of the total of macrophages in each tissue that express YFP. We interpret a decreased frequency of YFP+ RTM in the KO to mean reduced survival of DHPS KO cells compared to WT. Our revised manuscript now shows absolute numbers per gram of tissue of YFP+ macrophages, in addition to %. Employing our DHPS CX₃CR₁-ERT₂ model, we also demonstrate a gradual loss of YFP+ RTM over time in the KO, driven by a loss of RTM in targeted tissues. In Fig. 1 and Extended Data Figs. 2, 3 and 4.

3.4. *Experiments with n=1-2, especially showing p values, is not acceptable.*

All experiments now have increased *n*, with the exception of some groups within Fig. 2b (*n*=2) that is a longitudinal time course from 4 to 34 weeks. All other experiments were repeated with more mice per group and this is written into all Figure legends.

3.5. *The analysis of the Cx3cr1-CreERT models is somewhat distracting as it is looking at kidney, although before the focus was on the peritoneum, lung and liver. The manuscript would have profited from a holistic approach looking at all organs in all models to describe the differences of affects versus not-affected macrophage populations.*

This is a good point. In our revised manuscript we have utilized the excision of DHPS through two distinct Cres in order to assess the affects of hypusination across RTMs in many tissues targeted by these Cre recombinases. We have added new data looking at more populations known to express robust levels of CX₃CR₁ namely brain, kidney and heart and these data can be found in our revised manuscript in Fig. 2l (kidney), Fig. 2n (heart) and Fig. 2o (brain). The CX₃CR₁-CreERT model has the additional benefit of inducibly deleting DHPS in fully, normally formed, RTMs in adult mice.

3.6. *Extended Data Fig. 4b: the KO clearly has not only F4/80 macrophages, therefore any result for this RNA-seq analysis may stem from the change of cell composition from 100% F4/80+ cells to only 90%*

We understand what the reviewer is saying here, however a 10% change in composition is not likely to account for the significant change that we observe in, for example, ST₂ expression at the protein level (Fig. 4b and Reviewer only Fig. 6). Further, and perhaps more importantly, our analysis was done in a way where we only considered differences in mRNAs bound to ribosomes between WT and KO if they were transcriptionally robust and similar, meaning that if transcripts were high in the input for WT and absent in KO, then we did not consider these 'missing' on the ribosome. Our approach (Extended Data Fig. 11c) would even further minimize any potential differences resulting from a 10% difference in cell composition between WT and KO. To explain, a 10% change in the number of F₄/80+ cells would account for a 10% shift in transcript abundance. However, our cutoff is actually a 4-fold change (log₂-fold change >2) for transcripts associated with the ribosome and a 2-fold change for differences between WT and KO, which are both an order of magnitude greater than a 10% change.

Additionally, we decided to not sort the cells from the peritoneal cavity to avoid perturbing mRNA present in the cells, including those loaded on ribosomes. Given that we wanted to maintain the cells in a way most similar to how they exist in the tissue niche, we took the approach described instead. We performed this assay using peritoneal macrophages so that we wouldn't have to digest tissues to isolate macrophages, which we also feel could significantly impact mRNAs. We acknowledge that this is not a perfect approach. The other option we considered was using cultured bone-marrow derived macrophages, which would be uniform in composition, but also would not exhibit characteristics of true tissue residency making the validity of the data unclear.

3.7. Along these lines, the scRNA-seq shows that macrophages remain somewhat immature in KO animals. This means that for all RNAseq data, the authors should be careful regarding the DEG, especially if it comes to cell adhesion. It could be simply due to the immature state of monocyte-derived macrophages. An in-depth analysis of all clusters in the scRNA-seq may help, and if the cell adhesion is really affected, it would show up in the bona fide macrophage clusters.

We agree and felt that this was a key finding of our study: DHPS-KO macrophages remain in an immature state, lacking the ability to fully differentiate into tissue-resident macrophages (RTMs). Mechanistically, we attribute this to a reduced capacity to engage in cell adhesion, trafficking, signaling, and tissue interaction programs.

In cluster 2, which we consider a transition population, the few WT cells present express several genes involved in cell-cell interaction, migration, tissue remodeling, adhesion, and efferocytosis. These include *FOLR2*, which facilitates interactions of RTMs with T cells in the niche (Ramos RN et al Cell 2022), *CXCL1* and *CXCL2*, which act as chemoattractants and are implicated in fibrosis and tissue remodeling, *Lyve1*, a receptor for hyaluronan (expressed by the extracellular matrix) that regulates tissue homeostasis, and *Gas6*, a ligand for TAM receptors important for efferocytosis (Myers KV et al. Molecular Cancer 2019). Additional genes such as *Fxyd2*, involved in extracellular matrix pathways and cell adhesion (Jin M, et al Aging (Albany) 2021), *Cbr2*, expressed in resident macrophages (Willemsen L, et al J Pathol 2020), and *Ms4a7*, linked to tumor progression and fibrosis (Zhou L et al, Sci Transl Med 2024) further support the role of this cluster in tissue integration and interaction.

In the fully differentiated tissue-resident populations (clusters 3 and 4), WT macrophages express key markers associated with cell-cell interaction, efferocytosis, and adhesion. *TimD4*, a canonical RTM marker, binds integrin alphavbeta3 (Zhang Q, et al. Brit J Cancer 2015), while *Marco* is involved in efferocytosis through apoptotic cell binding. *Tgfb2* regulates the expression of cell adhesion molecules (CAMs) and modulates extracellular matrix interactions (Ignotz RL and Massagué J Cell 1987), while *Wnt2* signaling influences beta-catenin stabilization, which plays a crucial role in cell adhesion. Additionally, *Nt5e* (CD73), also known as lymphocyte-vascular adhesion protein-2, plays a role in cell adhesion and immune modulation (Airas L and Jalkanen S, Blood 1996).

Notably, we also observe expression of *Slc7a2*, which has been shown to transport polyamines such as putrescine and spermidine. This suggests a possible link between polyamine uptake and macrophage tissue residency, providing an interesting avenue for further investigation.

We think that it was our comparison of the WT to the DHPS KO that allowed us to focus on these clusters (as there were so few WT cells in clusters 1 and 2, we likely would not have focused on them if only assessing WT cells), as well as the programs that were enacted along the transition from

monocyte-derived macrophages to RTM. We hope that our paper will provide a foundation from which other laboratories can begin to explore on how the cell-to-cell adhesion and tissue interactions in the niche are important for monocyte-macrophage-RTM differentiation, as well as other cell types, across tissue sites.

If the reviewer was also asking about the DEGs when comparing WT and DHPS KO cells, volcano plots and pathway analysis of the individual clusters from our scRNA-seq analysis are now shown in Extended Data Fig. 9 and 10.

3.8. *The lung, with highly motile alveolar macrophages, seems not the best choice to show a lack of adherence. Here, rather the spleen and liver should be analysed. Could the lack of adherence be correlated with the reduced expression of ST2 in these tissues?*

This is a good point, and in our revised manuscript we also show the morphology of macrophages in the kidney and microglia using the CX₃CR₁-CreERT₂ inducible model to illustrate that the same morphological changes occur upon acute deletion of DHPS in fully formed (previously WT) cells, as they do in the LysM cre mice that have DHPS deleted during development. Please see Figs. 4k and 4m for lung, 4n for kidney and Extended Data Figs. 13d and 13e for brain.

We think that the reduced expression of ST₂, and the resulting loss of the signals it conveys, could, in part underlie the lack of adherence. The several papers outlined below show the critical role of ST₂ in the development and function of multiple macrophage populations. However, the reviewer does bring up a good point, and the lack of adherence capacity could precede expression of ST₂. To speculate, we wonder if the process of 'cell migration, adherence, tissue interaction, trafficking to tissues,' etc., all the pathways we highlight, induces some type of 'cell damage/perturbation', that then leads to the induction of IL-33/ST₂. This is something we plan on investigating in future studies.

- Lu, Yuning et al. "Interleukin-33 Signaling Controls the Development of Iron-Recycling Macrophages." *Immunity* vol. 52,5 (2020): 782-793.e5. doi:10.1016/j.immuni.2020.03.006;
- Dagher, Rania et al. "IL-33-ST₂ axis regulates myeloid cell differentiation and activation enabling effective club cell regeneration." *Nature communications* vol. 11,1 4786. 22 Sep. 2020, doi:10.1038/s41467-020-18466-w;
- Faas, Maria et al. "IL-33-induced metabolic reprogramming controls the differentiation of alternatively activated macrophages and the resolution of inflammation." *Immunity* vol. 54,11 (2021): 2531-2546.e5. doi:10.1016/j.immuni.2021.09.010;
- Yang, Yuanyuan et al. "ST₂/IL-33-Dependent Microglial Response Limits Acute Ischemic Brain Injury." *The Journal of neuroscience : the official journal of the Society for Neuroscience* vol. 37,18 (2017): 4692-4704. doi:10.1523/JNEUROSCI.3233-16.2017).

3.9. *It is highly unlikely that KO mice die due to the lack of KC and replenishment by Dhps after clodronate treatment as the normal KO without treatment has a normal liver morphology. Did the authors look at the gut? It may be that the gut barrier integrity is not functional any more, which could also have a secondary effect on the liver. The conclusion that the observed phenotype is liver-specific should be removed.*

We agree and thank the reviewer for pointing this out as it was not something we had considered. Possible alternative explanations were also suggested by Reviewer 2 (potential acute heart failure or liver failure). We agree with the reviewer that the pathology we observe in DHPS LysM cre mice in this experiment may not be driven intrinsically by Kupffer cells. But we can conclude it is driven by loss of DHPS in myeloid cells leading to altered functionality that ultimately perturbs liver homeostasis. It is quite possible that gut macrophages lose DHPS expression and as a consequence, gut integrity is lost. We have also expanded upon this in response to Reviewer 2 point 2.9, and we have amended the text in the manuscript to reflect this change as follows:

“A key function of RTM is to maintain organ homeostasis. We observed significant immune cell infiltration in the lungs of Dhps-ΔM mice (Fig. 5d, e), overtly indicating disrupted tissue homeostasis in the absence of myeloid DHPS expression and defective alveolar macrophages in these mice. To more acutely investigate the function of RTM in maintaining tissue homeostasis in another tissue, and to demonstrate the fact that myeloid expression of DHPS is required for this critical RTM function, we intraperitoneally administered CL in Dhps WT and Dhps ΔM mice to broadly deplete macrophages in the peritoneal cavity and in other organs located in that space (Fig. 5h). We focused on the liver, as it has been shown that upon RTM depletion, bone-marrow derived monocytes enter the tissue and differentiate into monocyte-derived macrophages, and over time into RTM, thereby replenishing the liver RTM niche²¹. In both Dhps WT and Dhps ΔM mice 5 days after CL administration F4/80+CD64+TIM4+ macrophages were depleted, and monocyte-derived macrophages were detected in the tissue by their increased expression of CD11b within the F4/80+CD64+ gate (Fig. 5i). In control mice, liver sections stained with H&E and Masson Trichrome (MAS) appeared identical before and 6 days after CL administration, while the livers of Dhps-ΔM mice exhibited significant congestion (Fig. 5j and Extended Data Fig. 14a), abnormal sinusoidal structures, with the endothelial lining detached from liver cell plates, central veins lifted from the parenchyma (Extended Data Fig. 14b, c), and extensive necrosis (Extended Data Fig. 14d). Although the damage to the liver here may not be specific to the depletion of liver RTM, but rather to a broad depletion of RTM across all organs in the peritoneal cavity, which would result in severely disrupted homeostasis, the data do illustrate two important concepts: 1) the bone-marrow derived monocytes that enter tissues in control mice quickly transition to macrophages with the functional ability to maintain tissue homeostasis after RTM depletion, and 2) DHPS expression in macrophages is critical for restoring tissue homeostasis after perturbation, highlighting its importance in RTM differentiation, function, and maintenance.”

3.10. *Is efferocytosis and phagocytosis in alveolar macrophages and other populations also affected? There are some good in vivo efferocytosis assays. As many macrophage populations are affected, they could be performed instead of looking only at the OD of BAL. Further, aged mice should be analyzed for progression to lung fibrosis or similar to make the point that Dhps is essential for macrophage functions.*

This is a good point and in response we have performed in vivo efferocytosis assays. In new Fig. 5 of our revised manuscript we show perturbed clearance of stressed RBCs in the blood in DHPS KO mice compared to controls (Fig. 5f) and in vitro (Fig. 5h). To support this, we also delivered stressed RBCs directly into the trachea of mice to test the ability of alveolar macrophages to efferocytose (Reviewer only Fig. 8a). These data reveal a significant defect in the frequency of Dhps^{-/-} alveolar macrophages to ingest sRBC in vivo (Reviewer only Fig. 8b, c).

We are currently assessing aged mice with macrophage specific deletions in DHPS, as well as in disease conditions, as part of future studies. Aged cohorts of mice require considerable resources that include significant mouse breeding space with major financial investment, and this is something we are trying to obtain funding for.

3.11. *The point of inflammatory capacity is somewhat weak and contradicts the reduced phagocytic capacity, especially in the Streptococcus model. Also, IL6 production was reduced upon stimulation.*

We agree and we now solely focus our manuscript on the monocyte to macrophage transition and the consequence of hypusine-driven perturbations in RTM formation to tissue homeostasis. Future studies will aim to further understand the intrinsic inflammatory programs of macrophages upon DHPS deletion and their potential relevance in disease contexts. We acknowledge that our conclusions in disease context were premature.

In the revised manuscript, we include Fig. 3h, which shows the upregulated pathways identified through DAVID pathway enrichment analysis using our proteomics data. Table 4 now lists the proteins contributing to these inflammatory pathways, which include responses related to cytokines, T cell activation, and monocyte chemotaxis. We see a similar finding when we performed pathway analysis of our bulk RNAseq data (Extended Data Fig. 12) of macrophages isolated from 3 different tissues in WT and DHPS KO mice. We use these data now to only to highlight that these pathways, which are defined by pathway analysis, are increased in the KO compared to WT.

The question of inflammation was also raised by Reviewer #2 and this is something we will dive more deeply into in disease contexts in future studies.

3.12. *The inoculation strategy with 1:1 mic of macrophages is elegant, but in this case has the caveat that KO macrophages are more prone to apoptosis as shown by the cleaved caspase assessment. Therefore, it could be simple cell death that affects tumor growth.*

We agree and we have removed this data as it will be a focus of future studies. We have much more work to do to understand how deletion of hypusine impacts anti-tumor immunity.

3.13. *The maturation process could be controlled via the altered metabolism of cells. Do they differ e.g. in their oxidative phosphorylation/glycolysis capacity?*

Yes, we do observe metabolic changes upon DHPS loss. These findings are consistent with our previous reports in bone marrow-derived macrophages, where we observed a shift between oxidative phosphorylation (OXPHOS) and glycolysis upon DHPS deletion (Puleston et al, Cell Metab 2020). Additionally, our omics data indicated a decrease in fatty acid metabolism in DHPS KO cells. Fatty acid metabolism is critical for cell persistence and longevity, a process conserved from worms and flies to mammals. These results are presented in Fig. 3g.

We have not included metabolomics or bioenergetics data in the paper, as we feel they do not significantly contribute to the main narrative. However, we have provided these data below as Reviewer only Fig. 9.

Since our bulk RNA-seq across tissues revealed that cell adhesion and signaling were the shared downregulated pathways in DHPS-deficient macrophages, we propose that the reduced persistence and incomplete development of full RTM programs in tissues may result from a combination of altered tissue signaling and could involve metabolic changes which is in line with previous reports.

3.14 *Minor points:*

1. *Fig. 1b-c: why are 4 samples shown per genotype in the WB but only 3 quantified?*

The quantification shown in 1c was done from blots that were located in Extended Data Fig. 1 in our previous manuscript.

2. *Gating strategies for representative WT and KO samples used for bulk RNA-seq should be shown*

Also something brought up by Reviewer 2 and we apologize for this oversight. We have now included the gating strategies for the bulk RNAseq (Fig. 4-g) in Extended Data Fig. 12a-c.

Reviewer only Figures

Reviewer only Fig. 1

[REDACTION]

[REDACTION]

[REDACTION]

[REDACTION]

[REDACTION]

[REDACTION]

[REDACTION]

Reviewer only Fig. 2

[REDACTION

[REDACTION]			
[REDACTION]			[REDACTION]
[REDACTION]	[REDACTION]	[REDACTION]	[REDACTION]
[REDACTION]			
[REDACTION]			
[REDACTION]			

[REDACTION]					
[REDACTION]					
[REDACTION]	[REDACTION]	[REDACTION]	[REDACTION]	[REDACTION]	[REDACTION]
[REDACTION]					
[REDACTION]					

Reviewer only Fig. 4

Reviewer only Fig. 4. DHPS-deficient monocytes fail to form mature alveolar macrophages (CD11b^{lo} CD11c^{hi}) in a competitive bone marrow transplant model, predominantly forming CD11b⁺CD11c^{low} populations. Bone marrow from CD45.1+ SJL mice was mixed 1:1 with either CD45.2+ *Dhps*-WT or CD45.2 *Dhps*-ΔM BM and transferred into lethally irradiated CD45.1+ SJL recipients. Chimerism in specific macrophage populations was assessed 12 weeks post BM transfer. This additional analysis shows that the lung macrophage pool (F4/80+CD64+) is constituted almost exclusively from WT progenitors. Furthermore, the small frequency of CD45.2+ *Dhps*^{-/-} macrophages observed in lung do not present as mature AMs, and are instead CD11b⁺ CD11c^{hi/lo}. Data are mean ± s.d. Exact P-values are indicated.

Reviewer only Fig. 5

Reviewer only Fig. 5. Normal abundance of circulating CX3CR1⁺ Ly6C^{hi} and Ly6C^{lo} monocytes in Dhps-ΔM mice, but Dhps^{-/-} monocytes exhibit decreased F4/80 expression when differentiated in vitro with CSF-1. a) Representative flow cytometry plot and quantification b) of CX3CR1⁺ Ly6C^{hi/lo} subpopulations within the CD11b⁺CSF1R⁺ gate in blood circulating monocytes from *Dhps-WT* and *Dhps-ΔM* mice. c-e) Monocytes were isolated from peripheral blood of *Dhps-WT* and *Dhps-ΔM* mice and cultured for 3 days in vitro with CSF-1. c,d) Expression of indicated protein on WT and *Dhps*^{-/-} cells pre- and post-differentiation. e) F4/80 and CD11b expression was assessed on WT and *Dhps*^{-/-} cells by flow cytometry on day 3 of culture. Data are mean ± s.d. Exact P-values are indicated.

Reviewer only Fig. 6

[REDACTED]

[REDACTED]

[REDACTED] [REDACTED]

[REDACTED]

[REDACTED]

[REDACTED]

[REDACTED]

[REDACTED]

[REDACTED] [REDACTED]

[REDACTED] [REDACTED] [REDACTED]

[REDACTED]

[REDACTED]

Reviewer only Fig. 7

a

b

Reviewer only Fig. 7. Bona fide YFP expression in microglia in LysM-Cre R26YFP reporter mice. YFP expression was compared between LysM-Cre⁻ and LysM-Cre⁺ Rosa26YFP reporter mice in brain and lung. YFP expression was detected only within the CD45⁺ gate and only in LysM-Cre⁺ positive mice and tracked to microglia in brain and subpopulations of macrophages in lung.

[REDACTION]

[REDACTED]

Referees' comments:

We would like to thank all the reviewers for taking the time to thoughtfully review our work and to provide insightful comments. We have carefully addressed each point, responding to each individually below. We believe that the new revisions made in response to the reviewers' feedback have significantly strengthened our manuscript, particularly by validating our initial single-cell sequencing data in an additional tissue. Our whole-tissue imaging across multiple organs supports our existing observations employing flow cytometry, that loss of Dhps results in a systemic loss of RTMs from the tissues.

Referee #1 (Remarks to the Author):

The authors have performed a substantial number of new experiments, in particular expanding to other tissue subpopulations beyond the peritoneum. The data indicate a generalized disruption of the macrophage differentiation when DHPS is targeted.

1.1 *One critical issue was the hypothesis that differentiation was blocked in the DHPS KO, and they show this most clearly in peritoneum with scRNA_seq and many other parallel approaches. The peritoneum is such a different compartment than the interstitium. It would solidify the conclusions and provide more generalizability if scRNA_seq was also performed in another tissue, such as the lung. I appreciate the authors provide flow cytometric evidence, but this is less convincing.*

This is an important point, and we have now performed scRNA-seq on digested lung tissue using the Rosa26YFP reporter strains in Dhps^{flox/flox} LysM-Cre⁺ and Dhps^{+/+} LysM-Cre⁺ control mice. We sorted total YFP⁺CD45⁺ cells (**new Extended Data Fig. 19** – gating strategy) and focused our analysis on macrophages.

Clustering revealed significant differences between DHPS KO and WT macrophages (**new Extended Data Fig. 20a, b**), with a loss of cluster 0 and enrichment of other clusters (e.g. cluster 1) in DHPS KO (**new Extended Data Fig. 20a** – see percentage). When examining genes expressed by alveolar macrophages (*Adgre1*, *Itgax*, and *Siglec1*), we found that cluster 0 is exclusively *Siglec1*⁺, therefore representing mature AMs, and is specifically absent in DHPS KO mice (**new Extended Data Fig. 20b, c**). These results mirror our flow cytometry data, which show that DHPS KO mice do not lack F4/80⁺ (*Adgre1*) CD11c⁺ (*Itgax*) macrophages, but rather lack fully developed Siglec-F⁺ RTM (**Fig. 1d, f**).

From our original scRNA-seq results in peritoneal macrophages (**Fig. 3a-e**), we previously proposed that the enriched intermediate clusters in DHPS KO mice (such as cluster 2), which were *not* TIM4⁺, could represent macrophages blocked in their differentiation towards RTMs. In further analyzing our new lung scRNA-Seq data (**new Extended Data Fig. 22**), we identified two clusters, 1 and 10, that are almost exclusively found in Dhps KO mice. We questioned whether these clusters would represent a population of immature lung macrophages blocked in their trajectory towards mature RTM, analogous to what we observe in the peritoneum. To test this hypothesis, we identified the top upregulated DEGs of cluster 2 (**Fig. 3a, b** - the main intermediate cluster) from our peritoneal macrophage dataset and searched for the same signature in our new lung macrophage scRNA-Seq dataset. Indeed, this signature mapped to clusters 1 and 10 in the lung, supporting that the cells which we proposed to be blocked in their differentiation in one tissue, can be observed in a second tissue. We interpret these data to indicate that these cells represent a transitional state from immature macrophages to RTMs that is independent of the tissue niche. When DHPS is absent, whether in the peritoneal cavity or the lung, these cells fail to fully acquire the RTM signature imposed by the tissue and remain as immature

macrophages. For completeness, we also include a heat map showing the DEGs across clusters when comparing control and DHPS KO macrophages in the lung (**new Extended Data Fig. 21**), which is the same analysis we provided in **Extended Data Fig. 16** for scRNA-seq from the peritoneal macrophages.

Importantly, these new results comprehensively support our conclusion that DHPS is required for macrophages to differentiate into mature RTMs across tissues. We thank the reviewer for suggesting that we perform this experiment in an additional tissue, and agree that it was needed to further strengthen our paper.

Referee #2 (Remarks to the Author):

The revised manuscript is greatly improved and much more focused.

The main findings are centered on the essential role of hypusination in the differentiation of macrophages into tissue resident cells.

The statistics, data, presentation, and conclusions are improved, as is the clarity of presentation.

I only have a couple of minor suggestions:

2.1 All protein names should be all caps, see line 236 where "encodes for Slc7a2" should be written as SLC7A2. All genes should be italicized, see line 231 where "Folr2" should be corrected.

Thank you for highlighting this, we have made the pertinent corrections in the text.

2.2 I do not see any method description for the active cascade-3 assays; the antibody information and method is needed; I assume this was by flow cytometry but I do not see where this is mentioned.

Thank you for pointing this out, we have added the antibody information and method information to the methods section.

Keith T. Wilson

Referee #3 (Remarks to the Author):

The authors have made a substantial effort to address many of the reviewer concerns raised in the first version of the manuscript. They provide additional experimental data, extend their analyses across more tissues, and perform deeper transcriptomic and proteomic assessments. However, they also remove functional data. Further, several key concerns remain insufficiently addressed, and fundamental conceptual issues persist regarding the interpretation of the data and the strength of the conclusions.

3.1 "These data illustrate that Dhps-ΔM mice maintain open RTM niches in adulthood." This is a very bold statement that is only deduced from flow cytometry data. The authors should provide counts of Iba1+ macrophages stained in situ across all tissues to support this statement. A lower expression of e.g. Tim4 does not necessarily mean that these macrophages are not tissue resident.

We thank the reviewer for inviting us to clarify these remarks. In response, we performed in situ staining across many tissues and have added additional conceptual explanation below as to how we view the niches in DHPS-ΔM mice based on our existing data and what "open" and "closed" niches are understood to mean in the RTM field (see Liu and Ginhoux *Cell* 178:1509-1525 2019). Since points 3.1-3.4 center on IF images, we will provide our imaging data in new Extended Data Figures (representative images and their quantification) and Reviewer only Figures (showing whole tissue images across all organs) at the end of point 3.4, but describe our thoughts regarding each point individually. We quantified Iba1+ cells across whole tissue sections in multiple organs and summarized these results in **new Extended Data Fig. 6**. We use a variety of pan-macrophage markers (Iba1, F4/80, and CD64) in our staining panels for IF images (**new Extended Data Figs. 5, 7, 8, 9, 13, and 15**).

We recognize that our original statement may have implied that an "open niche" simply refers to a numerical cell empty niche. We would like to clarify why we feel this statement is accurate in the context of our data and seminal prior studies that have established this nomenclature in response to the finding that tissues have differential dynamics when it comes to monocytic input to the RTM reservoir over time and under steady state conditions. Multiple studies using fate-mapping mouse models have demonstrated that, in early adulthood, bone marrow-derived macrophages contribute little to none of the resident macrophage population in the steady state in most organs (please see Liu, Z et al. Fate mapping via Ms4a3-expression history traces monocyte-derived cells (2019) *Cell* 178, 1509–1525.e19; and Yona, S. et al. Fate mapping reveals origins and dynamics of monocytes and tissue macrophages under homeostasis. (2013) *Immunity* 38, 79-91). The brain, epidermal Langerhans, and liver exhibit the lowest contribution, whereas the gut exhibits the highest contribution due to its rapid macrophage turnover. As such, the gut is said to be an "open" RTM niche, whereas the brain, epidermis, and liver are said to be "closed". In DHPS-deficient mice, we observed by flow cytometry that total macrophage numbers are not uniformly altered across tissues. This indicates that the anatomical space within the niche remains occupied by macrophages, but not by RTMs as shown in **Figure 1f** and **Extended Data Fig. 2-4**, and we hypothesized that this deficiency drives the recruitment of circulating monocytes to repopulate the missing RTMs. Ultimately, we believe this represents a futile cycle, where incoming monocytes attempting to refill the RTM pool get blocked along the differentiation trajectory towards mature RTM, remaining in an immature state, but one in which they express pan-macrophage markers. This inability to acquire the local tissue signature leads to their loss from the tissue, signaling to the bone marrow to mobilize monocytes, akin to what is observed in clodronate or diphtheria toxin models of macrophage depletion.

To conclusively address the dynamics of the RTM niche in DHPS- Δ M mice, we performed parabiosis studies that would allow us to investigate input from the blood and bone marrow to the RTM pools (**Fig. 2c** and **Extended Data Fig. 11a-e**). In DHPS-deficient mice, across multiple adult tissues, there is a significant contribution of wild-type cells derived from circulating monocytes of the WT parabiont to the DHPS- Δ M RTM reservoirs. It is under these circumstances that we claim the DHPS- Δ M niches to be “open”. In contrast, in WT mice, these tissues do not exhibit monocytic contribution from the opposing parabiont, and thus according to the nomenclature of the field would be said to be “closed”. The niche hypothesis is an active area of investigation (see Guilliams & Scott, *Nat Rev Immunol* 17, 451–460, 2017. <https://doi.org/10.1038/nri.2017.42>). The exact signals and components of the tissue niche that drive RTM persistence across tissues remain to be fully defined.

We agree that decreased or absent expression of TIM-4 or Siglec-F in macrophages does not, by itself, prove that these cells are not RTMs. However, these markers are widely recognized as reliable indicators that cells have acquired a long-lived tissue-resident phenotype (see referenced paper and its figure below as just one example). Consistent with this, our previous single-cell RNA-seq dataset of peritoneal macrophages, and our new single-cell RNA-seq dataset of lung macrophages (also discussed in response to Rev# 1) support this conclusion. In the peritoneum, Tim4 expression, along with multiple other genes, defines a distinct cluster of long-lived RTMs that is clearly different between WT and DHPS KO mice (**Fig. 3c**). Similarly, in the lung, Siglec-F marks alveolar macrophages in a unique cluster together with other genes in WT mice—a cluster that is completely absent in DHPS KO mice (**new Extended Data Fig. 20 and 22**). Thus, although our initial conclusions were based on flow cytometry using a limited set of markers, these markers remain the most reliable tools currently available to study RTMs, and our findings are now validated at the single-cell level in two distinct tissues.

Scott et al. (*Nat Commun* 7, 10321, 2016. <https://doi.org/10.1038/ncomms10321>) demonstrated that newly recruited liver macrophages progressively acquire Tim4 expression several weeks after depletion with DT (Figure 5h from their paper below), supporting its role as a marker of tissue-resident identity:

Additionally, we go on to define the functional capacity of the macrophages that are present in DHPS KO mice, and we show that they are defective in central features of RTM: proliferative self-renewal (**Fig. 2h** and **Extended Data Fig. 11h-k**), persistence (**Fig. 2 j-l** and **Fig. 2n-o**), efferocytosis capacity (**Fig. 5a, f, g**), and maintenance of tissue homeostasis (**Fig. 5b, d, e**).

3.2 "Overall these data show a collapse in the tissue-residency potential": is it now an empty niche or the lack of self maintenance that KO macrophages have? All approaches, including the clodronate and parabiosis experiments, should be supported by IF stainings for Iba1+/F4/80+/CD64+ macrophages. It

would also be advisable to combine this with apoptosis markers, as the isolation of macrophages for flow cytometry, as done in the manuscript, can artificially increase apoptosis, particularly affecting more vulnerable populations such as KO macrophages in this study.

As described above, the concept of an “empty niche” in our model is directly related to the inability of DHPS-deficient macrophages to self-maintain within the tissue, which is sensed by the bone marrow as empty, as supported by our parabiosis data that reveal a continual flux of monocytes into the tissues of DHPS KO mice that is likely a futile attempt to refill the RTM reservoir. As shown in **Fig. 2a**, RTMs progressively decline with age, leading to a continuous replacement by monocyte-derived macrophages that fail to acquire full long-lived tissue residency. Consequently, these cells remain short-lived and thus spur on constant replenishment from monocytes. This concept is further supported by our clodronate liposome experiments, in which macrophages were unable to repopulate the RTM niche when DHPS was not expressed, leading to a persistent influx of monocytes to the tissue site.

We have now altered the text in the paper where we used the word ‘empty’ to better clarify what our data show:

...bone marrow chimeras revealed that bone marrow progenitors from CD45.2+ *Dhps*-ΔM mice failed to repopulate ~~empty RTM niches~~ **RTM pools** in irradiated WT recipient mice when competing with CD45.1+ *Dhps*-WT bone marrow...

Parabiosis experiments were conducted in collaboration with Dr. Florent Ginhoux’s laboratory in Singapore, and we are unable to repeat these experiments to perform the suggested imaging. However, to complement our findings, and as suggested by this reviewer, we performed imaging of DHPS LysM-Cre RosaYFP reporter mice and control mice to quantify total macrophages and RTMs across tissues (**new Extended Data Figs. 5-9, 13, 15 and Reviewer only Figs. 1-7**). Our new imaging results closely mirror our flow cytometry data (**Fig. 1f**), showing that total macrophage numbers remain unchanged, while RTM populations are decreased in DHPS-deficient mice.

One limitation of imaging-based quantification, particularly in the kidney, is that kidney-resident macrophages (KRM) are defined by their expression of CD11c and CD11b. As described by Puranik et al. (*Sci Rep* 8, 13948, 2018; <https://doi.org/10.1038/s41598-018-31887-4>): “We classified kidney macrophages as CD11c^{hi}Mφ (CD11b^{hi}CD11c^{hi}), CD11c^{hi}Mφ (CD11b^{hi}CD11c^{lo/neg}), and kidney-resident macrophages (KRM) (CD11c^{int}CD11b^{int}).” A figure from this paper is shown here:

In our flow cytometry data, we observed a decrease in KRMs as defined by CD11c^{int}CD11b^{int} (**Fig. 1f** and **Extended Data Fig. 3a**). However, these subtle changes in CD11c and CD11b expression cannot be accurately captured by immunofluorescence, which limits our ability to quantify KRM populations

through imaging (**new Extended Data Fig. 9b**) in the LysM Cre mouse model. However, using the CX₃CR1-ERT₂^{Cre} R26^{YFP} mouse model, which allow us to accurately label KRM (CD11c^{int}CD11b^{int}) after *in vivo* tamoxifen administration, we could demonstrate that when DHPS is deleted in these cells, the number of YFP+ KRMs per area is decreased versus YFP+ controls over time (**new Extended Data Fig. 13**).

We also performed IF staining for active caspase 3 in KRM and observed a trend toward an increased proportion of caspase-3 positive cells in the DHPS KRM (**Extended Data Fig. 14**) after only 5 weeks post-tamoxifen treatment.

3.3 The authors now provide a few IF pictures, in the kidney only 3D rendered cells (include marker used in the figure). However, from these pictures (especially in the Extended data) I cannot see a reduction of macrophages (i.e. "an empty niche"). Especially in the lung there seems to be even more and enlarged alveolar macrophages. Microglia are certainly less ramified (please quantify morphometry across all tissues). But if cell bodies would be counted, there wouldn't be much less in the KO than in the WT from the picture that has been provided. Please provide large tissue sections in the supplements showing Iba1 expression (e.g. sagittal or coronal brain section, whole liver and whole lung lobe, sagittal kidney section etc.).

We have now included quantification of YFP+ kidney macrophages from the same images used for morphological analysis (**new Extended Data Fig. 13**). DHPS^{flox/flox} CX₃CR1-ERT₂^{Cre} R26^{YFP} mice and DHPS^{+/-} CX₃CR1-ERT₂^{Cre} R26^{YFP} control mice were treated with tamoxifen *in vivo*, and cells were quantified five weeks after tamoxifen administration. Our results show a significant decrease in kidney resident macrophages in DHPS^{flox/flox} mice following Cre induction, supporting our flow cytometry data (**Fig. 2j-l**) and confirming by imaging that these cells lose their ability to self-maintain when DHPS is acutely deleted.

We also quantified total F4/80⁺ macrophages in the lung of LysM-Cre R26^{YFP} mice and found a decrease in total macrophages in DHPS-deficient mice compared to controls (**new Extended Data Fig. 8**). Anti-Siglec-F antibody was tested, but no positive signal could be detected in any of the conditions tested, therefore we were unable to visualize and quantify mature alveolar macrophages in these sections.

For microglia, we quantified morphology as requested and observed that DHPS-deficient microglia exhibit a more spherical morphology with reduced ramification (**new Extended Data Fig. 25e**). Additionally, we repeated imaging following acute deletion of DHPS in microglia using tamoxifen-treated CX₃CR1-ERT₂^{Cre} R26^{YFP} mice. We observed a decrease in total IBA1⁺YFP⁺ cells in DHPS^{flox/flox} mice compared to control mice, confirming our flow cytometry findings that numbers of microglia decline upon DHPS deletion (**new Extended Data Fig. 15**). Of note, compared to other tissues, this reduction requires a longer period post-deletion to become apparent (11–40 weeks in our model), a phenotype we are continuing to study in our lab.

3.4 "there are few liver RTMs in Dhps-dM mice": please provide IF pictures of Iba1/F4/80 to be able to conclude this.

Our conclusion that the liver of DHPS-ΔM mice have fewer RTMs was based on total counts of TIM4⁺ macrophages per mg of tissue by flow cytometry (**Fig. 1f**). Liver RTMs are commonly defined as F4/80⁺ CD64⁺ TIM-4⁺ CLEC4F⁺. Iba1 and F4/80 are pan-macrophage markers that detect total macrophages within tissues, but cannot specifically distinguish RTMs. To address this request, we performed imaging of liver sections from DHPS^{flox/flox} LysM-Cre Rosa26^{YFP} reporter mice and control mice. We quantified total F4/80⁺ macrophages per area, F4/80⁺YFP⁺ (Cre⁺) macrophages per area, and F4/80⁺YFP⁺TIM4⁺ RTMs per area. Our results show that total F4/80⁺ macrophage numbers are unchanged, but within the F4/80⁺ population targeted by LysM-Cre, there is a clear reduction in liver RTMs when DHPS is deleted, as read out by reduced abundance of TIM-4⁺ macrophages (**new Extended Data Fig. 8a**).

To further clarify how the field of tissue-resident macrophage biology has evolved, we included a paragraph below from a recent review emphasizing the importance of using markers that discriminate total macrophages (e.g., F4/80⁺) from true RTMs when studying the relevance of transcription factors (TFs) for the development of these cells and how we should revisit previous studies that have reported no phenotypes based only on F4/80.

From T'Jonck W, Guilliams M, Bonnardel J. Niche signals and transcription factors involved in tissue-resident macrophage development. Cell Immunol. 2018;330:43-53. doi:10.1016/j.cellimm.2018.02.005:

"ID1 and ID3 are both TGFβ-regulated TFs [66] and their expression pattern largely overlaps during embryogenesis [67], which could indicate that TGFβ might be one of the niche factors that drives Kupffer cell development. It is important to note that this study only relied on F4/80 to identify Kupffer cells. However, it was recently described that the F4/80hi macrophage population in the liver can be subdivided in three maturation states based on expression of CLEC4F and TIM4 with bona fide Kupffer cells being CLEC4F+TIM4+ [64]. Other TFs linked to tissue-resident macrophage development include LXRα and SPIC, although these might be necessary for functional specialization, as no apparent defect in Kupffer cell abundance has been described in KO models involving these TFs [45,55]. This was however also concluded based on F4/80 expression, so it may be required to revisit these findings using more specific Kupffer cell markers such as CLEC4F and TIM4."

Whole-tissue sections from all organs analyzed by imaging are provided as **Reviewer only Figs. 1-7 below**. Quantification of whole-tissue sections and representative images are available as new **Extended Data Figs. 5-9, 13, and 15** and referred to in the main text.

Reviewer only Fig. 1

Liver: Dhps +/+ LysM Cre Rosa26eYFP (Control) vs Dhps flx/flx LysM Cre Rosa26eYFP (Dhps-ΔM Rosa26eYFP).

Reviewer only Fig. 2

Spleen: Dhps $+/+$ LysM Cre Rosa26eYFP (Control) vs Dhps flx/flx LysM Cre Rosa26eYFP (Dhps- Δ M Rosa26eYFP).

Reviewer only Fig. 3

Lung: Dhps +/+ LysM Cre Rosa26eYFP (Control) vs Dhps flx/flx LysM Cre Rosa26eYFP (Dhps-ΔM Rosa26eYFP). Siglec-F could not be visualized by IF.

Reviewer only Fig. 4

Spleen: Dhps $+/+$ LysM Cre Rosa26eYFP (Control) vs Dhps flx/flx LysM Cre Rosa26eYFP (Dhps- Δ M Rosa26eYFP).

Reviewer only Fig. 5

Brain: Dhps +/+ LysM Cre Rosa26eYFP (Control) vs Dhps flx/flx LysM Cre Rosa26eYFP (Dhps-ΔM Rosa26eYFP).

Reviewer only Fig. 6

Heart: Dhps +/+ LysM Cre Rosa26eYFP (Control) vs Dhps flx/flx LysM Cre Rosa26eYFP (Dhps-ΔM Rosa26eYFP).

Reviewer only Fig. 7

Heart: Dhps +/- LysM Cre Rosa26eYFP (Control) vs Dhps flx/flx LysM Cre Rosa26eYFP (Dhps-ΔM Rosa26eYFP).

DHPS WT

DHPS KO

3.5 Since LysM-Cre labels also monocytes, neutrophils and dendritic cells, the authors should correct their sentence "the pulldown was exclusively HA-tagged macrophages." This would also mean that even 5–10% of transcripts coming from other cell types would have a big impact on DEGs (see point 3.6). Therefore, to convince the reader that the few DEGs are indeed macrophage-specific, one would need to show WT and KO gating strategy, including the labelling of other myeloid cells, before showing the histogram for ST2 expression.

We thank the reviewer for pointing this out and have changed the text to accurately reflect what the data show:

"...the pulldown was exclusively HA-tagged LysM-expressing cells..."

In previous Reviewer only Fig. 6 from the first round of revision (shown at the end of this document as **First revision Reviewer only Fig. 6**), we demonstrated that when mice are treated with clodronate liposomes, infiltrating neutrophils do not express appreciable ST2 by flow cytometry as compared to peritoneal macrophages and infiltrating monocytes in the peritoneal cavity. While it is true that monocytes could partially contribute to the observed effect, we validated by protein expression that ST2 is indeed decreased specifically in peritoneal macrophages, which supports the validity of our results. Furthermore, in **First revision Reviewer only Fig. 6c**, we showed that culturing *ex vivo* isolated peritoneal macrophages from CX3CR1-ERT2Cre Rosa26YFP mice with 4-OHT (4-hydroxytamoxifen) to induce Cre expression and *Dhps* deletion also leads to decreased ST2 expression in YFP+ peritoneal macrophages that were previously WT, further strengthening our conclusion that DHPS regulates ST2 expression in macrophages.

As requested here by the reviewer, we reanalyzed the data shown in **Fig. 4b** (expression of IL33R/ST2 by peritoneal macrophages) to quantify receptor expression in other myeloid populations (**Reviewer only Fig. 8a**). Since our experiments primarily focused on macrophages, we did not include antibodies to detect Ly6C and Ly6G at the time that would allow us to identify monocytes and neutrophils, respectively. Therefore, we based our analysis on F4/80-negative cells and CD11b SSC high or SSC low populations to distinguish monocytes and neutrophils. Additionally, we reanalyzed the data shown in **First Revision Reviewer only Fig. 6b**, where we used the LysM-Cre R26YFP reporter to assess IL33R/ST2 expression again in peritoneal macrophages (**Reviewer only Fig. 8b**). Both analyses consistently demonstrate that differences in IL33R/ST2 expression, at least in terms of the data we present, are macrophage-specific, as CD11b+ SSC high or SSC low cells did not show downregulation of the receptor. Therefore, we conclude that the decreased IL33R/ST2 mRNA transcripts observed in our RiboTAG pulldown experiment, where the pulldown was performed exclusively in LysM-Cre-expressing cells, accurately reflect the downregulation occurring in macrophages.

Reviewer only Fig. 8

a

DHPS flox/flox LysM Cre - Peritoneal lavage

b

DHPS flox/flox LysM Cre⁺ R26YFP vs DHPS +/+ LysM Cre⁺ R26YFP - Peritoneal lavage

Reviewer only Fig. 8. DHPS-deficient Peritoneal macrophages specifically show decreased expression of IL33R/ST2 in LysM-Cre mice with no changes on this receptor in other CD11b⁺

subpopulations. a) Peritoneal exudate cells were isolated from DHPS flox/flox LysM-Cre negative (WT) or Cre positive (DHPS-dM) and IL33R/ST2 was assessed within CD11b+ subpopulations by flow cytometry. IL33R/ST2 expression is shown as MFI within each gate. **b)** Peritoneal exudate cells were isolated from DHPS +/- LysM-Cre positive (Control) or DHPS flox/flox LysM-Cre positive (DHPS-dM) and IL33R/ST2 was assessed within YFP+CD11b+ subpopulations by flow cytometry. IL33R/ST2 expression is shown as MFI within each gate.

3.6 Supplementary tables should contain the list of all genes and their normalized expression. This is also true for all other RNAseq experiments, where for now only genes from certain GO terms are shown (in line with the reviewer #2 point 2.6). The reviewers should have a chance to assess the raw data in Excel Tables, including all genes and GO terms, not only the once chosen for the figures. It would be useful to plot heatmaps/dotplots/violin plots to show expression of relevant genes across all experiments, instead of only showing GO terms.

As requested, we have now added supplementary tables containing the differentially expressed genes (DEGs) from the bulk RNA-seq data set (**new Supplementary Tables 5 and 7**). The proteomics and single-cell RNA-seq (scRNA-seq) data were presented in tables in our first revision (**Supplementary Tables 1-4**). We feel that provide Excel Tables for the scRNA-seq shown in Fig. 3, which would mean 22 additional Tables, is somewhat impractical, however, volcano plots for all clusters from the scRNA-seq analysis of peritoneal macrophages are shown in **Extended Data Fig. 17**. These DEGs were subsequently used for pathway analysis, shown in **Extended Data Fig. 18**.

All of our raw sequencing and proteomics data are deposited and are immediately available to the reviewers upon request. Our newly generated scRNA-seq data from lung macrophages will also be deposited and made available, but with the current government shutdown we are unable to deposit the data at this time. Having the data available to everyone will allow readers to perform analyses and assess all the results as they see fit.

3.7 If ST2 is indeed a DHPS-dependent molecule required for macrophage maturation, its expression should be also shown across all tissues analysed (both flow cytometry and IF).

We understand what the reviewer is saying here, but in this manuscript we do not wish to conclude that ST2 is the sole critical molecule required for all macrophage maturation across all tissues. We provide data that suggests that ST2 is DHPS-dependent and it is expressed in peritoneal macrophages and that this, along with other hypusine-dependent transcripts, are likely important for macrophage differentiation in this tissue.

First revision Reviewer only Fig. 6b showed that DHPS-KO macrophages in both the lung and brain also exhibit decreased ST2 expression. We do not want make claims to the importance of this molecule in these tissue sites, but other papers have shown its role in macrophages (below are some examples). There are other labs that are investigating the critical factors that regulate RTMs in distinct niches and we do not wish to overextend our findings regarding ST2 as a critical factor in every niche in this paper.

Lu Y, Basatemur G, Scott IC, *et al.* Interleukin-33 Signaling Controls the Development of Iron-Recycling Macrophages. *Immunity*. 2020;52(5):782-793.e5.

He D, Xu H, Zhang H, *et al.* Disruption of the IL-33–ST2–AKT Signaling Axis Impairs Neurodevelopment by Inhibiting Microglial Metabolic Adaptation and Phagocytic Function. *Immunity*. 2022;55(1):159-173.e9.

Dagher, R., Copenhaver, A.M., Besnard, V. *et al.* IL-33-ST2 axis regulates myeloid cell differentiation and activation enabling effective club cell regeneration. *Nat Commun* **11**, 4786 (2020).

Additionally, our results demonstrate that ST2 expression is downregulated, but not completely lost, making immunofluorescence a less desirable method for quantifying this effect. We do not observe a distinct population of ST2-negative macrophages in DHPS-deficient mice; rather, we see a reduction in ST2 expression levels (MFI) within these cells (**Fig. 4b** and in **First revision Reviewer only Fig. 6**, shown at the end of this document).

3.8 To my knowledge, ST2 KO mice do not have reduced macrophage numbers. Do the authors have an explanation for their phenotype beyond ST2? (see also point 3.8, which in my opinion should be addressed in the context of the current study)

It has been shown that ST2 (*Il1rl1*) knockout mice exhibit decreased numbers of red pulp macrophages (RPMs), which are the embryonically derived RTMs in the spleen:

Lu Y, Basatemur G, Scott IC, *et al.* Interleukin-33 Signaling Controls the Development of Iron-Recycling Macrophages. *Immunity*. 2020;52(5):782-793.e5. doi:10.1016/j.immuni.2020.03.006, quantification of these cells is presented in Figure 5A, demonstrating by flow cytometry a decrease in F4/80^{hi}CD11b^{lo/-} cells, as well as by imaging of spleen sections stained for F4/80. Their results (shown below here in panels A and B) are consistent with our new spleen imaging data (**new Extended Data Fig. 5**), which show a decrease in RPMs when DHPS is deleted in YFP⁺ (LysM-Cre⁺) cells compared with controls.

In another report:

He D, Xu H, Zhang H, *et al.* **Disruption of the IL-33–ST2–AKT Signaling Axis Impairs Neurodevelopment by Inhibiting Microglial Metabolic Adaptation and Phagocytic Function.** *Immunity.* 2022;55(1):159-173.e9. doi:10.1016/j.immuni.2021.12.001, the authors characterized the effects of ST2 knockout in microglia, using both whole-body KO mice and ST2^{fl^{ox}} mice crossed with CX3CR1-Cre. Although they provided imaging of Iba1⁺ cells in the brain (see panel L below), quantification was performed only in WT mice treated with IL-33, and no quantification was provided for their ST2 KO models. As a result, these data do not allow us to conclude whether ST2 is required for maintaining RTM numbers in the brain. Nevertheless, the authors clearly demonstrate in their Figure 2L that ST2-deficient microglia display altered morphology, which closely mirrors our own findings in DHPS-deficient microglia, where we observe reduced ramification and a more spherical morphology (**new Extended Data Fig. 25e**). Additionally, they observed decreased efferocytic capacity and decreased mitochondrial metabolism in these cells. Phenotypes that are also observed in DHPS-deficient macrophages.

To the best of our knowledge, the spleen remains the only tissue where RTM abundance has been conclusively assessed in ST2 KO mice. Most studies do not report steady-state macrophages numbers (i.e. no injury or disease), and most are comparing macrophages in response to injury, fibrosis, or stress. Further, most other studies do not distinguish between RTM and infiltrating monocyte-derived macrophages, and measure F480+ macrophages or use markers like CD206 to define them as M1 or M2.

Regarding how we explain our observed phenotype in addition to ST2, we paraphrase text from our paper:

“Overall, these data suggest that ST2 is amongst a subset of ribosome engaged transcripts in peritoneal macrophages that is dependent on hypusinated-eIF5A for translation, and as such, its reduced protein expression contributes to the defect in the monocyte to RTM transition in the peritoneal cavity observed in *Dhps*-ΔM mice.

... rather than any sole gene, likely contribute to the complex in vivo program of RTM differentiation. Further, while the lack of ST2 expression may be a determining factor for the defect in RTM differentiation in DHPS-deficient peritoneal macrophages, ST2 may not have such a prominent role in the RTM transition of macrophages in other tissues. Since *Dhps*-ΔM mice manifested a defect in all RTM we measured thus far, we sought to understand which pathways might be critical to the differentiation of macrophages in all tissues. Therefore, we performed bulk RNA-sequencing on F4/80⁺ macrophages (gating strategies in **Extended Data Fig. 24a-c**) isolated from 3 distinct tissues in *Dhps*-WT and *Dhps*-ΔM mice: the lung, liver, and the peritoneal cavity (**Fig. 4e, f**). Pathway analysis of DEGs that were common amongst all DHPS-deficient macrophages from each tissue revealed cell adhesion, signaling, and migration pathways as the most significantly downregulated (**Fig. 4g** and **Supplementary Tables 5, 6**), with inflammatory response and apoptotic pathways increased (**Extended Data Fig. 24d** and **Supplementary Tables 7, 8**). Notably, our sequencing of ribosome transcripts revealed several genes implicated in cell adhesion (**Fig. 4a**). For example, *Tnik* is critical for augmenting expression of target genes in the Wnt/β-catenin pathway that controls cell adhesion^{51,52}, and as such, β-catenin-deficient macrophages lack cell adherence capacity⁶⁴. The genes encoding for the cell adhesion molecules L1CAM (*11cam*) and E-Cadherin (*cdh1*) were downregulated in the pathway analysis of DHPS-deficient macrophages isolated from 3 distinct tissues (**Supplementary Table 6**), which correlated with decreased protein expression (**Fig. 4h**), and with earlier work showing that E-Cadherin expression is polyamine dependent in alternatively activated macrophages⁶⁵.

... taken all together, our results indicate that cell adhesion and signaling are critical to RTM development across tissue sites, and that this program is controlled by the polyamine-hypusine axis.”

Another aspect to consider regarding mechanism:

“... it is possible that a specific protein or proteins, such as a transcription factor or chromatin modifying factor, could be dependent on eIF5A for optimal translation, and in this way the transcriptional program required for RTM differentiation, that includes cell adhesion and signaling proteins, is never enacted in DHPS-deficient cells.”

We feel that through our results and discussion we provide the most complete picture possible at this time regarding how the polyamine-hypusine axis controls macrophage tissue residency across tissues.

3.9 *The liver phenotype after clodronate could also be due to impaired neutrophils, which a both targeted by the LysM-cre and the clodronate. As the authors state, this is likely completely KC independent, so I am not sure how the authors deduce last paragraph of the results part. Regarding point 1): this is a well-known fact, and has not been shown by this study. Regarding point 2) : adoptive transfer (WT BM into Dhsp-dM mice) could support this statement. But as outlined above, in the manuscript and the rebuttal letter, it remains unclear if this is at all liver macrophage-specific. Models such as Clec4f-Cre will help to delineate this, at least for the liver.*

We thank the reviewer for pointing this out, which gives us the opportunity to more accurately interpret what our data show. In our experiment (**Fig. 5h-j**) every cell type that expresses LysM will delete DHPS, and this includes all myeloid populations in the mouse. In addition, neutrophils take up CL, and have been shown to be 'stunned' (rather than depleted like is the case for macrophages) by CL (Culemann S, Knab K, Euler M, et al. Stunning of neutrophils accounts for the anti-inflammatory effects of clodronate liposomes. *J Exp Med.* 2023;220(6):e20220525). Thus, as the reviewer highlights, our results do not show that our phenotype is KC-dependent, however, nor do they show that it is KC-independent. Therefore, we have amended our text to accurately reflect what our data show with no further overstatements beyond:

“These data show that myeloid cell DHPS expression is critical for maintaining and/or restoring liver tissue homeostasis after CL administration.

We feel that the results in **Fig. 5h-j** and **Extended Data Fig. 26** support our overall hypothesis in that DHPS expression in the myeloid compartment is critical for tissue homeostasis and with the revised text, importantly, we now do not overstate any claims beyond this conclusion.

As a separate point, we do appreciate the reviewer's comment regarding the *Clec4fcre* mice, as the model to test the role of DHPS in liver RTM, aka Kupffer cells. Our data in **Fig. 5** of the paper show that CL efficiently depletes liver RTM, and that monocyte-derived macrophages enter the liver by 5 days post CL administration. The data also show that it is in this timeframe that the liver is restored to, or maintained at, proper homeostasis in the control mice. Granted, we can only conclude that the 'recovery' of the liver in the control setting is due to myeloid cells, and not specifically due to liver RTM. Notwithstanding, the interpretation of these data is that the liver damage in the DHPS KO mice after CL is dependent on DHPS being deleted prior to the monocytes entering the tissues to become macrophages, or, as the reviewer argues, in neutrophils that are developing in the tissue after CL. In either case, in this experiment, the protective effect of DHPS-sufficient myeloid-derived cells occurs before day 5 post CL.

In response to this reviewer, we purchased *Clec4f cre* mice and began breeding them in our facility. After we obtained the first adult mice we did an experiment to determine when *Clec4f* would be expressed in monocyte-derived macrophages after CL depletion. Our results show that *Clec4f* is not expressed in recently recruited monocyte-derived macrophages 5 days post CL depletion in the liver (**Reviewer only Fig. 9**). Given these results, we conclude that we cannot use these mice to remove DHPS from cells that will undergo the monocyte-to-macrophage transition in the tissue. We could use them to delete DHPS in liver RTM as they begin to express *Clec4f* (which is not until at least 5 days after they enter the tissue – and the figure 5H from the Scott paper shown above in response to comment 3.1 indicates that *Clec4f* is not robustly expressed on monocyte-derived macrophages until well beyond d30 after entering the tissue) and then administer CL to determine if DHPS expression in the mature (or maturing) liver RTM are important for maintaining tissue homeostasis. Importantly, this is a different question from the one we address in **Fig 5**, where DHPS is deleted in monocytes, prior to the monocyte to macrophage transition in the repopulated tissue. We will continue these experiments as part of our future studies, and as noted above, we have removed any conclusions regarding cell type specificity, beyond to say that DHPS needs to be expressed in the myeloid compartment, based on our data in **Fig. 5**.

Reviewer only Fig. 9

Reviewer only Fig. 9. Clec4F-Cre is not expressed in recently recruited monocyte-derived macrophages 5 days after depleting liver RTM with clodronate liposomes. *Clec4F-Cre* TdTomato Rosa26^{eYFP} mice were administered clodronate liposomes (CL) intraperitoneally to deplete macrophages. Liver was isolated 5 days post CL administration and tissue processed for flow cytometry. Clec4F-Cre expression was assessed by TdTomato reporter expression and Cre activity was assessed by Rosa26eYFP expression within the F4/80+CD64+ total macrophage population. CD11b MFI was also compared between macrophages from CL-treated mice and control (untreated). Data represent 2 mice per experimental condition.

For ease of reference, we have included here the **Reviewer only Fig. 6a-d** from the first revision:

First revision Reviewer only Fig. 6

Referees' comments:

We would like to thank all the reviewers for taking the time to thoughtfully review our work and provide insightful comments. We have carefully addressed each point from Reviewer 3 and have fully agreed to make the suggested changes in the manuscript to better reflect our data and conclusions. We believe that the revisions made in response to the reviewers' feedback have significantly strengthened our manuscript, and we hope that this revised version reflects those improvements.

Referee #3 (Remarks to the Author):

I appreciate the additional work and clarifications provided in this revised version. The manuscript has certainly improved, and several of the earlier points have been addressed. However, I still have a major conceptual disagreement with the authors' interpretation of an "empty macrophage niche," which remains a central claim in the paper.

I would strongly urge the authors to reconsider this interpretation, as the data presented do not support the idea of an "empty" niche. The notion that a macrophage niche becomes "empty" implies that the tissue lacks macrophages altogether, which is clearly not the case here. The authors show that $Iba1^+$ and $F4/80^+$ macrophage numbers remain unchanged between wild-type and knockout mice. This demonstrates that macrophages continue to occupy the tissue space, and therefore the niche is not empty. The absence of $Tim4$ expression does not define the absence of a macrophage niche—it merely reflects that these macrophages have not acquired the full tissue-resident Kupffer cell-like program.

Furthermore, the authors' interpretation misrepresents the concept of "open" and "closed" tissues. As discussed extensively in the field, this dichotomy has evolved significantly since the 2016 review the authors cite in the rebuttal letter. We now know that even tissues previously considered "open," such as the gut, contain a substantial fraction of long-lived, yolk sac-derived macrophages. The current understanding is therefore that macrophage niches are dynamic but continuously occupied; they may change in cellular composition or turnover rate, but they are not "empty."

I recommend that the authors revise this conceptual framing throughout the manuscript. The phenotype they observe can be more accurately explained as a failure of maintenance or maturation, likely due to impaired DHPS-dependent longevity or differentiation capacity. The macrophages that occupy the Kupffer cell niche in the knockout mice are thus distinct in ontogeny or turnover dynamics, but they are still macrophages. Describing this situation as an "empty niche" risks confusing the reader and contradicts the current consensus in macrophage biology.

In summary, I encourage the authors to:

- Remove the use of the term "empty niche" and related phrasing implying complete absence of macrophages.

*This clarification is important and we thank the reviewer for making this point. We have ensured that the final version of our manuscript is absent of the terms *empty or open niche*, and also does not imply an absence of macrophages.*

- Reframe the interpretation in terms of impaired maintenance, reduced longevity, or altered differentiation within an occupied macrophage niche.

We have reframed the interpretations in the original manuscript using the suggested terms. These changes can be found in **yellow highlighted sections** of the word doc:

Abstract

“Mice with myeloid cell deletions in DHPS (*Dhps*-ΔM mice) had a global defect in RTM across tissues, **resulting in persistent**, but ultimately futile, monocyte influx. Transcriptional analyses of DHPS-deficient macrophages indicated a block in their ability to **differentiate into mature RTM**, while proteomics revealed defects in cell adhesion and signaling pathways.”

“Together, our results demonstrate a novel cell-intrinsic, tissue-agnostic, pathway **that drives the differentiation of monocyte-derived macrophages into RTM.**”

Main

Page 4:

“However, unlike in control mice, we observed a persistent monocyte presence in the peritoneal cavity of *Dhps*-ΔM **mice that we reasoned reflected sensing of a lack of mature TIM-4⁺ RTM (Fig. 2b)**”

Page 5:

Title change: “**Mature RTM persistence requires DHPS**”

Page 6:

“**We hypothesize that cluster 1 represents monocyte-derived macrophage infiltration triggered by a scarcity of mature RTM in *Dhps*-ΔM mice.**”

Figure 2 title:

Fig. 2. DHPS is essential for monocyte to RTM maturation and macrophage survival

- Clarify that the tissue remains populated by macrophages (*Iba1*⁺/*F4/80*⁺), albeit with different identity and stability.

These changes have been made and can be found **in yellow highlighted sections** of the word doc:

Main

Page 4:

“These data illustrate that the tissues of *Dhps*-ΔM mice, **despite being populated with F4/80⁺ macrophages, receive continual monocytic influx that we hypothesize is driven by a dearth of fully developed RTM.**”

Addressing this conceptual issue will make the paper more accurate and much clearer for the field.